# SoTTA: Robust Test-Time Adaptation on Noisy Data Streams

**Taesik Gong**[†][*]  **Yewon Kim**[‡][*]  **Taeckyung Lee**[‡][*]  **Sorn Chottananurak**[‡]  **Sung-Ju Lee**[‡]

[†]Nokia Bell Labs  [‡]KAIST

taesik.gong@nokia-bell-labs.com
{yewon.e.kim,taeckyung,sorn111930,profsj}@kaist.ac.kr

## Abstract

Test-time adaptation (TTA) aims to address distributional shifts between training and testing data using only unlabeled test data streams for continual model adaptation. However, most TTA methods assume benign test streams, while test samples could be unexpectedly diverse in the wild. For instance, an unseen object or noise could appear in autonomous driving. This leads to a new threat to existing TTA algorithms; we found that prior TTA algorithms suffer from those noisy test samples as they blindly adapt to incoming samples. To address this problem, we present Screening-out Test-Time Adaptation (SoTTA), a novel TTA algorithm that is robust to noisy samples. The key enabler of SoTTA is two-fold: (i) input-wise robustness via high-confidence uniform-class sampling that effectively filters out the impact of noisy samples and (ii) parameter-wise robustness via entropy-sharpness minimization that improves the robustness of model parameters against large gradients from noisy samples. Our evaluation with standard TTA benchmarks with various noisy scenarios shows that our method outperforms state-of-the-art TTA methods under the presence of noisy samples and achieves comparable accuracy to those methods without noisy samples. The source code is available at https://github.com/taeckyung/SoTTA.

## 1 Introduction

Deep learning has achieved remarkable performance in various domains [6, 8, 33], but its effectiveness is often limited when the test and training data distributions are misaligned. This phenomenon, known as domain shift [31], is prevalent in real-world scenarios where unexpected environmental changes and noises result in poor model performance. For instance, in autonomous driving, weather conditions can change rapidly. To address this challenge, Test-Time Adaptation (TTA) [1, 5, 29, 38, 39, 44] has emerged as a promising paradigm that aims to improve the generalization ability of deep learning models by adapting them to test samples, without requiring further data collection or labeling costs.

While TTA has been acknowledged as a promising method for enhancing the robustness of machine learning models against domain shifts, the evaluation of TTA frequently relies on the assumption that the test stream contains only benign test samples of interest. However, test data can be unexpectedly diverse in real-world settings, containing not only relevant data but also extraneous elements that are outside the model's scope, which we refer to *noisy* samples (Figure 1). For instance, unexpected noises can be introduced in autonomous driving scenarios, such as dust on the camera or adversarial samples by malicious users. As shown in Figure 2, we found that most of the prior TTA algorithms showed significantly degraded accuracy with the presence of noisy samples (e.g., $81.0\% \rightarrow 52.1\%$ in TENT [38] and $82.2\% \rightarrow 54.8\%$ in CoTTA [39]).

---

[*]*Equal contribution.*

37th Conference on Neural Information Processing Systems (NeurIPS 2023).

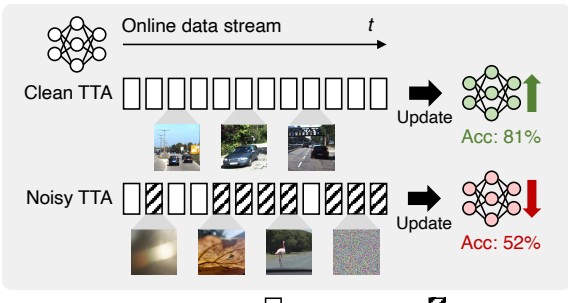

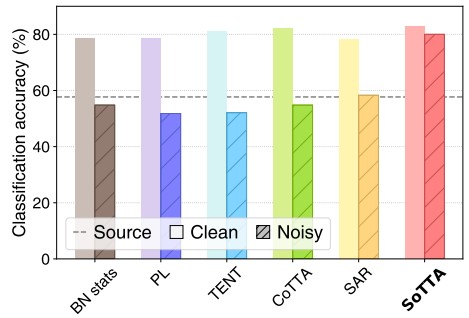

Figure 1: Unlike prior assumptions (Clean TTA), real-world test streams could include unexpected noisy samples out of the model's scope (Noisy TTA), such as glare, fallen leaf covering the lens, unseen objects (e.g., a flamingo), and noise in autonomous driving scenarios. The accuracy of existing TTA methods degrades in such cases.

Figure 2: Average classification accuracy (%) of existing TTA methods and our method (SoTTA) on CIFAR10-C. The performance of existing methods degrades when noisy data are mixed into the test stream (Noisy) compared with the original assumption (Clean). Higher is better.

To ensure the robustness of TTA against noisy samples, an intuitive solution might be screening out noisy samples from the test stream. Out-of-distribution (OOD) detection [10, 11, 18, 19, 20, 21, 24, 25, 43] is a representative method for this, as it tries to detect whether a sample is drawn from the same distribution as the training data or not. Similarly, open-set domain adaptation (OSDA) [30, 35] and universal domain adaptation (UDA) [34, 42] generalize the adaptation scenario by assuming that unknown classes are present in test data that are not in training data. However, these methods require access to a whole batch of training data and unlabeled target data, which do not often comply with TTA settings where the model has no access to train data at test time due to privacy issues [38] and storing a large batch of data is often infeasible due to resource constraints [12]. Therefore, how to make online TTA robust under practical noisy settings is still an open question.

In this paper, we propose Screening-out Test-Time Adaptation (SoTTA) that is robust to noisy samples. SoTTA achieves robustness to noisy samples in two perspectives: (i) *input-wise* robustness and (ii) *parameter-wise* robustness. Input-wise robustness aims to filter out noisy samples so that the model will be trained only with benign samples. We achieve this goal via High-confidence Uniform-class Sampling (HUS) that avoids selecting noisy samples when updating the model (Section 3.1). Parameter-wise robustness pursues updating the model weights in a way that prevents model drifting due to large gradients caused by noisy samples. We achieve this via entropy-sharpness minimization (ESM) that makes the loss landscape smoother and parameters resilient to weight perturbation caused by noisy samples (Section 3.2).

We evaluate SoTTA with three common TTA benchmarks (CIFAR10-C, CIFAR100-C, and ImageNet-C [9]) under four noisy scenarios with different levels of distributional shifts: Near, Far, Attack, and Noise (Section 2). We compare SoTTA with eight state-of-the-art TTA algorithms [1, 17, 27, 28, 29, 38, 39, 44], including the latest studies that address temporal distribution changes in TTA [1, 28, 29, 39, 44]. SoTTA showed its effectiveness with the presence of noisy samples. For instance, in CIFAR10-C, SoTTA achieved 80.0% accuracy under the strongest shift case (Noise), which is a 22.3%p improvement via TTA and 6.4%p better than the best baseline [44]. In addition, SoTTA achieves comparable performance to state-of-the-art TTA algorithms without noisy samples, e.g., showing 82.2% accuracy when the best baseline's accuracy is 82.4% in CIFAR10-C.

**Contributions.** (i) We highlight that test sample diversity in real-world scenarios is an important problem but has not been investigated yet in the literature. We found that most existing TTA algorithms undergo significant performance degradation with sample diversity. (ii) As a solution, we propose SoTTA that is robust to noisy samples by achieving input-wise and parameter-wise robustness. (iii) Our evaluation with three TTA benchmarks (CIFAR10-C, CIFAR100-C, and ImageNet-C) show that SoTTA outperforms the existing baselines.

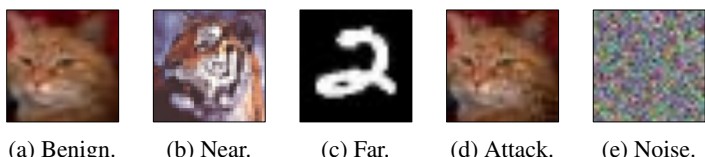

| (a) Benign. | (b) Near. | (c) Far. | (d) Attack. | (e) Noise. |

Figure 3: Five test sample scenarios considered in this work: Benign, Near, Far, Attack, and Noise.

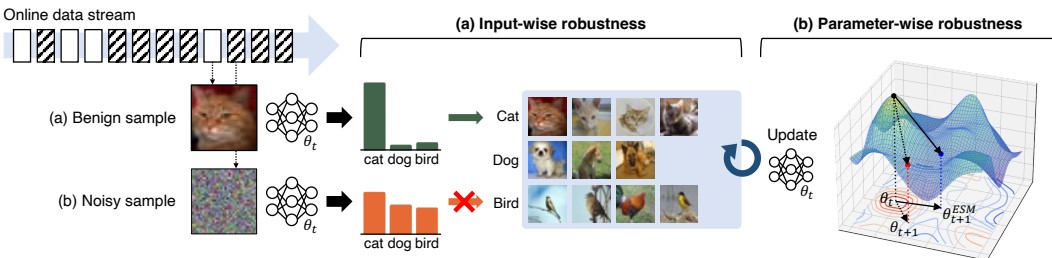

Figure 4: Overview of SoTTA. SoTTA achieves *input-wise robustness* via high-confidence uniform-class sampling (HUS) and *parameter-wise robustness* via entropy-sharpness minimization (ESM).

## 2 Preliminaries

**Test-time adaptation.** Let $\mathcal{D}_\mathcal{S} = \{\mathcal{X}^\mathcal{S}, \mathcal{Y}\}$ be source data and $(\mathbf{x}_i, y_i) \in \mathcal{X}^\mathcal{S} \times \mathcal{Y}$ be each instance and the label pair that follows a probability distribution of the source data $P_\mathcal{S}(\mathbf{x}, y)$. Similarly, let $\mathcal{D}_\mathcal{T} = \{\mathcal{X}^\mathcal{T}, \mathcal{Y}\}$ be target data and $(\mathbf{x}_j, y_j) \in \mathcal{X}^\mathcal{T} \times \mathcal{Y}$ be each target instance and the label pair following a target probability distribution $P_\mathcal{T}(\mathbf{x}, y)$, where $y_j$ is usually unknown to the learning algorithm. The covariate shift assumption [31] is given between source and target data distributions, which is defined as $P_\mathcal{S}(\mathbf{x}) \neq P_\mathcal{T}(\mathbf{x})$ and $P_\mathcal{S}(y|\mathbf{x}) = P_\mathcal{T}(y|\mathbf{x})$. Given an off-the-shelf model $f(\cdot; \Theta)$ pre-trained from $\mathcal{D}_\mathcal{S}$, the (fully) test-time adaptation (TTA) [38] aims to adapt $f(\cdot; \Theta)$ to the target distribution $P_\mathcal{T}$ utilizing only $\mathbf{x}_j$ given test time.

**Noisy test samples.** We define *noisy* test samples to represent any samples that are not included in the target data distribution, i.e., $\tilde{\mathbf{x}} \notin \mathcal{X}^\mathcal{T}$. We use the term noisy to distinguish it from out-of-distribution (OOD), as TTA typically aims to adapt to OOD samples, such as corrupted ones. Theoretically, there could be numerous categories of noisy samples. In this study, we consider five scenarios: Benign, Near, Far[2], Attack, and Noise. Figure 3 shows examples of these scenarios. Benign is the typical setting of TTA studies without noisy samples, Near represents a semantic shift [41] from target distribution, Far is a severer shift where covariate shift is evident [41], Attack refers to intelligently generated adversarial attack with perturbation [40], and Noise refers to random noise. We focus on these scenarios to understand the impact of noisy samples in TTA as well as the simplicity of analysis. Detailed settings are described in Section 4.

## 3 Methodology

**Problem and challenges.** Prior TTA methods assume test samples are benign and blindly adapt to incoming batches of test samples. The presence of noisy samples during test time can significantly degrade their performance for those TTA algorithms, which has not been explored in the literature yet. Dealing with noisy samples in TTA scenarios is particularly challenging as (i) TTA has no access to the source data, (ii) no labels are given for target test data, and (iii) the model is continually adapted and thus a desirable solution should apply to varying models. This makes it difficult to apply existing solutions that deal with a similar problem. For instance, out-of-distribution (OOD) detection studies [11, 19, 20, 43] are built on the assumption that a model is fixed at test time, and open-set domain adaptation (OSDA) methods [30, 35, 42] require labeled source and unlabeled target data for training.

---

[2]We borrowed the term Near and Far from the OOD detection benchmark [41].

**Methodology overview.** To address the problem, we propose Screening-out Test-Time Adaptation (SoTTA), whose overview is described in Figure 4. SoTTA achieves robustness to noisy samples in two perspectives: (i) *input-wise* robustness via high-confidence uniform-class sampling that avoids selecting noisy samples when updating the model (Section 3.1), and (ii) *parameter-wise* robustness via entropy-sharpness minimization that makes parameters resilient to weight perturbation caused by noisy samples (Section 3.2).

## 3.1 Input-wise robustness via high-confidence uniform-class sampling

Our first approach is to ensure input-wise robustness to noisy samples by filtering out them when selecting samples for adaptation. As locating noisy samples without their labels is challenging, our idea is based on the empirical observation of the model predictions with respect to noisy samples.

**Observation.** Our hypothesis is that noisy samples have distinguished properties from benign samples due to the distributional shifts, and this could

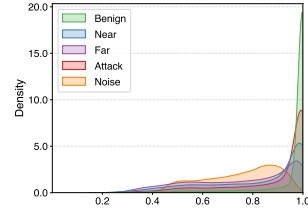 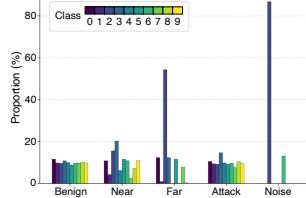

(a) Confidence distribution.     (b) Predicted class distribution.

Figure 5: Model predictions on benign (CIFAR10-C) and noisy samples.

be observable via the models' prediction outputs. We investigate two types of features that work as proxies for identifying benign samples: (i) confidence of the samples and (ii) predicted class distributions. Specifically, we compared the distribution of the softmax confidence (Figure 5a) and the predicted class distribution (Figure 5b) of benign samples with noisy samples. First, the confidence of the samples is relatively lower than that of benign samples. The more severe the distribution shift, the lower the confidence (e.g., Far is less confidence than Near), which is also in line with findings of previous studies that pre-trained models show higher confidence on target distribution than out-of-distribution data [10, 21]. Second, we found that noisy samples are often skewed in terms of predictions, and this phenomenon is prominent in more severe shifts (e.g., Noise), except for Attack, whose objective is to make the model fail to correctly classify. These skewed distributions could lead to an undesirable bias in $p(y)$ and thus might negatively impact the TTA objective, such as entropy minimization [38].

---

**Algorithm 1** High-confidence Uniform-class Sampling (HUS)

---

**Input:** test data stream $\mathbf{x}_t$, memory $M$ with capacity $N$
   **for** test time $t \in \{1, \cdots, T\}$ **do**
      $\hat{y}_t \leftarrow f(\mathbf{x}; \Theta)$
      **if** $C(\mathbf{x}; \Theta) > C_0$ **then**                          ▷ Sampling confident data
         **if** $|M| < N$ **then**
            Add $(\mathbf{x}_t, \hat{y}_t)$ to $M$
         **else**
            $\mathcal{Y}^* \leftarrow$ the most prevalent class(es) in $M$
            **if** $\hat{y}_t \notin \mathcal{Y}^*$ **then**                  ▷ Balancing classes
               Randomly discard $(\mathbf{x}_i, \hat{y}_i)$ from $M$ where $\hat{y}_i \in \mathcal{Y}^*$
            **else**
               Randomly discard $(\mathbf{x}_i, \hat{y}_i)$ from $M$ where $\hat{y}_i = \hat{y}_t$
            Add $(\mathbf{x}_t, \hat{y}_t)$ to $M$

---

**Solution.** Based on the aforementioned empirical analysis, we propose High-confidence Uniform-class Sampling (HUS) that avoids using noisy samples for adaptation by utilizing a small memory. We maintain confident samples while balancing their predicted classes in the memory. The selected samples in the memory are then used for adaptation. We describe the procedure as a pseudo-code code in Algorithm 1. Given a target test sample $\mathbf{x}$, HUS measures its confidence. Specifically, we define the confidence $C(\mathbf{x}; \Theta)$ of each test sample $\mathbf{x}$ as:

$$C(\mathbf{x}; \Theta) = \max_{i=1\cdots n}\left(\frac{e^{\hat{y}_i}}{\sum_{j=1}^{n} e^{\hat{y}_j}}\right) \text{ where } \hat{y} = f(\mathbf{x}; \Theta). \tag{1}$$

We store the sample if its confidence is higher than the predefined threshold $C_0$. In this way, we maintain only high-confidence samples in the memory used for adaptation and thus reduce the impact of low-confidence noisy samples.

Furthermore, while storing data in the memory, we balance classes among them. Specifically, if the predicted class of the current test sample is not in the most prevalent class in the memory, then HUS randomly replaces one random sample in the most prevalent class with the new sample. Otherwise, if the current sample belongs to the most prevalent class in the memory, HUS replaces one random sample in the same class with the current one. We can maintain classes uniformly with this strategy, which is effective for not only filtering out noisy samples but also removing class biases among samples when used for adaptation, which we found is beneficial for TTA. We found these two memory management strategies not only effectively reduce the impact of noisy samples for adaptation but also improve the model performance in benign-only cases by avoiding model drifting due to biased and low-confidence samples (Section 4).

With the stored samples in the memory, we update the normalization statistics and affine parameters in batch normalization (BN) [13] layers, following the prior TTA methods [29, 38, 44]. This is known as not only computationally efficient but also showed comparable performance improvement to updating the whole layers. While we aim to avoid using noisy samples for adaptation, a few noisy samples could still be stored in the memory, e.g., when they are similar to benign samples or outliers. To be robust to temporal variances of the samples in the memory, we take the exponential moving average to update the BN statistics (means $\mu$ and variances $\sigma^2$) instead of directly using the statistics from the samples in the memory. Specifically, we update the means and variances of BN layers as: (i) $\hat{\mu}_t = (1 - m)\hat{\mu}_{t-1} + m\mu_t$ and (ii) $\hat{\sigma}_t^2 = (1 - m)\hat{\sigma}_{t-1}^2 + m\sigma_t^2$, where $m \in [0, 1]$ is a momentum hyperparameter. We describe updating the affine parameters in Section 3.2.

## 3.2 Parameter-wise robustness via entropy-sharpness minimization

Our second approach is to secure parameter-wise robustness to noisy samples by training the model in a way that is robust to noisy samples. Our idea is based on the observation that the parameter update is often corrupted with noisy samples, and this could be mitigated by smoothing the loss landscape with respect to parameters.

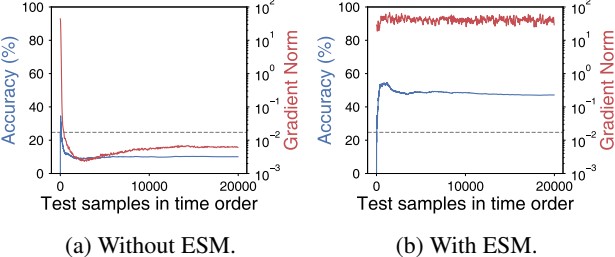

(a) Without ESM.   (b) With ESM.

Figure 6: Cumulative accuracy (%) of benign samples (CIFAR10-C) and gradient norm of noisy samples (Noise) as the adaptation proceeds. The dotted line refers to the source model accuracy.

**Observation.** While most existing TTA algorithms utilize the entropy minimization loss [5, 38, 44], it could drift the model with high gradient samples [29]. We observed that adaptation with noisy samples often hinders the model from adapting to benign samples. Figure 6 shows an example of this phenomenon. Specifically, test samples consist of 10k benign samples and 10k noisy samples (Noise), which are randomly shuffled. We computed the cumulative accuracy for the benign test data and the gradient norm of noisy samples at each step. As shown in Figure 6a, the gradient norm of noisy samples dropped rapidly, indicating that the model is gradually adapting to these noise data. This leads to a significant accuracy drop for benign samples. The key question here is how to prevent the model from overfitting to noisy samples. Figure 6b shows the result with entropy-sharpness minimization (ESM) that we explain in the following paragraph. With ESM, the gradient norm of noise data remains high, and the accuracy for benign samples improved after adaptation as intended.

**Solution.** To make the model parameters robust to adaptation with noisy samples, the entropy loss landscape should be smoother so that the model becomes resilient to unexpected model drift due to noisy samples. To that end, we jointly minimize the naive entropy loss and the sharpness of the entropy loss and thus make the loss landscape robust to model weight perturbations by large gradients from noisy samples. Specifically, we replace the naive entropy minimization $E$ with the

entropy-sharpness minimization (ESM) as:

$$\min_\Theta E_S(\mathbf{x}, \Theta) = \min_\Theta \max_{||\epsilon||_2 \leq \rho} E(\mathbf{x}; \Theta + \epsilon), \tag{2}$$

where the entropy-sharpness $E_S(\mathbf{x}, \Theta)$ is defined as the maximum objective around the weight perturbation with L2-norm constraint $\rho$. To tackle this joint optimization problem, we follow sharpness-aware minimization [4] similar to [29], which originally aims to improve the generalizability of models over standard optimization algorithms such as stochastic gradient descent (SGD). We repurpose this optimization to make the model robust to noisy samples in TTA scenarios.

Specifically, assuming $\rho \ll 1$, the optimization can be approximated via a first-order Taylor expansion:

$$\epsilon^*(\Theta) \triangleq \arg\max_{||\epsilon||_2 \leq \rho} E(\mathbf{x}; \Theta + \epsilon) \approx \arg\max_{||\epsilon||_2 \leq \rho} \epsilon^T \nabla_\Theta E(\mathbf{x}; \Theta). \tag{3}$$

The solution for this approximation problem is given by the classical dual norm problem:

$$\hat{\epsilon}(\Theta) = \rho \, \text{sign}(\nabla_\Theta E(\mathbf{x}; \Theta)) \, |\nabla_\Theta E(\mathbf{x}; \Theta)|^{q-1} / \left( ||\nabla_\Theta E(\mathbf{x}; \Theta)||_q^q \right)^{1/p}, \tag{4}$$

where $1/p + 1/q = 1$. We set $p = 2$ for further implementation, following the suggestion [4]. By substituting $\hat{\epsilon}(\Theta)$ to the original entropy-sharpness minimization problem, the final gradient approximation is:

$$\nabla_\Theta E_S(\mathbf{x}, \Theta) \approx \nabla_\Theta E(\mathbf{x}, \Theta)|_{\Theta + \hat{\epsilon}(\Theta)}. \tag{5}$$

In summary, we calculate the entropy-sharpness minimization objective via two steps. First, at time step $t$, it calculates the $\hat{\epsilon}(\Theta_t)$ with previous parameters and entropy loss. It generates the temporal model with the new parameters: $\Theta_t + \hat{\epsilon}(\Theta_t)$. Second, based on the temporary model, the second step updates the original model's parameters with the approximation in Equation 5. By putting it together with HUS (Section 3.1), parameters are updated by:

$$\Theta_t = \Theta_{t-t_0} - \eta \nabla_\Theta E(\mathbf{x}, \Theta)|_{\mathbf{x}=\text{HUS}_{C_0}(t), \ \Theta = \Theta_{t-t_0} + \hat{\epsilon}(\Theta_{t-t_0})}, \tag{6}$$

where $\eta$ is the step size and $t_0$ is the model adaptation interval. For simplicity, we set the model adaptation interval the same as the memory capacity, i.e., $t_0 = N$. In the experiments, we use $N = 64$, which is one of the most widely-used batch sizes in TTA methods [36, 38, 44].

## 4 Experiments

This section describes our experimental setup and demonstrates the results in various settings. Please refer to Appendix A and B for further details.

**Scenario.** To mimic the presence of noisy samples in addition to the original target test samples, we injected various noisy datasets into target datasets and randomly shuffled them, which we detail in the following paragraphs. We report the classification accuracy of the original target samples to measure the performance of the model in the presence of noisy samples. We ran all experiments with three random seeds (0, 1, 2) and reported the average accuracy and standard deviations in Appendix B.

**Target datasets.** We used three standard TTA benchmarks: **CIFAR10-C**, **CIFAR100-C**, and **ImageNet-C** [9] as our target datasets. All datasets contain 15 different types and five levels of corruption, where we use the most severe corruption level of five. For each corruption type, the CIFAR10-C/CIFAR100-C dataset has 10,000 test data with 10/100 classes, and the ImageNet-C dataset has 50,000 test data with 1000 classes. We use ResNet18 [8] as the backbone network. We pre-trained the model for CIFAR10 with training data and used the TorchVision [23] pre-trained model for ImageNet.

**Noisy datasets.** Besides the target datasets (**Benign**) mentioned above, we consider four noisy scenarios (Figure 3): CIFAR100 [15]/ImageNet [3] (**Near**), MNIST [26] (**Far**), adversarial attack (**Attack**), and uniform random noise (**Noise**). As both CIFAR10-C and ImageNet-C have real-world images for object recognition tasks, CIFAR100 would be a near dataset to them. For CIFAR100-C, we select ImageNet [3] for a near dataset. We select MNIST as a far dataset because they are not real-world images. We referred to the OOD benchmark [41] for choosing the term Near and Far and the datasets. For the adversarial attack, we adopt the Distribution Invading Attack (DIA), which is an

Table 1: Classification accuracy (%) on CIFAR10-C for 15 types of corruptions under five scenarios: Benign, Near, Far, Attack, and Noise. Benign contains only benign target samples, while other scenarios include both benign and each type of noisy samples specified. **Bold** numbers are the highest accuracy. Averaged over three different random seeds.

| | | Noise | | | Blur | | | | Weather | | | | Digital | | | | Avg. |
|---|---|---|---|---|---|---|---|---|---|---|---|---|---|---|---|---|---|
| | Method | Gau. | Shot | Imp. | Def. | Gla. | Mot. | Zoom | Snow | Fro. | Fog | Brit. | Cont. | Elas. | Pix. | JPEG | |
| Benign | Source | 26.0 | 33.2 | 24.7 | 56.7 | 52.0 | 67.4 | 64.8 | 78.0 | 67.0 | 74.1 | 91.5 | 33.9 | 76.6 | 46.4 | 73.2 | 57.7 |
| | BN Stats [27] | 67.0 | 69.0 | 60.4 | 87.8 | 65.6 | 86.3 | 87.4 | 81.6 | 80.3 | 85.4 | 90.7 | 86.9 | 76.7 | 79.3 | 71.9 | 78.4 |
| | PL [17] | 71.1 | 72.9 | 62.2 | 86.9 | 64.4 | 85.3 | 86.6 | 80.8 | 78.8 | 84.9 | 89.6 | 84.0 | 76.2 | 80.0 | 73.1 | 78.5 |
| | TENT [38] | 74.5 | 77.6 | 66.6 | 88.2 | 66.2 | 86.9 | 88.8 | 83.7 | 81.3 | 86.0 | 91.1 | 86.9 | 77.9 | 82.7 | 76.7 | 81.0 |
| | LAME [1] | 21.8 | 29.2 | 19.7 | 53.3 | 52.1 | 65.9 | 62.5 | 79.2 | 69.3 | 73.1 | 90.1 | 28.0 | 75.7 | 43.8 | 74.1 | 55.9 |
| | CoTTA [39] | **76.9** | **78.6** | **72.3** | 88.2 | **70.9** | 86.8 | 88.1 | 83.4 | 83.4 | 86.1 | 91.2 | 84.9 | 79.2 | 83.0 | **79.9** | 82.2 |
| | EATA [28] | 76.0 | 78.2 | 68.2 | 88.4 | 70.1 | 87.4 | 88.4 | 84.5 | **85.0** | **88.0** | 91.5 | **89.9** | 77.8 | **84.8** | 78.4 | **82.4** |
| | SAR [29] | 68.3 | 69.7 | 58.9 | 87.8 | 62.9 | 86.3 | 87.4 | 81.6 | 80.3 | 85.4 | 90.7 | 86.9 | 76.7 | 79.3 | 72.0 | 78.3 |
| | RoTTA [44] | 65.2 | 67.4 | 58.3 | 87.2 | 64.4 | 85.8 | 87.3 | 81.2 | 76.9 | 85.3 | 90.7 | 57.2 | 76.5 | 77.7 | 71.6 | 75.5 |
| | SoTTA | 75.0 | 77.5 | 68.8 | **88.8** | 70.7 | **87.5** | **89.0** | **85.4** | 84.0 | 88.2 | **91.9** | 83.9 | **79.8** | 83.9 | 78.3 | 82.2 |
| Near | Source | 26.0 | 33.2 | 24.7 | 56.7 | 52.0 | 67.4 | 64.8 | 78.0 | 67.0 | 74.1 | 91.5 | 33.9 | 76.6 | 46.4 | 73.2 | 57.7 |
| | BN Stats [27] | 64.9 | 66.3 | 58.2 | 84.1 | 62.7 | 84.2 | 85.0 | 83.1 | 82.5 | 85.5 | 92.4 | 85.3 | 76.5 | 67.4 | 70.3 | 76.6 |
| | PL [17] | 63.2 | 63.5 | 51.7 | 81.6 | 58.4 | 78.4 | 83.3 | 79.2 | 79.7 | 80.3 | 89.2 | 79.5 | 73.1 | 70.9 | 69.3 | 73.4 |
| | TENT [38] | 64.7 | 64.7 | 50.2 | 81.3 | 59.6 | 80.8 | 83.8 | 79.6 | 78.3 | 80.0 | 88.6 | 83.7 | 73.5 | 74.3 | 71.1 | 74.3 |
| | LAME [1] | 24.3 | 31.6 | 19.9 | 53.9 | 53.2 | 65.9 | 62.5 | 79.0 | 69.5 | 73.1 | 90.1 | 28.4 | 75.0 | 44.8 | 74.2 | 56.4 |
| | CoTTA [39] | 72.7 | 74.3 | 66.0 | 82.6 | **67.6** | 81.8 | 84.1 | 84.1 | **85.5** | 82.5 | 91.1 | 69.9 | 78.1 | 76.1 | **79.3** | 78.4 |
| | EATA [28] | 51.5 | 50.3 | 40.1 | 70.4 | 45.8 | 72.8 | 77.2 | 66.8 | 67.4 | 74.4 | 83.9 | 68.2 | 60.2 | 67.3 | 62.1 | 63.9 |
| | SAR [29] | 59.0 | 60.9 | 52.8 | 78.2 | 55.4 | 83.7 | 81.8 | 78.8 | 78.6 | 85.5 | 92.4 | 85.3 | 66.6 | 64.0 | 63.8 | 72.4 |
| | RoTTA [44] | 66.3 | 68.3 | 59.4 | 86.0 | 63.2 | 85.8 | 86.4 | 82.4 | 83.8 | 86.4 | **92.4** | 84.8 | 77.1 | 71.6 | 71.6 | 77.7 |
| | SoTTA | **74.3** | **76.7** | **66.5** | **87.5** | 66.9 | **86.4** | **87.8** | **84.4** | 83.8 | **87.2** | 91.3 | **88.4** | **78.7** | **82.4** | 78.0 | **81.4** |
| Far | Source | 26.0 | 33.2 | 24.7 | 56.7 | 52.0 | 67.4 | 64.8 | 78.0 | 67.0 | 74.1 | 91.5 | 33.9 | 76.6 | 46.4 | 73.2 | 57.7 |
| | BN Stats [27] | 62.7 | 66.1 | 56.0 | 86.5 | 60.7 | 84.2 | 87.2 | 79.8 | 78.4 | 85.7 | 89.9 | 80.2 | 75.5 | 69.8 | 65.0 | 75.2 |
| | PL [17] | 55.2 | 54.1 | 48.3 | 83.2 | 49.3 | 80.0 | 83.0 | 76.8 | 73.3 | 80.6 | 87.6 | 74.6 | 70.5 | 66.8 | 63.6 | 69.8 |
| | TENT [38] | 51.6 | 57.4 | 43.7 | 84.8 | 43.5 | 83.3 | 85.3 | 80.4 | 73.1 | 83.5 | 88.9 | 80.8 | 73.5 | 72.2 | 66.7 | 71.2 |
| | LAME [1] | 22.8 | 29.6 | 19.3 | 53.2 | 50.4 | 64.6 | 60.7 | 79.1 | 67.9 | 72.8 | 90.1 | 28.1 | 74.7 | 44.4 | 74.3 | 55.5 |
| | CoTTA [39] | 67.4 | 71.1 | 59.4 | 83.3 | 61.2 | 82.3 | 84.3 | 80.4 | 80.4 | 83.8 | 87.2 | 58.0 | 76.0 | 70.3 | 72.9 | 74.5 |
| | EATA [28] | 40.0 | 46.4 | 34.9 | 73.5 | 35.2 | 59.3 | 76.4 | 61.1 | 59.5 | 68.4 | 85.2 | 55.2 | 46.5 | 53.5 | 49.0 | 56.3 |
| | SAR [29] | 60.3 | 62.6 | 50.9 | 86.5 | 55.4 | 84.3 | 87.2 | 79.8 | 78.3 | 85.7 | 89.9 | 80.2 | 70.1 | 67.8 | 60.9 | 73.3 |
| | RoTTA [44] | 67.3 | 69.8 | 61.1 | 88.1 | 66.0 | 86.6 | 88.0 | 82.0 | 78.8 | 86.2 | 91.2 | 61.7 | 77.5 | 79.6 | 73.0 | 77.1 |
| | SoTTA | **73.3** | **76.3** | **66.3** | **88.5** | **68.3** | **86.8** | **88.3** | **84.1** | **84.2** | **87.2** | **92.0** | **89.0** | **77.8** | **83.8** | **77.8** | **81.6** |
| Attack | Source | 26.0 | 33.2 | 24.7 | 56.7 | 52.0 | 67.4 | 64.8 | 78.0 | 67.0 | 74.1 | 91.5 | 33.9 | 76.6 | 46.4 | 73.2 | 57.7 |
| | BN Stats [27] | 44.5 | 46.8 | 39.8 | 63.3 | 42.1 | 63.5 | 62.5 | 62.3 | 59.3 | 62.1 | 75.4 | 65.1 | 50.6 | 53.9 | 46.9 | 55.9 |
| | PL [17] | 59.1 | 61.4 | 53.1 | 73.1 | 51.9 | 71.7 | 72.3 | 71.3 | 69.1 | 69.8 | 81.5 | 72.1 | 60.9 | 67.4 | 60.1 | 66.3 |
| | TENT [38] | 62.9 | 65.2 | 56.7 | 74.8 | 54.5 | 73.4 | 75.2 | 73.8 | 71.8 | 72.1 | 83.0 | 73.7 | 63.1 | 70.0 | 64.0 | 68.9 |
| | LAME [1] | 21.0 | 28.0 | 17.3 | 52.7 | 52.9 | 65.9 | 62.0 | 79.3 | 70.4 | 73.9 | 90.3 | 28.4 | 75.6 | 44.2 | 74.2 | 55.9 |
| | CoTTA [39] | 53.1 | 57.0 | 49.7 | 67.9 | 55.9 | 69.7 | 71.3 | 74.3 | 69.5 | 68.1 | 84.9 | 47.0 | 68.6 | 62.3 | 71.6 | 69.5 |
| | EATA [28] | 66.6 | 68.7 | 57.9 | 75.2 | 57.2 | 74.7 | 75.5 | 75.1 | 73.5 | 73.8 | 83.5 | 76.0 | 64.3 | 73.5 | 67.8 | 70.9 |
| | SAR [29] | 46.1 | 48.1 | 40.3 | 63.3 | 42.2 | 63.5 | 62.5 | 62.3 | 59.3 | 62.1 | 75.4 | 65.1 | 50.6 | 53.9 | 47.6 | 56.2 |
| | RoTTA [44] | 69.7 | 71.8 | 62.9 | 88.6 | 67.8 | 87.4 | 88.7 | 83.2 | 80.1 | 87.1 | 91.8 | 63.5 | 78.4 | 80.6 | 74.1 | 78.4 |
| | SoTTA | **78.2** | **80.8** | **72.3** | **90.1** | **73.6** | **89.2** | **90.3** | **87.4** | **86.2** | **89.3** | **92.9** | **87.8** | **81.3** | **86.6** | **81.0** | **84.5** |
| Noise | Source | 26.0 | 33.2 | 24.7 | 56.7 | 52.0 | 67.4 | 64.8 | 78.0 | 67.0 | 74.1 | **91.5** | 33.9 | 76.6 | 46.4 | 73.2 | 57.7 |
| | BN Stats [27] | 51.7 | 53.9 | 45.5 | 52.7 | 41.5 | 51.0 | 55.1 | 62.8 | 63.8 | 53.8 | 76.9 | 55.8 | 46.8 | 54.8 | 56.4 | 54.8 |
| | PL [17] | 47.6 | 52.7 | 44.7 | 48.9 | 36.1 | 49.4 | 54.1 | 61.9 | 56.5 | 50.9 | 77.1 | 45.2 | 43.1 | 49.4 | 59.5 | 51.8 |
| | TENT [38] | 54.0 | 57.1 | 36.7 | 48.9 | 28.3 | 50.5 | 51.0 | 64.0 | 64.7 | 49.5 | 80.5 | 43.7 | 38.4 | 56.7 | 57.0 | 52.1 |
| | LAME [1] | 21.8 | 28.6 | 18.5 | 51.6 | 50.8 | 64.3 | 60.9 | 78.4 | 67.3 | 71.7 | 90.5 | 27.0 | 75.1 | 43.0 | 73.4 | 54.9 |
| | CoTTA [39] | 60.4 | 60.3 | 52.4 | 47.3 | 41.6 | 44.1 | 52.0 | 62.7 | 66.6 | 47.7 | 79.0 | 44.7 | 42.8 | 60.2 | 60.2 | 54.8 |
| | EATA [28] | 42.2 | 41.0 | 33.2 | 32.7 | 25.0 | 27.9 | 34.3 | 40.8 | 42.6 | 31.6 | 61.5 | 20.3 | 27.5 | 35.8 | 43.1 | 36.0 |
| | SAR [29] | 57.5 | 59.3 | 49.6 | 57.2 | 43.7 | 54.4 | 59.4 | 64.8 | 65.4 | 57.9 | 77.1 | 60.2 | 50.0 | 58.3 | 59.8 | 58.3 |
| | RoTTA [44] | 64.4 | 66.9 | 56.1 | 80.1 | 59.1 | 79.8 | 82.2 | 79.7 | 78.7 | 77.8 | 91.2 | 69.0 | 72.3 | 73.4 | 72.8 | 73.6 |
| | SoTTA | **73.3** | **77.7** | **66.8** | **86.1** | **64.0** | **84.3** | **86.6** | **83.1** | **82.0** | **85.7** | 91.1 | **84.1** | **77.1** | **81.6** | **77.2** | **80.0** |

adversarial attack algorithm designed for TTA [40]. We inject the small perturbations to malicious samples to increase the overall error rate on benign data in the same batch. For the uniform random noise, we generate pixel-wise uniform-random images with the same size as the target images. We set the number of noisy samples equal to each target dataset as default, and we also investigated the impact of the number of noisy samples in the following ablation study (Figure 7). In cases where the number of noisy samples is different from the target datasets, we randomly resampled them. To ensure that the learning algorithm does not know the information of noisy samples beforehand, the pixel values of all noisy images are normalized with respect to the target dataset.

**Baselines.** We consider various state-of-the-art TTA methods as baselines. **Source** evaluates the pre-trained model directly on the target data without adaptation. Test-time batch normalization (**BN stats**) [27] updates the BN statistics from the test batch. Pseudo-Label (**PL**) [17] optimizes the trainable BN parameters via pseudo labels. Test entropy minimization (**TENT**) [38] updates the BN parameters via entropy minimization.

We also consider the latest TTA algorithms that improve robustness to temporal distribution changes in test streams to understand their performance to noisy samples. Laplacian adjusted maximum-likelihood estimation (**LAME**) [1] modifies the classifier output probability without modifying model internal parameters. Continual test-time adaptation (**CoTTA**) [39] uses weight-averaged and

Table 2: Classification accuracy (%) on CIFAR100-C. **Bold** numbers are the highest accuracy. Averaged over three different random seeds for 15 types of corruption.

| Method | Benign | Near | Far | Attack | Noise | Avg. |
|---|---|---|---|---|---|---|
| Source | 33.2 ± 0.4 | 33.2 ± 0.4 | 33.2 ± 0.4 | 33.2 ± 0.4 | 33.2 ± 0.4 | 33.2 ± 0.4 |
| BN Stats [27] | 53.7 ± 0.2 | 50.8 ± 0.1 | 46.8 ± 0.1 | 29.2 ± 0.4 | 28.3 ± 0.3 | 41.8 ± 0.1 |
| PL [17] | 56.6 ± 0.2 | 48.0 ± 0.3 | 42.8 ± 0.7 | 39.0 ± 0.4 | 23.8 ± 0.6 | 42.1 ± 0.3 |
| TENT [38] | 59.5 ± 0.0 | 46.4 ± 1.4 | 40.0 ± 1.3 | 31.9 ± 0.7 | 20.0 ± 0.9 | 39.5 ± 0.7 |
| LAME [1] | 31.0 ± 0.5 | 31.5 ± 0.5 | 30.8 ± 0.7 | 31.0 ± 0.6 | 31.1 ± 0.7 | 31.1 ± 0.6 |
| CoTTA [39] | 55.8 ± 0.4 | 50.0 ± 0.3 | 42.4 ± 0.4 | 37.2 ± 0.2 | 27.3 ± 0.3 | 42.6 ± 0.2 |
| EATA [28] | 23.5 ± 1.9 | 6.1 ± 0.3 | 4.8 ± 0.5 | 3.7 ± 0.6 | 2.4 ± 0.2 | 8.1 ± 0.3 |
| SAR [29] | 57.3 ± 0.3 | 55.4 ± 0.1 | 51.2 ± 0.1 | 34.4 ± 0.3 | 38.1 ± 1.2 | 47.3 ± 0.3 |
| RoTTA [44] | 48.7 ± 0.6 | 49.4 ± 0.5 | 49.8 ± 0.9 | 51.5 ± 0.4 | 48.3 ± 0.5 | 49.6 ± 0.6 |
| SoTTA | **60.5 ± 0.0** | **57.1 ± 0.2** | **59.0 ± 0.4** | **61.9 ± 0.0** | **58.6 ± 1.0** | **59.4 ± 0.3** |

Table 3: Classification accuracy (%) on ImageNet-C. **Bold** numbers are the highest accuracy. Averaged over three different random seeds for 15 types of corruption.

| Method | Benign | Near | Far | Attack | Noise | Avg. |
|---|---|---|---|---|---|---|
| Source | 14.6 ± 0.0 | 14.6 ± 0.0 | 14.6 ± 0.0 | 14.6 ± 0.0 | 14.6 ± 0.0 | 14.6 ± 0.0 |
| BN Stats [27] | 27.1 ± 0.0 | 18.9 ± 0.1 | 14.8 ± 0.0 | 17.4 ± 0.8 | 12.8 ± 0.0 | 18.2 ± 0.1 |
| PL [17] | 30.5 ± 0.1 | 6.9 ± 0.0 | 5.1 ± 0.2 | 18.1 ± 1.3 | 3.4 ± 0.6 | 12.8 ± 0.2 |
| TENT [38] | 27.1 ± 0.0 | 18.9 ± 0.1 | 14.8 ± 0.0 | 17.4 ± 0.8 | 12.8 ± 0.0 | 18.2 ± 0.1 |
| LAME [1] | 14.4 ± 0.0 | 14.4 ± 0.1 | 14.4 ± 0.0 | 14.0 ± 0.6 | 14.3 ± 0.0 | 14.3 ± 0.1 |
| CoTTA [39] | 32.2 ± 0.1 | 23.3 ± 0.2 | 17.6 ± 0.2 | 28.3 ± 1.3 | 16.0 ± 0.9 | 23.4 ± 0.2 |
| EATA [28] | 38.0 ± 0.1 | 25.6 ± 0.4 | 23.1 ± 0.1 | 26.1 ± 0.1 | 20.7 ± 0.2 | 26.7 ± 0.0 |
| SAR [29] | 36.1 ± 0.1 | 27.6 ± 0.3 | 23.5 ± 0.4 | 26.8 ± 1.0 | 22.0 ± 0.4 | 27.2 ± 0.2 |
| RoTTA [44] | 29.7 ± 0.0 | 25.6 ± 0.4 | 29.2 ± 0.2 | 32.0 ± 1.2 | 31.2 ± 0.2 | 29.5 ± 0.3 |
| SoTTA | **39.8 ± 0.0** | **27.9 ± 0.3** | **36.1 ± 0.1** | **41.1 ± 0.1** | **39.0 ± 0.1** | **36.8 ± 0.0** |

augmentation-averaged predictions and avoids catastrophic forgetting by stochastically restoring a part of the model. Efficient anti-forgetting test-time adaptation (**EATA**) [28] uses entropy and diversity weight with Fisher regularization to prevent forgetting. Sharpness-aware and reliable optimization (**SAR**) [29] removes high-entropy samples and optimizes entropy with sharpness minimization [4]. Robust test-time adaptation (**RoTTA**) [44] utilizes the teacher-student model to stabilize while selecting the data with category-balanced sampling with timeliness and uncertainty.

**Hyperparameters.** We adopt the hyperparameters of the baselines from the original paper or official codes. We use the test batch size of 64 in all methods for a fair comparison. Accordingly, we set the memory size to 64 and adapted the model for every 64 samples for our method and RoTTA [44]. We conduct TTA in an online manner. We used a fixed hyperparameter of BN momentum $m = 0.2$ and updated the BN affine parameters via the Adam optimizer [14] with a fixed learning rate of $l = 0.001$ and a single adaptation epoch. The confidence threshold $C_0$ is set to 0.99 for CIFAR10-C, 0.66 for CIFAR100-C, and 0.33 for ImageNet-C. We set the sharpness threshold $\rho = 0.05$ as previous works [4, 29]. We specify further details of the hyperparameters in Appendix A.

**Overall result.** Table 1 shows the result on CIFAR10-C for 15 types of corruptions under five noisy scenarios described in Figure 3. We observed significant performance degradation in most of the baselines under the noisy settings. In addition, the extent of accuracy degradation from the Benign case is more prominent in more severe shift cases (e.g., the degree of degradation is generally Near < Far < Attack < Noise). Popular TTA baselines (BN stats, PL, and TENT) might fail due to updating with noisy samples. State-of-the-art TTA baselines that address temporal distribution shifts in TTA (LAME, CoTTA, EATA, SAR, and RoTTA) still suffered from noisy scenarios, as they are not designed to deal with unexpected samples. On the other hand, SoTTA showed its robustness across different scenarios. This validates the effectiveness of our approaches to ensure both input-wise and parameter-wise robustness. Interestingly, we found that SoTTA showed comparable performance to state-of-the-art baselines in the Benign case as well. Our interpretation of this result is two-fold. First, our high-confidence uniform-class sampling strategy filters not only noisy samples but also benign samples that would negatively impact the algorithm's objective. This implies that there exist samples that are more beneficial for adaptation, which aligns with the findings that high-entropy samples

Table 4: Classification accuracy (%) and corresponding standard deviation of varying ablative settings in SoTTA on CIFAR10-C. **Bold** numbers are the highest accuracy. Averaged over three different random seeds for 15 types of corruption.

| Method | Benign | Near | Far | Attack | Noise | Avg. |
|---|---|---|---|---|---|---|
| Source | 57.7 ± 1.0 | 57.7 ± 1.0 | 57.7 ± 1.0 | 57.7 ± 1.0 | 57.7 ± 1.0 | 57.7 ± 1.0 |
| HC | 34.9 ± 4.8 | 13.6 ± 0.3 | 17.6 ± 3.8 | 16.9 ± 1.6 | 16.8 ± 0.2 | 20.0 ± 2.0 |
| UC | 66.4 ± 3.0 | 62.1 ± 0.8 | 56.5 ± 2.0 | 70.0 ± 3.9 | 59.5 ± 3.0 | 62.9 ± 0.7 |
| HC + UC (HUS) | 69.8 ± 1.1 | 61.7 ± 1.3 | 58.4 ± 0.5 | 40.9 ± 5.5 | 58.9 ± 2.6 | 57.9 ± 0.8 |
| ESM | **82.6** ± 0.2 | 77.9 ± 0.4 | 72.8 ± 0.7 | 83.4 ± 0.2 | 60.5 ± 1.8 | 75.4 ± 0.5 |
| HC + ESM | 82.3 ± 0.2 | 80.9 ± 0.6 | 74.9 ± 2.4 | 83.5 ± 0.2 | 68.7 ± 7.0 | 78.0 ± 2.0 |
| UC + ESM | 82.2 ± 0.2 | 78.0 ± 0.4 | 75.9 ± 0.5 | 84.3 ± 0.1 | 77.7 ± 0.7 | 79.6 ± 0.2 |
| HUS + ESM (SoTTA) | 82.2 ± 0.3 | **81.4** ± 0.5 | **81.6** ± 0.6 | **84.5** ± 0.2 | **80.0** ± 1.4 | **81.9** ± 0.5 |

harm adaptation performance [29]. Second, entropy-sharpness minimization helps ensure both the robustness to noisy samples and the generalizability of the model by preventing model drifts from large gradients, leading to performance improvement with benign samples. We found similar patterns for the CIFAR100-C (Table 2) and ImageNet-C (Table 3). More details are in Appendix B.

**Impact of individual components of SoTTA.** We conducted an ablative study to further investigate the effectiveness of SoTTA's individual components. Table 4 shows the result of the ablation study for CIFAR10-C. For input-wise robustness, **HC** refers to high-confidence sampling, and **UC** refers to uniform-class sampling. The two strategies are integrated into our high-confidence uniform-class sampling (**HUS**). **ESM** is our entropy-sharpness minimization for parameter-wise robustness. Note that we utilized FIFO memory with the same size for the UC case without HC and the native entropy minimization where we did not utilize ESM. Overall, the accuracy is improved as we sequentially added each approach of SoTTA. This validates our claim that ensuring both input-wise and parameter-wise robustness via HUS and ESM is a synergetic strategy to combat noisy samples in TTA.

**Impact of the number of noisy samples.** We also investigate the effect of the size of noisy samples on TTA algorithms. Specifically, for the CIFAR10-C dataset with the Noise case, we varied the noise samples from 5,000 (5k) to 20,000 (20k) while fixing the size of benign samples as 10,000. Figure 7 shows the result. We observe that the accuracy of most baselines tends to deteriorate with a larger number of noisy samples and is sometimes even worse than without adaptation (Source). SoTTA showed its resilience to the size of noisy samples with 1.9%p degradation from 5k to 20k samples. RoTTA also showed its robustness to noisy samples to some extent, but the performance gain is around 6.4%p lower than SoTTA.

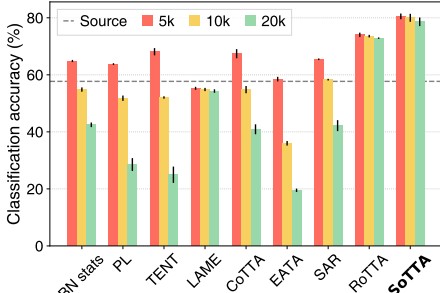

Figure 7: Classification accuracy (%) varying the size of noisy samples on CIFAR10-C under Noise.

## 5 Related work

**Test-time adaptation.** While test-time adaptation (TTA) attempts to optimize model parameters with unlabeled test data streams, it lacks consideration of spoiling of the sample itself; i.e., it inevitably adapts to potential outlier samples mixed in the stream, such as adversarial data or mere noise. Most existing TTA algorithms directly optimize the model with incoming sample data. Test-time normalization [27, 36] updates the batch norm statistics in test-time to minimize the expected loss. TENT [38] minimizes the entropy of the model's predictions on a batch of test data.On the one hand, recent studies promote the robustness of the model, yet the consideration is limited to temporal distribution shifts of test data [1, 4, 5, 28, 29, 39, 44]. For instance, CoTTA [39] tries to adapt the model in a continually changing target environment via self-training and stochastical restoring. NOTE [5] proposes instance-aware batch normalization combined with prediction-balanced reservoir sampling to ensure model robustness towards temporally correlated data stream, which requires retraining with source data. We provide a method-wise comparison with EATA [28], SAR [29], and RoTTA [44] in Appendix D.2. To conclude, while existing TTA methods seek the robustness of the

model to temporal distribution shifts, they do not consider scenarios where noisy samples appear in test streams.

**Out-of-distribution detection.** Out-of-distribution (OOD) detection [10, 11, 18, 19, 20, 21, 24, 25, 43] aims to ensure the robustness of the model by identifying when a given data falls outside the training distribution. A representative method is a thresholding approach [10, 19, 20, 21], that defines a scoring function given an input and pretrained classifier. The sample is detected as OOD data if the output of the scoring function is higher than a threshold. Importantly, OOD detection studies are built on the condition that training and test domains are the same, which differs from TTA's scenario. Furthermore, OOD detection assumes that a model is fixed during test time, while a model changes continually in TTA. These collectively make it difficult to apply OOD detection studies directly to TTA scenarios.

**Open-set domain adaptation.** Open-set Domain Adaptation (OSDA) assumes that a target domain contains unknown classes not discovered in a source domain [30, 35] in domain adaptation scenarios. These methods aim to learn a mapping function between the source and target domains. While the target scenario that a model could encounter unknown classes of data in the test time is similar to our objective, these methods do not fit into TTA as it assumes both labeled source and unlabeled target data are available in the training time. Universal domain adaptation (UDA) [34, 42] further generalizes the assumption of OSDA by allowing unknown classes to present in both the source and the target domains. However, the same problem still remains as it requires labeled source and unlabeled target data in training time, which do not often comply with TTA settings where the model has no access to train data at test time due to privacy issues [38].

# 6    Discussion and conclusion

We investigate the problem of having noisy samples and the performance degradations caused by those samples in existing TTA methods. To address this issue, we propose SoTTA that is robust to noisy samples by high-confidence uniform-class sampling and entropy-sharpness minimization. Our evaluation with four noisy scenarios reveals that SoTTA outperforms state-of-the-art TTA methods in those scenarios. We believe that the takeaways from this study are a meaningful stride towards practical advances in overcoming domain shifts in test time.

**Limitations and future work.** While we focus on four noisy test stream scenarios, real-world test streams might have other types of sample diversities that are not considered in this work. Furthermore, recent TTA algorithms consider various temporal distribution shifts, such as temporally-correlated streams [5] and domain changes [39]. Towards developing a TTA algorithm robust to any test streams in the wild, more comprehensive and realistic considerations should be taken into account, which we believe is a meaningful future direction.

**Potential negative societal impacts.** As TTA requires continual computations for every test sample for adaptation, environmental concerns might be raised such as carbon emissions [37]. Recent studies such as memory-economic TTA [12] might be an effective way to mitigate this problem.

# Acknowledgments and Disclosure of Funding

This work was supported by the Institute of Information & communications Technology Planning & Evaluation (IITP) grant funded by the Korea government (MSIT) (No.2022-0-00495, On-Device Voice Phishing Call Detection).

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

# A Experiment details

We conducted all experiments in the paper using three random seeds (0, 1, 2) and reported the average accuracies and their corresponding standard deviations. The experiments were performed on NVIDIA GeForce RTX 3090 and NVIDIA TITAN RTX GPUs. For a single execution of SoTTA, the test-time adaptation phase consumed 1 minutes for CIFAR10-C/CIFAR100-C and 10 minutes for ImageNet-C.

## A.1 Baseline details

In this study, we utilized the official implementations of the baseline methods. To ensure consistency, we adopted the reported best hyperparameters documented in the respective papers or source code repositories. Furthermore, we present supplementary information regarding the implementation specifics of the baseline methods and provide a comprehensive overview of our experimental setup, including detailed descriptions of the employed hyperparameters.

**SoTTA (Ours).** We used ADAM optimizer [14], with a BN momentum of $m = 0.2$, and learning rate of $l = 0.001$ with a single adaptation epoch. We set the HUS size to 64 and the confidence threshold $C_0$ to 0.99 for CIFAR10-C (10 classes), 0.66 for CIFAR100-C (100 classes), and 0.33 for ImageNet-C (1,000 classes). We set entropy-sharpness L2-norm constraint $\rho = 0.5$ following the suggestion [4].

**PL.** For PL [17], we only updated the BN layers following the previous studies [38, 39]. We set the learning rate as $LR = 0.001$ as the same as TENT [38].

**TENT.** For TENT [38], we set the learning rate as $LR = 0.001$ for CIFAR10-C and $LR = 0.00025$ for ImageNet-C, following the guidance provided in the original paper. We referred to the official code[3] for implementations.

**LAME.** LAME [1] relies on an affinity matrix and incorporates hyperparameters associated with it. We followed the hyperparameter selection specified by the authors in their paper and referred to their official code[4] for implementation details. Specifically, we employed the kNN affinity matrix with a value of k set to 5.

**CoTTA.** CoTTA [39] incorporates three hyperparameters: the augmentation confidence threshold $p_{th}$, restoration factor $p$, and exponential moving average (EMA) factor $m$. To ensure consistency, we adopted the hyperparameter values recommended by the authors. Specifically, we set the restoration factor to $p = 0.01$ and the EMA factor to $\alpha = 0.999$. For the augmentation confidence threshold, the authors provide a guideline for its selection, suggesting using the 5% quantile of the softmax predictions' confidence on the source domains. We followed this guideline, which results in $p_{th} = 0.92$ for CIFAR10-C, $p_{th} = 0.72$ for CIFAR100-C, and $p_{th} = 0.1$ for ImageNet-C. We referred to the official code[5] for implementing CoTTA.

**EATA.** For EATA [28], we followed the settings from the original paper. We set $LR = 0.005/0.005/0.00025$ for CIFAR10-C/CIFAR100-C/ImageNet-C, entropy constant $E_0 = 0.4 \times \ln|\mathcal{Y}|$ where $|\mathcal{Y}|$ is number of classes. We set cosine sample similarity threshold $\epsilon = 0.4/0.4/0.05$, trade-off parameter $\beta = 1/1/2,000$, the moving average factor $\alpha = 0.1$. We utilized 2,000 samples for calculating Fisher importance as suggested. We referred to the official code[6] for implementing EATA.

**SAR.** SAR [29] aims to adapt to diverse batch sizes, and we chose a typical batch size of 64 for a fair comparison. We followed the learning rate as $LR = 0.00025$, sharpness threshold $\rho = 0.5$, and entropy threshold $E_0 = 0.4 \times \ln|\mathcal{Y}|$ where $|\mathcal{Y}|$ is the total number of classes, as suggested in the original paper. Finally, we froze the top layer (layer4 for ResNet18) as the original paper, and SoTTA also follows this implementation. We referred to the original code[7] for implementing SAR.

---

[3] https://github.com/DequanWang/tent
[4] https://github.com/fiveai/LAME
[5] https://github.com/qinenergy/cotta
[6] https://github.com/mr-eggplant/EATA
[7] https://github.com/mr-eggplant/SAR

**RoTTA.** RoTTA [44] uses Adam Optimizer by setting learning rate as $LR = 0.001$ and $\beta = 0.9$. We followed the authors' hyperparameters selection from the paper, including BN-statistic exponential moving average updating rate as $\alpha = 0.05$, the Teacher model's exponential moving average updating rate as $\nu = 0.001$, timeliness parameter as $\lambda_t = 1.0$, and uncertainty parameter as $\lambda_u = 1.0$. We referred to the original code[8] for implementing RoTTA.

### A.2 Target dataset details

**CIFAR10-C/CIFAR100-C.** CIFAR10-C/CIFAR100-C [9] serves as a widely adopted benchmark for evaluating the robustness of models against corruptions [27, 36, 38, 39]. Both datasets consist of 50,000 training samples and 10,000 test samples, categorized into 10/100 classes. To assess the robustness of models, datasets introduce 15 types of corruptions to the test data, including Gaussian Noise, Shot Noise, Impulse Noise, Defocus Blur, Frosted Glass Blur, Motion Blur, Zoom Blur, Snow, Frost, Fog, Brightness, Contrast, Elastic Transformation, Pixelate, and JPEG Compression. For our experiments, we adopt the highest severity level of corruption, level 5, in line with previous studies [27, 36, 38, 39]. Consequently, the datasets consist of 150,000 corrupted test samples. To train our models, we employ the ResNet18 [8] architecture as the backbone network. The model is trained on the clean training data to generate the source models. We utilize stochastic gradient descent with a momentum of 0.9 and cosine annealing learning rate scheduling [22] for 200 epochs. The initial learning rate is set to 0.1, and a batch size 128 is used during training.

**ImageNet-C.** ImageNet-C is another widely adopted benchmark for evaluating the robustness of models against corruptions [1, 27, 36, 38, 39]. The ImageNet dataset [3] consists of 1,281,167 training samples and 50,000 test samples. Similar to CIFAR10-C, ImageNet-C applies the same 15 types of corruptions, resulting in 750,000 corrupted test samples. We utilize the highest severity level of corruption, equivalent to CIFAR10-C. For our experiments, we employ a pre-trained ResNet18 [8] model from the TorchVision library [23], which is pre-trained on the ImageNet dataset [3] and is widely used as a backbone for various computer vision tasks.

### A.3 Noisy dataset details

**CIFAR100 (Near).** CIFAR100 [15] consists of 50,000/10,000 training/test data with 100 classes. We utilized training data without any corruption. We undersampled the dataset to 10,000 for the CIFAR10-C and CIFAR100-C target cases by randomly removing samples and used the entire training set (50,000) for the ImageNet-C target case.

**ImageNet (Near).** ImageNet [3] consists of 1,281,167/50,000 training/test data with 1,000 classes. We utilized test data without any corruption. We undersampled the dataset to 10,000 for the CIFAR100-C target case by randomly removing samples.

**MNIST (Far).** MNIST [16] contains 60,000/10,000 training/test data with 10 classes. We utilized test data without any corruption. We used the entire test set for the CIFAR10-C/CIFAR100-C target case, and oversampled the dataset by randomly resampling, which results in 50,000 samples that is equivalent to the size of each ImageNet-C target data.

**Attack.** We implemented the modified indiscriminate distribution invading attack (DIA) [40]. First, we duplicated the entire set of target samples and treated them as malicious samples. Subsequently, we randomly shuffled these duplicated samples within the original target sample set. During the adaptation phase, we injected perturbations into the malicious samples to increase the overall error rate on benign samples within the same batch. As a result, we perturbed 10,000 samples (CIFAR10-C/CIFAR100-C) and 50,000 samples (ImageNet-C) to serve as attack samples. For CIFAR10-C/CIFAR100-C, we used hyperparameters of maximum perturbation constraint $\epsilon = 0.1$, attack learning rate $\alpha = 1/255$, and attacking steps $N = 10$. For ImageNet-C, we used hyperparameters of maximum perturbation constraint $\epsilon = 0.2$, attack learning rate $\alpha = 1/255$, and attacking steps $N = 1$.

**Uniform random noise (Noise).** We generated a uniform random valued image in the scaled RGB range $[0, 1]$, with the same height and width as the corresponding target dataset. We generated the same amount of noise samples as each target dataset.

---

[8] https://github.com/BIT-DA/RoTTA

# B    Result details

Table 5: Classification accuracy (%) and their corresponding standard deviations on CIFAR10-C for 15 types of corruptions under five scenarios. **Bold** numbers are the highest accuracy. Averaged over three different random seeds.

| | Method | Noise | | | Blur | | | | Weather | | | | Digital | | | | Avg. |
| --- | --- | --- | --- | --- | --- | --- | --- | --- | --- | --- | --- | --- | --- | --- | --- | --- | --- |
| | | Gau. | Shot | Imp. | Def. | Gla. | Mot. | Zoom | Snow | Fro. | Fog | Brit. | Cont. | Elas. | Pix. | JPEG | |
| Benign | Source | 26.0 ±3.3 | 33.2 ±3.5 | 24.7 ±4.2 | 56.7 ±2.7 | 52.0 ±2.7 | 67.4 ±1.2 | 64.8 ±2.6 | 78.0 ±0.4 | 67.0 ±2.5 | 74.1 ±0.8 | 91.5 ±0.3 | 33.9 ±1.8 | 76.6 ±0.7 | 46.4 ±0.6 | 73.2 ±0.8 | 57.7 ±1.0 |
| | BN Stats [27] | 67.0 ±0.5 | 69.0 ±0.8 | 60.4 ±0.9 | 87.8 ±0.2 | 65.6 ±0.1 | 86.3 ±0.4 | 87.4 ±0.4 | 81.6 ±0.4 | 80.3 ±0.6 | 85.4 ±0.2 | 90.7 ±0.4 | 86.9 ±0.4 | 76.7 ±0.2 | 79.3 ±0.0 | 71.9 ±0.6 | 78.4 ±0.3 |
| | PL [17] | 71.1 ±1.2 | 72.9 ±0.6 | 62.2 ±1.3 | 86.9 ±0.0 | 64.4 ±0.4 | 85.3 ±1.2 | 86.6 ±0.9 | 80.8 ±1.0 | 78.8 ±1.8 | 84.9 ±0.3 | 89.6 ±1.0 | 84.0 ±0.2 | 76.2 ±0.6 | 80.0 ±0.7 | 73.1 ±1.4 | 78.5 ±0.3 |
| | TENT [38] | 74.5 ±0.7 | 77.6 ±0.8 | 66.6 ±1.3 | 88.2 ±0.3 | 66.2 ±2.0 | 86.9 ±0.8 | 88.8 ±0.4 | 83.7 ±0.5 | 81.3 ±1.3 | 86.0 ±1.5 | 91.1 ±0.4 | 86.9 ±0.3 | 77.9 ±0.9 | 82.7 ±1.4 | 76.7 ±0.4 | 81.0 ±0.4 |
| | LAME [1] | 21.8 ±3.6 | 29.2 ±4.0 | 19.7 ±4.7 | 53.3 ±1.6 | 52.1 ±3.7 | 65.9 ±0.3 | 62.5 ±1.4 | 79.2 ±0.7 | 69.3 ±4.4 | 73.1 ±1.7 | 90.1 ±0.3 | 28.0 ±1.1 | 75.7 ±0.7 | 43.8 ±1.1 | 74.1 ±0.9 | 55.9 ±0.5 |
| | CoTTA [39] | **76.9 ±0.6** | **78.6 ±0.1** | **72.3 ±0.2** | 88.2 ±0.5 | **70.9 ±1.0** | 86.8 ±0.2 | 88.1 ±0.5 | 83.4 ±0.3 | 83.4 ±0.5 | 86.1 ±0.5 | 91.2 ±0.2 | 84.9 ±0.2 | 79.2 ±0.5 | 83.0 ±0.3 | **79.9 ±0.6** | 82.2 ±0.2 |
| | EATA [28] | 76.0 ±0.8 | 78.2 ±0.8 | 68.2 ±0.7 | 88.4 ±1.9 | 70.1 ±0.5 | 87.4 ±0.5 | 88.4 ±0.1 | 84.5 ±0.3 | **85.0 ±0.3** | 88.0 ±0.2 | 91.5 ±0.3 | **89.9 ±0.6** | 77.8 ±0.5 | **84.8 ±0.5** | 78.4 ±1.2 | **82.4 ±0.2** |
| | SAR [29] | 68.3 ±1.2 | 69.7 ±1 | 58.9 ±6.9 | 87.8 ±0.2 | 62.9 ±5.3 | 86.3 ±0.1 | 87.4 ±0.4 | 81.6 ±0.4 | 80.3 ±0.5 | 85.4 ±0.6 | 90.7 ±0.2 | 86.9 ±0.4 | 76.7 ±0.2 | 79.3 ±0 | 72.0 ±0.5 | 78.3 ±0.7 |
| | RoTTA [44] | 65.2 ±0.8 | 67.4 ±1.1 | 58.3 ±0.9 | 87.2 ±0.2 | 64.4 ±1.2 | 85.8 ±0.5 | 87.3 ±0.5 | 81.2 ±1 | 76.9 ±1 | 85.3 ±0.6 | 90.7 ±0.5 | 57.2 ±2.9 | 76.5 ±0.5 | 77.7 ±0.4 | 71.6 ±0.9 | 75.5 ±0.7 |
| | SoTTA | 75.0 ±1.1 | 77.5 ±0.6 | 68.8 ±0.7 | **88.8 ±0.4** | 70.7 ±1.2 | **87.5 ±0.5** | **89.0 ±0.5** | **85.4 ±0.3** | 84.0 ±0.7 | **88.2 ±0.2** | **91.9 ±0.1** | 83.9 ±1.5 | **79.8 ±0.4** | 83.9 ±0.5 | 78.3 ±0.7 | 82.2 ±0.3 |
| Near | Source | 26.0 ±3.3 | 33.2 ±3.5 | 24.7 ±4.2 | 56.7 ±2.7 | 52.0 ±2.7 | 67.4 ±1.2 | 64.8 ±2.6 | 78.0 ±0.4 | 67.0 ±2.5 | 74.1 ±0.8 | 91.5 ±0.3 | 33.9 ±1.8 | 76.6 ±0.7 | 46.4 ±0.6 | 73.2 ±0.8 | 57.7 ±1.0 |
| | BN Stats [27] | 64.9 ±0.8 | 66.3 ±0.9 | 58.2 ±0.7 | 84.1 ±0.5 | 62.7 ±0.6 | 84.2 ±0.4 | 85.0 ±0.2 | 83.1 ±0.6 | 82.5 ±0.5 | 85.5 ±0.5 | 92.4 ±0.1 | 85.3 ±0.2 | 76.5 ±0.6 | 67.4 ±0.3 | 70.3 ±1.0 | 76.6 ±0.4 |
| | PL [17] | 63.2 ±1.0 | 63.5 ±3.4 | 51.7 ±4.4 | 81.6 ±1.0 | 58.4 ±1.5 | 78.4 ±1.6 | 83.3 ±0.6 | 79.2 ±0.5 | 79.7 ±2.1 | 80.3 ±1.9 | 89.2 ±1.1 | 79.5 ±0.1 | 73.1 ±1.6 | 70.9 ±0.6 | 69.3 ±1.7 | 73.4 ±0.2 |
| | TENT [38] | 64.7 ±3.6 | 64.7 ±5.4 | 50.2 ±4.9 | 81.3 ±2.3 | 59.6 ±2.9 | 80.8 ±1.1 | 83.8 ±1.0 | 79.6 ±1.1 | 78.3 ±2.7 | 80.0 ±3.8 | 88.6 ±1.0 | 83.7 ±1.4 | 73.5 ±1.0 | 74.3 ±0.5 | 71.1 ±2.8 | 74.3 ±0.9 |
| | LAME [1] | 24.3 ±3.0 | 31.6 ±3.3 | 19.9 ±4.3 | 53.9 ±1.4 | 53.2 ±3.6 | 65.9 ±0.7 | 62.5 ±1.2 | 79.0 ±0.4 | 69.5 ±3.8 | 73.1 ±1.5 | 90.1 ±0.2 | 28.4 ±1.1 | 75.0 ±0.7 | 44.8 ±1.0 | 74.2 ±1.1 | 56.4 ±0.6 |
| | CoTTA [39] | 72.7 ±0.2 | 74.3 ±0.7 | 66.0 ±0.1 | 82.6 ±0.6 | **67.6 ±0.9** | 81.8 ±0.5 | 84.1 ±0.4 | 84.1 ±0.8 | **85.5 ±0.3** | 82.5 ±1.1 | 91.1 ±0.3 | 69.9 ±0.5 | 78.1 ±1.0 | 76.1 ±0.5 | **79.3 ±0.7** | 78.4 ±0.4 |
| | EATA [28] | 51.5 ±3.4 | 50.3 ±2.4 | 40.1 ±2.2 | 70.4 ±5.6 | 45.8 ±2.4 | 72.8 ±0.9 | 77.2 ±1.4 | 66.8 ±2.0 | 67.4 ±5.0 | 74.4 ±1.2 | 83.9 ±1.2 | 68.2 ±7.5 | 60.2 ±1.8 | 67.3 ±2.0 | 62.1 ±5.4 | 63.9 ±0.4 |
| | SAR [29] | 59.0 ±15.4 | 60.9 ±14.3 | 52.8 ±14.4 | 78.2 ±10.0 | 55.4 ±14.7 | 83.7 ±0.7 | 81.8 ±5.7 | 78.8 ±7.2 | 78.6 ±6.7 | 85.5 ±0.5 | 92.4 ±0.1 | 85.3 ±0.1 | 66.6 ±17.8 | 64.0 ±12.2 | 63.8 ±14.4 | 72.4 ±8.8 |
| | RoTTA [44] | 66.3 ±0.7 | 68.3 ±1.3 | 59.4 ±0.7 | 86.0 ±0.4 | 63.2 ±0.7 | 85.4 ±0.3 | 87.0 ±0.4 | 83.5 ±0.8 | 82.8 ±0.7 | 86.4 ±0.5 | 92.4 ±0.3 | 84.8 ±1.4 | 77.1 ±0.8 | 71.6 ±0.4 | 71.6 ±0.6 | 77.7 ±0.6 |
| | SoTTA | 74.3 ±1.4 | 76.7 ±0.9 | 66.5 ±2.2 | 87.5 ±0.1 | 66.9 ±0.8 | 86.4 ±0.6 | 87.8 ±0.5 | 84.4 ±0.6 | 83.8 ±0.2 | 87.2 ±0.5 | 91.3 ±0.2 | 88.4 ±0.7 | 78.7 ±1.1 | 82.4 ±0.5 | 78.0 ±0.6 | 81.4 ±0.5 |
| Far | Source | 26.0 ±3.3 | 33.2 ±3.5 | 24.7 ±4.2 | 56.7 ±2.7 | 52.0 ±2.7 | 67.4 ±1.2 | 64.8 ±2.6 | 78.0 ±0.4 | 67.0 ±2.5 | 74.1 ±0.8 | 91.5 ±0.3 | 33.9 ±1.8 | 76.6 ±0.7 | 46.4 ±0.6 | 73.2 ±0.8 | 57.7 ±1.0 |
| | BN Stats [27] | 62.7 ±1.3 | 66.1 ±1.4 | 56.0 ±0.9 | 86.5 ±0.2 | 60.7 ±0.7 | 84.2 ±0.8 | 87.2 ±0.1 | 79.8 ±0.4 | 78.4 ±0.2 | 85.7 ±0.2 | 89.9 ±0.0 | 80.2 ±1.0 | 75.5 ±0.6 | 69.8 ±1.0 | 65.0 ±0.1 | 75.2 ±0.3 |
| | PL [17] | 55.2 ±2.9 | 54.1 ±5.2 | 48.3 ±5.2 | 83.2 ±0.8 | 49.3 ±4.4 | 80.0 ±2.4 | 83.0 ±1.9 | 76.8 ±1.2 | 73.3 ±2.8 | 80.6 ±1.9 | 87.6 ±1.4 | 74.6 ±2.0 | 70.5 ±2.7 | 66.8 ±5.9 | 63.6 ±2.6 | 69.8 ±1.5 |
| | TENT [38] | 51.6 ±8.6 | 57.4 ±6.2 | 43.7 ±11.4 | 84.8 ±0.8 | 43.5 ±7.5 | 83.3 ±0.8 | 85.3 ±0.6 | 80.4 ±1.1 | 73.1 ±3.5 | 83.5 ±0.4 | 88.9 ±0.4 | 80.8 ±7.5 | 73.5 ±1.7 | 72.2 ±2.4 | 66.7 ±4.3 | 71.2 ±1.0 |
| | LAME [1] | 22.8 ±3.4 | 29.6 ±3.7 | 19.3 ±4.2 | 53.2 ±1.6 | 50.4 ±3.7 | 64.6 ±0.8 | 60.7 ±1.2 | 79.1 ±0.7 | 67.9 ±4.0 | 72.8 ±1.5 | 90.1 ±0.2 | 28.1 ±1.1 | 74.7 ±0.8 | 44.4 ±0.7 | 74.3 ±1.0 | 55.5 ±0.4 |
| | CoTTA [39] | 67.4 ±1.9 | 71.1 ±1.0 | 59.4 ±2.7 | 83.3 ±0.5 | 61.2 ±0.6 | 82.3 ±0.9 | 84.3 ±0.3 | 80.4 ±1.2 | 80.4 ±1.5 | 83.8 ±1.3 | 87.2 ±1.4 | 58.0 ±4.7 | 76.0 ±0.4 | 70.3 ±3.5 | 72.9 ±2.7 | 74.5 ±1.2 |
| | EATA [28] | 40.0 ±2.3 | 46.4 ±3.9 | 34.9 ±0.9 | 73.5 ±0.3 | 35.2 ±2.9 | 59.3 ±2.8 | 76.4 ±0.7 | 61.1 ±3.0 | 59.5 ±2.9 | 68.4 ±7.1 | 85.2 ±2.1 | 55.2 ±9.0 | 46.5 ±1.2 | 53.5 ±5.1 | 49.0 ±5.5 | 56.3 ±0.5 |
| | SAR [29] | 60.3 ±7.7 | 62.6 ±7.9 | 50.9 ±9.8 | 86.5 ±0.2 | 55.4 ±9.6 | 84.3 ±0.7 | 87.2 ±0.1 | 79.8 ±0.3 | 78.3 ±0.5 | 85.7 ±0.2 | 89.9 ±0.1 | 80.2 ±0.1 | 70.1 ±10.1 | 67.8 ±6.2 | 60.9 ±9.5 | 73.3 ±3.9 |
| | RoTTA [44] | 67.3 ±0.5 | 69.8 ±1.0 | 61.1 ±1.9 | 88.1 ±0.4 | 66.0 ±0.9 | 86.6 ±0.5 | 88.0 ±0.6 | 82.0 ±1.4 | 78.8 ±0.6 | 86.2 ±0.4 | 91.2 ±1.1 | 61.7 ±9.5 | 77.5 ±0.8 | 79.6 ±1.0 | 73.0 ±1.0 | 77.1 ±1.1 |
| | SoTTA | 73.3 ±1.2 | 76.3 ±1.9 | 66.3 ±2.5 | 88.5 ±0.6 | 68.3 ±2.3 | 86.8 ±0.7 | 88.3 ±0.2 | 84.1 ±1.0 | 84.2 ±0.6 | 87.2 ±0.4 | 92.0 ±0.4 | 89.0 ±1.1 | 77.8 ±1.8 | 83.8 ±0.9 | 77.8 ±1.2 | 81.6 ±0.6 |
| Attack | Source | 26.0 ±3.3 | 33.2 ±3.5 | 24.7 ±4.2 | 56.7 ±2.7 | 52.0 ±2.7 | 67.4 ±1.2 | 64.8 ±2.6 | 78.0 ±0.4 | 67.0 ±2.5 | 74.1 ±0.8 | 91.5 ±0.3 | 33.9 ±1.8 | 76.6 ±0.7 | 46.4 ±0.6 | 73.2 ±0.8 | 57.7 ±1.0 |
| | BN Stats [27] | 44.5 ±1.3 | 46.8 ±1.5 | 39.8 ±1.1 | 63.3 ±1.2 | 42.1 ±1.7 | 63.5 ±1.7 | 62.5 ±1.2 | 62.3 ±1.7 | 59.3 ±1.7 | 62.1 ±1.0 | 75.4 ±1.4 | 65.1 ±1.4 | 50.6 ±1.3 | 53.9 ±1.6 | 46.9 ±1.9 | 55.9 ±1.4 |
| | PL [17] | 59.1 ±0.9 | 61.4 ±1.5 | 53.1 ±2.0 | 73.1 ±0.9 | 51.9 ±2.0 | 71.7 ±1.8 | 72.3 ±1.1 | 71.3 ±1.7 | 69.1 ±1.7 | 69.8 ±0.8 | 81.5 ±0.9 | 72.1 ±0.7 | 60.9 ±1.8 | 67.4 ±1.2 | 60.1 ±1.4 | 66.3 ±1.3 |
| | TENT [38] | 62.9 ±0.6 | 65.2 ±0.3 | 56.7 ±1.1 | 74.8 ±1.0 | 54.5 ±2.0 | 73.4 ±0.8 | 75.2 ±0.9 | 73.8 ±0.6 | 71.8 ±1.8 | 72.1 ±0.6 | 83.0 ±0.7 | 73.7 ±0.5 | 63.1 ±1.3 | 70.0 ±1.0 | 64.0 ±1.4 | 68.9 ±0.9 |
| | LAME [1] | 21.0 ±3.5 | 28.0 ±4.1 | 17.3 ±4.3 | 52.7 ±1.9 | 52.9 ±3.6 | 65.9 ±0.5 | 62.0 ±1.2 | 79.3 ±0.6 | 70.4 ±4.5 | 73.9 ±1.6 | 90.3 ±0.2 | 28.4 ±1.0 | 75.6 ±0.7 | 44.2 ±1.0 | 74.2 ±1.1 | 55.9 ±0.5 |
| | CoTTA [39] | 53.1 ±1.6 | 57.0 ±1.2 | 49.7 ±1.8 | 67.9 ±0.7 | 55.9 ±2.4 | 69.7 ±1.6 | 71.3 ±0.6 | 74.3 ±1.8 | 69.5 ±1.9 | 68.1 ±2.0 | 84.9 ±0.4 | 47.0 ±2.9 | 68.6 ±1.9 | 62.3 ±1.6 | 71.6 ±1.3 | 69.5 ±1.5 |
| | EATA [28] | 66.6 ±0.4 | 68.7 ±0.5 | 57.9 ±0.5 | 75.2 ±0.7 | 57.2 ±0.8 | 74.7 ±1.0 | 75.5 ±0.9 | 75.1 ±1.0 | 73.5 ±0.8 | 73.8 ±0.5 | 83.5 ±0.6 | 76.0 ±0.9 | 64.3 ±1.0 | 73.5 ±0.6 | 67.8 ±1.1 | 70.9 ±0.6 |
| | SAR [29] | 46.1 ±3.9 | 48.1 ±3.5 | 40.3 ±1.7 | 63.3 ±1.2 | 42.2 ±1.6 | 63.5 ±1.8 | 62.5 ±1.2 | 62.3 ±1.7 | 59.3 ±1.7 | 62.1 ±1.5 | 75.4 ±1.0 | 65.1 ±1.4 | 50.6 ±1.2 | 53.9 ±1.7 | 47.6 ±2.9 | 56.2 ±1.8 |
| | RoTTA [44] | 69.7 ±0.7 | 71.8 ±1.0 | 62.9 ±1.0 | 88.6 ±0.3 | 67.8 ±0.7 | 87.4 ±0.2 | 88.7 ±0.6 | 83.2 ±0.7 | 80.1 ±0.8 | 87.1 ±0.3 | 91.8 ±0.3 | 63.5 ±3.7 | 78.4 ±0.8 | 80.6 ±0.2 | 74.1 ±0.7 | 78.4 ±0.7 |
| | SoTTA | 78.2 ±0.3 | 80.8 ±0.1 | 72.3 ±0.8 | 90.1 ±0.2 | 73.6 ±0.9 | 89.2 ±0.4 | 90.3 ±0.5 | 87.4 ±0.5 | 86.2 ±0.6 | 89.3 ±0.6 | 92.9 ±0.1 | 87.8 ±0.7 | 81.3 ±0.8 | 86.6 ±0.3 | 81.0 ±0.3 | 84.5 ±0.2 |
| Noise | Source | 26.0 ±3.3 | 33.2 ±3.5 | 24.7 ±4.2 | 56.7 ±2.7 | 52.0 ±2.7 | 67.4 ±1.2 | 64.8 ±2.6 | 78.0 ±0.4 | 67.0 ±2.5 | 74.1 ±0.8 | **91.5 ±0.3** | 33.9 ±1.8 | 76.6 ±0.7 | 46.4 ±0.6 | 73.2 ±0.8 | 57.7 ±1.0 |
| | BN Stats [27] | 51.7 ±0.3 | 53.9 ±0.6 | 45.5 ±0.7 | 52.7 ±2.0 | 41.5 ±1.7 | 51.0 ±0.7 | 55.1 ±1.5 | 62.8 ±0.7 | 63.8 ±0.2 | 53.8 ±0.6 | 76.9 ±0.3 | 55.8 ±2.5 | 46.8 ±1.8 | 54.8 ±0.7 | 56.4 ±1.0 | 54.8 ±0.8 |
| | PL [17] | 47.6 ±9.9 | 52.7 ±2.4 | 44.7 ±4.3 | 48.9 ±12.6 | 36.1 ±5.1 | 49.4 ±1.8 | 54.1 ±2.9 | 61.9 ±2.4 | 56.5 ±4.4 | 50.9 ±1.3 | 77.1 ±3.7 | 45.2 ±4.8 | 43.1 ±4.5 | 49.4 ±5.6 | 59.5 ±4.7 | 51.8 ±0.9 |
| | TENT [38] | 54.0 ±6.7 | 57.1 ±5.6 | 36.7 ±9.1 | 48.9 ±6.8 | 28.3 ±4.5 | 50.5 ±3.1 | 51.0 ±5.0 | 64.0 ±4.1 | 64.7 ±5.2 | 49.5 ±1.9 | 80.5 ±1.4 | 43.7 ±3.0 | 38.4 ±2.2 | 56.7 ±6.4 | 57.0 ±4.5 | 52.1 ±0.4 |
| | LAME [1] | 21.8 ±3.5 | 28.6 ±3.7 | 18.5 ±3.1 | 51.6 ±2.3 | 50.8 ±3.6 | 64.3 ±0.2 | 60.9 ±1.8 | 78.4 ±0.5 | 67.3 ±3.8 | 71.7 ±1.2 | 90.5 ±0.2 | 27.0 ±1.2 | 75.1 ±0.7 | 43.0 ±0.9 | 73.4 ±1.0 | 54.9 ±0.6 |
| | CoTTA [39] | 60.4 ±2.1 | 60.3 ±3.5 | 52.4 ±1.6 | 47.3 ±3.0 | 41.6 ±0.4 | 44.1 ±2.7 | 52.0 ±4.7 | 62.7 ±0.8 | 66.6 ±2.4 | 47.7 ±1.7 | 79.0 ±1.1 | 44.7 ±4.3 | 42.8 ±0.5 | 60.2 ±1.0 | 60.2 ±1.3 | 54.8 ±1.3 |
| | EATA [28] | 42.2 ±1.1 | 41.0 ±1.1 | 33.2 ±5.9 | 32.7 ±5.1 | 25.0 ±1.5 | 27.9 ±2.1 | 34.3 ±5.4 | 40.8 ±2.7 | 42.6 ±6.5 | 31.6 ±11.5 | 61.5 ±5.7 | 20.3 ±2.2 | 27.5 ±4.1 | 35.8 ±4.5 | 43.1 ±8.3 | 36.0 ±0.8 |
| | SAR [29] | 57.5 ±1.0 | 59.3 ±0.2 | 49.6 ±1.7 | 57.2 ±1.1 | 43.7 ±1.7 | 54.4 ±1.5 | 59.4 ±1.6 | 64.8 ±1.0 | 65.4 ±0.4 | 57.9 ±0.2 | 77.1 ±1.8 | 60.2 ±0.4 | 50.0 ±0.6 | 58.3 ±0.6 | 59.8 ±0.1 | 58.3 ±0.3 |
| | RoTTA [44] | 64.4 ±0.5 | 66.9 ±0.8 | 56.1 ±1.4 | 80.1 ±0.5 | 59.1 ±0.5 | 79.8 ±0.2 | 82.2 ±0.8 | 79.7 ±0.8 | 78.7 ±0.7 | 77.8 ±0.4 | 91.2 ±0.6 | 69.0 ±4.0 | 72.3 ±1.2 | 73.4 ±0.2 | 72.8 ±0.3 | 73.6 ±0.5 |
| | SoTTA | 73.3 ±1.5 | 77.7 ±0.8 | 66.8 ±1.8 | 86.1 ±2.1 | 64.0 ±2.8 | 84.3 ±0.7 | 86.6 ±1.1 | 83.1 ±0.7 | 82.0 ±1.8 | 85.7 ±2.7 | 91.1 ±0.4 | 84.1 ±2.4 | 77.1 ±3.3 | 81.6 ±2.8 | 77.2 ±2.2 | 80.0 ±1.4 |

Table 6: Classification accuracy (%) and their corresponding standard deviations on CIFAR100-C for 15 types of corruptions under five scenarios. **Bold** numbers are the highest accuracy. Averaged over three different random seeds.

| | | Noise | | | Blur | | | | Weather | | | | Digital | | | | |
| | Method | Gau. | Shot | Imp. | Def. | Gla. | Mot. | Zoom | Snow | Fro. | Fog | Brit. | Cont. | Elas. | Pix. | JPEG | Avg. |
|---|---|---|---|---|---|---|---|---|---|---|---|---|---|---|---|---|---|
| **Benign** | Source | 10.6 ±1.3 | 12.1 ±1.2 | 7.2 ±0.9 | 34.9 ±0.3 | 19.6 ±0.6 | 44.1 ±0.6 | 41.9 ±0.4 | 46.3 ±0.2 | 34.2 ±0.4 | 41.1 ±0.9 | 67.3 ±0.1 | 18.5 ±0.6 | 50.4 ±0.3 | 24.9 ±2.5 | 44.6 ±0.6 | 33.2 ±0.4 |
| | BN stats [27] | 39.2 ±0.9 | 40.7 ±0.6 | 34.1 ±0.7 | 66.1 ±0.1 | 42.5 ±0.4 | 63.6 ±0.6 | 64.8 ±0.1 | 53.8 ±0.5 | 53.5 ±0.2 | 58.1 ±0.3 | 68.2 ±0.2 | 64.5 ±0.5 | 53.9 ±0.4 | 56.6 ±0.7 | 45.2 ±0.3 | 53.7 ±0.2 |
| | PL [17] | 46.5 ±0.2 | 48.7 ±0.9 | 40.8 ±1.2 | 66.3 ±0.5 | 45.5 ±1.2 | 63.7 ±0.7 | 65.7 ±0.6 | 56.8 ±0.6 | 55.1 ±0.6 | 61.0 ±0.2 | 68.6 ±0.8 | 64.6 ±0.6 | 54.6 ±0.5 | 60.9 ±0.4 | 49.6 ±0.3 | 56.6 ±0.2 |
| | TENT [38] | 50.0 ±0.5 | 52.0 ±0.8 | 44.2 ±0.5 | 67.9 ±0.3 | 48.7 ±0.5 | 66.1 ±0.5 | 68.0 ±0.4 | 59.7 ±0.5 | 59.3 ±0.3 | 63.4 ±0.2 | 70.8 ±0.3 | **67.3** ±0.3 | 57.5 ±0.4 | 63.6 ±0.3 | 53.7 ±0.2 | 59.5 ±0.0 |
| | LAME [1] | 7.8 ±1.7 | 9.0 ±1.7 | 5.9 ±0.8 | 31.6 ±0.3 | 16.6 ±0.6 | 42.3 ±0.8 | 39.8 ±0.5 | 45.5 ±0.2 | 31.7 ±1.0 | 38.3 ±1.1 | 66.4 ±0.3 | 15.1 ±0.6 | 49.5 ±0.4 | 21.6 ±2.8 | 43.9 ±0.7 | 31.0 ±0.5 |
| | CoTTA [39] | 47.5 ±0.4 | 48.5 ±0.5 | 43.2 ±0.5 | 64.0 ±0.3 | 46.4 ±0.8 | 61.7 ±0.8 | 62.8 ±0.3 | 55.3 ±0.4 | 56.1 ±0.3 | 56.7 ±0.4 | 68.1 ±0.1 | 58.4 ±0.1 | 54.3 ±0.4 | 60.2 ±0.2 | 53.4 ±0.4 | 55.8 ±0.4 |
| | EATA [28] | 11.1 ±1.7 | 12.2 ±1.3 | 7.2 ±0.2 | 35.0 ±1.2 | 10.4 ±1.0 | 31.5 ±4.6 | 39.6 ±7.0 | 21.6 ±1.8 | 17.9 ±2.0 | 23.6 ±3.2 | 56.4 ±5.5 | 34.7 ±1.9 | 15.2 ±3.4 | 21.7 ±1.1 | 14.7 ±1.1 | 23.5 ±1.9 |
| | SAR [29] | 46.5 ±0.6 | 48.5 ±0.8 | 40.9 ±0.7 | 67.4 ±0.3 | 46.1 ±0.6 | 64.9 ±0.8 | 66.3 ±0.3 | 56.9 ±0.5 | 56.4 ±0.3 | 61.2 ±0.4 | 69.8 ±0.2 | 66.8 ±0.5 | 56.1 ±0.3 | 60.3 ±0.4 | 50.8 ±0.3 | 57.3 ±0.3 |
| | RoTTA [44] | 35.7 ±1.2 | 36.9 ±1.1 | 31.6 ±1.0 | 63.9 ±0.3 | 40.3 ±0.2 | 61.6 ±0.7 | 63.0 ±0.8 | 51.2 ±0.5 | 44.1 ±0.9 | 56.4 ±0.2 | 66.1 ±0.5 | 31.5 ±1.4 | 52.3 ±0.6 | 52.9 ±1.1 | 43.1 ±0.6 | 48.7 ±0.6 |
| | SoTTA | **52.0** ±0.6 | **53.4** ±0.4 | **45.0** ±1.1 | **68.8** ±0.2 | **49.1** ±0.8 | **66.7** ±0.6 | **69.0** ±0.2 | **61.7** ±0.6 | **60.2** ±0.2 | **64.7** ±0.6 | **72.2** ±0.3 | 66.4 ±0.5 | **58.6** ±0.5 | **64.1** ±0.2 | **55.0** ±0.4 | **60.5** ±0.0 |
| **Near** | Source | 10.6 ±1.3 | 12.1 ±1.2 | 7.2 ±0.9 | 34.9 ±0.3 | 19.6 ±0.6 | 44.1 ±0.6 | 41.9 ±0.4 | 46.3 ±0.2 | 34.2 ±0.4 | 41.1 ±0.9 | 67.3 ±0.1 | 18.5 ±0.6 | 50.4 ±0.3 | 24.9 ±2.5 | 44.6 ±0.6 | 33.2 ±0.4 |
| | BN stats [27] | 36.0 ±0.6 | 37.1 ±0.6 | 31.5 ±1.0 | 58.6 ±0.3 | 37.7 ±0.2 | 58.2 ±0.1 | 60.1 ±0.8 | 56.1 ±0.4 | 56.1 ±0.3 | 56.6 ±0.1 | 71.7 ±0.3 | 54.9 ±0.5 | 52.4 ±0.5 | 49.8 ±0.7 | 45.4 ±0.6 | 50.8 ±0.1 |
| | PL [17] | 32.4 ±2.0 | 32.1 ±2.1 | 26.5 ±1.7 | 58.4 ±1.8 | 33.5 ±0.6 | 56.9 ±1.5 | 58.8 ±1.9 | 51.5 ±0.6 | 50.5 ±1.5 | 53.4 ±2.0 | 66.7 ±1.2 | 51.4 ±3.9 | 49.3 ±0.9 | 53.1 ±0.5 | 45.3 ±0.8 | 48.0 ±0.3 |
| | TENT [38] | 26.8 ±5.6 | 27.1 ±0.6 | 21.4 ±2.9 | 58.7 ±2.1 | 24.5 ±2.1 | 58.0 ±1.1 | 60.8 ±0.8 | 50.6 ±1.1 | 47.7 ±2.2 | 52.6 ±2.1 | 66.3 ±1.2 | 58.2 ±0.0 | 46.2 ±0.3 | 52.1 ±1.0 | 44.5 ±1.6 | 46.4 ±1.4 |
| | LAME [1] | 8.1 ±1.6 | 9.6 ±1.6 | 5.9 ±0.9 | 32.6 ±0.4 | 17.2 ±0.5 | 43.0 ±0.6 | 40.4 ±0.6 | 45.7 ±0.4 | 32.7 ±0.7 | 39.1 ±0.8 | 66.8 ±0.1 | 15.6 ±0.6 | 49.8 ±0.4 | 22.3 ±2.5 | 44.0 ±0.6 | 31.5 ±0.5 |
| | CoTTA [39] | 40.5 ±0.8 | 41.3 ±0.3 | 36.9 ±0.5 | 51.5 ±0.6 | 39.5 ±0.5 | 53.4 ±0.4 | 54.8 ±0.9 | 56.8 ±0.4 | 57.2 ±0.5 | 52.6 ±0.8 | 67.3 ±0.1 | 38.2 ±0.5 | 51.3 ±0.9 | 56.4 ±0.1 | 52.7 ±0.3 | 50.0 ±0.3 |
| | EATA [28] | 4.5 ±0.9 | 4.3 ±0.3 | 3.6 ±0.3 | 7.4 ±0.4 | 4.9 ±0.3 | 7.6 ±1.3 | 7.5 ±0.6 | 7.0 ±1.3 | 5.7 ±1.0 | 5.8 ±0.2 | 9.9 ±1.3 | 5.2 ±0.4 | 6.6 ±0.3 | 5.7 ±0.5 | 5.7 ±0.7 | 6.1 ±0.3 |
| | SAR [29] | 43.3 ±0.1 | 44.6 ±0.7 | 38.3 ±0.9 | 62.6 ±0.5 | 40.5 ±0.3 | 61.5 ±0.4 | 63.5 ±0.8 | 58.6 ±0.5 | **58.5** ±0.1 | 61.4 ±0.5 | 72.1 ±0.4 | 62.0 ±0.3 | **54.6** ±0.1 | 57.5 ±0.3 | 51.6 ±0.5 | 55.4 ±0.1 |
| | RoTTA [44] | 36.6 ±0.7 | 38.6 ±0.4 | 30.5 ±1.5 | 64.0 ±0.4 | 38.1 ±0.1 | 61.9 ±0.6 | 63.7 ±0.5 | 55.1 ±1.0 | 50.3 ±2.2 | 58.4 ±0.3 | 68.2 ±0.6 | 26.6 ±1.1 | 52.7 ±0.6 | 52.3 ±1.2 | 44.3 ±1.0 | 49.4 ±0.5 |
| | SoTTA | 47.2 ±1.1 | 48.5 ±0.7 | 40.4 ±1.8 | 64.8 ±0.4 | 42.4 ±0.3 | 63.4 ±0.3 | 65.8 ±0.5 | 59.1 ±0.3 | 58.2 ±0.6 | 62.2 ±0.5 | 70.8 ±0.3 | 65.8 ±0.5 | 54.3 ±0.1 | 60.7 ±0.4 | 53.4 ±0.7 | 57.1 ±0.2 |
| **Far** | Source | 10.6 ±1.3 | 12.1 ±1.2 | 7.2 ±0.9 | 34.9 ±0.3 | 19.6 ±0.6 | 44.1 ±0.6 | 41.9 ±0.4 | 46.3 ±0.2 | 34.2 ±0.4 | 41.1 ±0.9 | 67.3 ±0.1 | 18.5 ±0.6 | 50.4 ±0.3 | 24.9 ±2.5 | 44.6 ±0.6 | 33.2 ±0.4 |
| | BN stats [27] | 32.5 ±0.4 | 34.4 ±0.4 | 27.9 ±0.7 | 59.1 ±0.7 | 34.2 ±0.3 | 56.3 ±0.8 | 59.6 ±0.2 | 48.4 ±0.1 | 48.6 ±0.2 | 53.5 ±0.4 | 64.1 ±0.4 | 51.9 ±0.6 | 47.6 ±0.7 | 46.9 ±0.4 | 37.0 ±0.4 | 46.8 ±0.1 |
| | PL [17] | 24.7 ±1.9 | 26.3 ±3.6 | 19.7 ±1.0 | 57.6 ±1.1 | 26.1 ±2.7 | 53.7 ±1.7 | 57.5 ±1.7 | 47.2 ±1.8 | 44.0 ±0.5 | 50.2 ±1.4 | 62.1 ±0.8 | 48.1 ±1.8 | 42.0 ±0.7 | 45.6 ±2.1 | 37.0 ±1.3 | 42.8 ±0.7 |
| | TENT [38] | 17.3 ±2.1 | 16.9 ±2.0 | 13.9 ±1.6 | 57.5 ±2.2 | 19.8 ±2.1 | 55.0 ±0.5 | 60.0 ±0.6 | 39.9 ±5.2 | 40.8 ±2.9 | 49.8 ±2.7 | 63.6 ±0.6 | 51.6 ±4.5 | 37.0 ±2.7 | 44.5 ±3.7 | 32.7 ±3.6 | 40.0 ±1.3 |
| | LAME [1] | 7.8 ±1.7 | 9.2 ±1.8 | 5.9 ±1.0 | 31.2 ±0.9 | 17.0 ±0.6 | 41.4 ±1.2 | 38.5 ±1.2 | 44.9 ±0.4 | 31.9 ±1.1 | 38.6 ±0.7 | 65.5 ±0.9 | 14.9 ±0.9 | 49.2 ±0.3 | 22.1 ±2.6 | 43.3 ±0.4 | 30.8 ±0.7 |
| | CoTTA [39] | 32.1 ±0.7 | 34.5 ±1.2 | 28.6 ±0.4 | 47.2 ±0.9 | 32.7 ±1.2 | 49.8 ±0.4 | 51.1 ±0.5 | 45.9 ±2.5 | 46.7 ±0.2 | 49.3 ±0.7 | 56.7 ±0.8 | 29.8 ±0.7 | 44.0 ±0.2 | 46.4 ±0.8 | 41.8 ±0.6 | 42.4 ±0.4 |
| | EATA [28] | 3.3 ±0.4 | 3.4 ±0.5 | 3.2 ±0.6 | 6.7 ±0.9 | 3.6 ±1.1 | 5.9 ±0.5 | 6.8 ±1.2 | 4.4 ±0.7 | 4.5 ±0.1 | 4.6 ±1.0 | 7.2 ±0.3 | 4.3 ±1.0 | 4.1 ±0.3 | 5.0 ±0.8 | 4.8 ±0.8 | 4.8 ±0.5 |
| | SAR [29] | 37.4 ±0.4 | 38.9 ±1.1 | 32.2 ±0.8 | 62.3 ±0.3 | 36.9 ±0.3 | 60.3 ±0.1 | 63.3 ±0.4 | 51.8 ±0.7 | 52.5 ±0.4 | 56.7 ±0.7 | 66.8 ±0.3 | 61.1 ±0.7 | 50.4 ±0.7 | 53.6 ±0.8 | 44.2 ±0.1 | 51.2 ±0.1 |
| | RoTTA [44] | 39.3 ±2.4 | 40.6 ±2.2 | 35.2 ±2.2 | 64.7 ±0.3 | 42.3 ±1.6 | 62.4 ±0.6 | 63.7 ±0.5 | 51.8 ±0.9 | 45.3 ±1.7 | 57.4 ±0.3 | 66.2 ±0.9 | 26.3 ±0.7 | 52.6 ±0.6 | 55.2 ±1.1 | 44.0 ±1.4 | 49.8 ±0.9 |
| | SoTTA | 50.8 ±1.1 | 51.5 ±0.9 | 42.3 ±0.9 | 66.9 ±0.5 | 46.2 ±1.0 | 64.5 ±0.3 | 67.3 ±0.3 | 60.3 ±0.0 | 59.5 ±0.5 | 63.8 ±0.1 | 70.7 ±0.6 | 68.6 ±0.9 | 55.8 ±0.4 | 62.5 ±0.4 | 54.0 ±1.1 | 59.0 ±0.4 |
| **Attack** | Source | 10.6 ±1.3 | 12.1 ±1.2 | 7.2 ±0.9 | 34.9 ±0.3 | 19.6 ±0.6 | 44.1 ±0.6 | 41.9 ±0.4 | 46.3 ±0.2 | 34.2 ±0.4 | 41.1 ±0.9 | 67.3 ±0.1 | 18.5 ±0.6 | 50.4 ±0.3 | 24.9 ±2.5 | 44.6 ±0.6 | 33.2 ±0.4 |
| | BN stats [27] | 19.0 ±0.4 | 19.5 ±0.4 | 15.6 ±0.7 | 36.2 ±0.8 | 20.3 ±0.3 | 37.7 ±0.4 | 36.2 ±0.6 | 31.2 ±0.2 | 30.1 ±0.4 | 31.2 ±0.6 | 45.8 ±0.4 | 35.2 ±0.1 | 28.0 ±0.4 | 30.8 ±0.7 | 21.8 ±0.5 | 29.2 ±0.4 |
| | PL [17] | 33.8 ±1.3 | 34.8 ±0.8 | 29.0 ±0.5 | 43.8 ±1.3 | 29.3 ±0.4 | 44.4 ±0.2 | 45.1 ±0.7 | 41.9 ±0.1 | 40.1 ±0.4 | 39.8 ±0.2 | 53.7 ±0.4 | 39.8 ±0.3 | 36.0 ±0.7 | 42.5 ±0.2 | 35.3 ±0.6 | 39.3 ±0.4 |
| | TENT [38] | 28.7 ±0.9 | 29.9 ±1.0 | 23.3 ±0.4 | 37.0 ±1.0 | 21.7 ±0.7 | 36.5 ±0.6 | 37.4 ±0.7 | 34.4 ±0.4 | 32.6 ±0.8 | 30.7 ±0.7 | 46.4 ±0.4 | 29.1 ±1.5 | 26.5 ±0.6 | 35.1 ±0.4 | 29.0 ±1.1 | 31.9 ±0.7 |
| | LAME [1] | 7.6 ±1.6 | 9.1 ±1.6 | 5.9 ±0.8 | 31.8 ±0.5 | 16.4 ±0.6 | 42.3 ±0.7 | 39.4 ±0.5 | 45.5 ±0.3 | 31.9 ±0.9 | 38.4 ±1.1 | 66.3 ±0.3 | 15.0 ±0.7 | 49.4 ±0.4 | 21.5 ±2.7 | 43.7 ±0.6 | 31.0 ±0.6 |
| | CoTTA [39] | 34.4 ±0.8 | 34.5 ±0.6 | 29.9 ±0.6 | 41.7 ±1.0 | 31.1 ±0.9 | 40.9 ±0.4 | 42.5 ±1.0 | 38.4 ±0.2 | 37.8 ±0.7 | 32.5 ±0.4 | 50.6 ±0.4 | 25.2 ±0.8 | 35.3 ±1.1 | 43.4 ±0.3 | 39.8 ±0.4 | 37.2 ±0.2 |
| | EATA [28] | 2.2 ±0.9 | 2.0 ±0.7 | 2.5 ±0.5 | 4.4 ±0.1 | 1.7 ±0.4 | 3.7 ±0.8 | 3.8 ±0.4 | 3.1 ±0.3 | 2.8 ±1.1 | 3.5 ±0.6 | 15.1 ±7.3 | 2.9 ±0.7 | 2.5 ±0.6 | 3.4 ±0.7 | 2.7 ±0.3 | 3.7 ±0.6 |
| | SAR [29] | 27.0 ±0.5 | 28.2 ±0.6 | 23.1 ±0.3 | 40.7 ±0.6 | 24.3 ±0.0 | 40.8 ±0.1 | 40.6 ±0.5 | 35.9 ±0.2 | 34.7 ±0.3 | 35.6 ±0.4 | 49.3 ±0.2 | 39.4 ±0.1 | 31.0 ±0.3 | 37.1 ±0.6 | 28.5 ±0.6 | 34.4 ±0.3 |
| | RoTTA [44] | 40.5 ±0.9 | 41.7 ±0.7 | 35.9 ±0.9 | 66.2 ±0.3 | 43.5 ±0.1 | 64.1 ±0.7 | 65.5 ±0.4 | 54.5 ±0.3 | 49.5 ±1.3 | 59.8 ±0.0 | 68.3 ±0.5 | 25.7 ±4.0 | 54.7 ±0.1 | 56.6 ±0.6 | 46.2 ±0.7 | 51.5 ±0.4 |
| | SoTTA | 54.3 ±0.7 | 55.6 ±0.7 | 47.6 ±0.2 | 69.6 ±0.2 | 51.6 ±0.5 | 67.8 ±0.1 | 69.7 ±0.2 | 62.7 ±0.3 | 61.7 ±0.3 | 66.2 ±0.1 | 72.5 ±0.1 | 68.3 ±1.2 | 59.4 ±0.5 | 65.3 ±0.4 | 56.5 ±0.6 | 61.9 ±0.0 |
| **Noise** | Source | 10.6 ±1.3 | 12.1 ±1.2 | 7.2 ±0.9 | 34.9 ±0.3 | 19.6 ±0.6 | 44.1 ±0.6 | 41.9 ±0.4 | 46.3 ±0.2 | 34.2 ±0.4 | 41.1 ±0.9 | 67.3 ±0.1 | 18.5 ±0.6 | 50.4 ±0.3 | 24.9 ±2.5 | 44.6 ±0.6 | 33.2 ±0.4 |
| | BN stats [27] | 25.5 ±1.0 | 25.5 ±0.5 | 20.8 ±0.9 | 28.2 ±0.3 | 22.0 ±0.1 | 28.4 ±0.3 | 31.0 ±0.4 | 30.7 ±0.3 | 32.6 ±0.2 | 24.5 ±0.5 | 44.2 ±0.5 | 25.8 ±0.3 | 26.3 ±0.2 | 29.4 ±0.8 | 29.4 ±0.6 | 28.3 ±0.3 |
| | PL [17] | 21.4 ±2.4 | 25.6 ±3.4 | 15.8 ±2.4 | 22.5 ±2.1 | 16.1 ±0.2 | 19.8 ±2.3 | 23.8 ±1.0 | 28.1 ±1.4 | 30.7 ±1.4 | 21.1 ±1.6 | 47.4 ±1.2 | 14.0 ±2.2 | 21.6 ±2.6 | 26.4 ±4.5 | 23.4 ±2.6 | 23.8 ±0.6 |
| | TENT [38] | 16.4 ±3.0 | 17.3 ±5.5 | 11.3 ±3.3 | 20.1 ±4.2 | 11.2 ±0.1 | 17.8 ±2.3 | 24.3 ±1.9 | 20.0 ±2.7 | 24.6 ±4.6 | 16.3 ±1.5 | 50.9 ±1.8 | 10.9 ±1.6 | 13.5 ±0.8 | 25.3 ±3.8 | 19.5 ±1.8 | 20.0 ±0.9 |
| | LAME [1] | 7.9 ±1.6 | 9.2 ±1.7 | 6.1 ±0.8 | 32.0 ±0.5 | 16.5 ±0.6 | 42.5 ±1.0 | 39.8 ±0.7 | 45.0 ±1.0 | 31.9 ±1.2 | 38.8 ±1.1 | 66.2 ±0.5 | 15.1 ±0.9 | 49.8 ±0.4 | 21.7 ±2.7 | 43.6 ±0.2 | 31.1 ±0.7 |
| | CoTTA [39] | 28.6 ±0.9 | 28.4 ±0.3 | 25.3 ±0.7 | 25.6 ±0.8 | 21.8 ±1.1 | 25.3 ±1.5 | 28.2 ±0.7 | 28.4 ±1.4 | 31.0 ±0.4 | 20.4 ±0.4 | 38.2 ±1.2 | 20.1 ±0.5 | 25.1 ±0.9 | 33.3 ±0.7 | 30.4 ±0.8 | 27.3 ±0.3 |
| | EATA [28] | 2.8 ±1.5 | 2.7 ±0.6 | 2.3 ±0.3 | 2.6 ±0.3 | 2.0 ±0.6 | 2.0 ±0.3 | 2.8 ±0.3 | 1.9 ±0.1 | 2.4 ±0.4 | 2.0 ±0.2 | 2.7 ±0.2 | 2.2 ±0.3 | 3.0 ±0.3 | 2.1 ±0.3 | 2.6 ±0.2 | 2.4 ±0.2 |
| | SAR [29] | 37.5 ±1.9 | 37.8 ±2.1 | 29.3 ±2.9 | 38.8 ±0.5 | 26.4 ±2.4 | 38.9 ±2.2 | 42.4 ±1.0 | 41.2 ±3.9 | 42.7 ±1.6 | 36.4 ±1.2 | 57.2 ±1.3 | 32.1 ±3.5 | 32.9 ±1.8 | 40.7 ±1.2 | 37.7 ±0.6 | 38.1 ±1.2 |
| | RoTTA [44] | 36.6 ±1.0 | 37.9 ±1.5 | 31.2 ±1.4 | 61.9 ±1.0 | 40.4 ±0.4 | 60.7 ±0.4 | 61.9 ±0.3 | 51.7 ±0.5 | 45.7 ±0.7 | 55.9 ±0.4 | 66.5 ±0.6 | 25.5 ±1.5 | 51.6 ±0.7 | 53.0 ±0.9 | 44.5 ±1.0 | 48.3 ±0.5 |
| | SoTTA | 50.4 ±2.5 | 52.3 ±0.7 | 41.9 ±3.5 | 66.3 ±0.8 | 45.5 ±0.7 | 65.4 ±0.5 | 66.7 ±0.6 | 60.1 ±0.9 | 59.3 ±0.2 | 63.1 ±1.0 | 70.7 ±0.3 | 65.6 ±1.1 | 55.9 ±1.4 | 61.8 ±1.9 | 54.6 ±0.6 | 58.6 ±1.0 |

Table 7: Classification accuracy (%) and their corresponding standard deviations on ImageNet-C for 15 types of corruptions under five scenarios. **Bold** numbers are the highest accuracy. Averaged over three different random seeds.

| | | Noise | | | Blur | | | | Weather | | | | Digital | | | | |
| | Method | Gau. | Shot | Imp. | Def. | Gla. | Mot. | Zoom | Snow | Fro. | Fog | Brit. | Cont. | Elas. | Pix. | JPEG | Avg. |
|---|---|---|---|---|---|---|---|---|---|---|---|---|---|---|---|---|---|
| Benign | Source | 1.2 ±0.0 | 1.8 ±0.0 | 1.0 ±0.0 | 11.4 ±0.0 | 8.7 ±0.0 | 11.2 ±0.0 | 17.6 ±0.0 | 10.9 ±0.0 | 16.5 ±0.0 | 14.3 ±0.0 | 51.3 ±0.0 | 3.4 ±0.0 | 16.8 ±0.0 | 23.1 ±0.0 | 29.6 ±0.0 | 14.6 ±0.0 |
| | BN stats [27] | 13.0 ±0.1 | 14.1 ±0.1 | 13.4 ±0.0 | 11.7 ±0.0 | 12.8 ±0.1 | 23.1 ±0.0 | 33.3 ±0.1 | 29.1 ±0.0 | 28.1 ±0.0 | 40.3 ±0.0 | 57.7 ±0.0 | 11.9 ±0.1 | 38.4 ±0.1 | 43.8 ±0.1 | 36.4 ±0.1 | 27.1 ±0.0 |
| | PL [17] | 14.9 ±1.2 | 18.3 ±1.3 | 16.5 ±0.7 | 11.2 ±0.7 | 13.2 ±1.2 | 29.1 ±1.6 | 39.1 ±0.5 | 35.5 ±1.1 | 26.0 ±1.2 | 47.6 ±0.6 | 58.3 ±0.3 | 5.2 ±0.3 | 46.5 ±0.6 | 50.5 ±0.2 | 45.6 ±0.1 | 30.5 ±0.1 |
| | TENT [38] | 13.0 ±0.1 | 14.1 ±0.1 | 13.4 ±0.0 | 11.7 ±0.0 | 12.8 ±0.1 | 23.1 ±0.0 | 33.3 ±0.1 | 29.1 ±0.0 | 28.1 ±0.0 | 40.3 ±0.0 | 57.7 ±0.0 | 11.9 ±0.1 | 38.4 ±0.1 | 43.8 ±0.1 | 36.4 ±0.1 | 27.1 ±0.0 |
| | LAME [1] | 0.7 ±0.0 | 1.1 ±0.0 | 0.5 ±0.0 | 11.4 ±0.0 | 8.6 ±0.0 | 11.1 ±0.0 | 17.5 ±0.0 | 10.3 ±0.0 | 16.4 ±0.0 | 14.1 ±0.0 | 51.3 ±0.0 | 3.4 ±0.0 | 16.5 ±0.0 | 23.0 ±0.0 | 29.6 ±0.0 | 14.4 ±0.0 |
| | CoTTA [39] | 17.7 ±0.1 | 19.0 ±0.2 | 18.0 ±0.1 | 15.7 ±0.1 | 17.4 ±0.3 | 30.6 ±0.1 | 39.0 ±0.1 | 34.0 ±0.3 | 32.4 ±0.1 | 46.9 ±0.2 | 59.3 ±0.1 | 18.7 ±0.1 | 43.1 ±0.1 | 49.8 ±0.2 | 42.2 ±0.1 | 32.2 ±0.1 |
| | EATA [28] | 25.9 ±0.2 | 27.5 ±0.1 | 25.9 ±0.2 | 23.7 ±0.4 | 23.9 ±0.3 | 35.2 ±0.2 | 43.2 ±0.2 | 40.2 ±0.2 | 36.2 ±0.1 | 50.3 ±0.2 | 59.9 ±0.1 | **30.6** ±0.3 | 48.4 ±0.0 | 51.8 ±0.1 | 47.0 ±0.0 | 38.0 ±0.1 |
| | SAR [29] | 24.5 ±0.3 | 26.4 ±0.1 | 24.5 ±0.2 | 21.0 ±0.2 | 21.6 ±0.0 | 33.3 ±0.1 | 41.1 ±0.1 | 38.3 ±0.1 | 34.6 ±0.1 | 49.2 ±0.1 | 59.4 ±0.1 | 24.8 ±1.0 | 46.5 ±0.1 | 50.7 ±0.2 | 45.8 ±0.0 | 36.1 ±0.1 |
| | RoTTA [44] | 15.1 ±0.1 | 16.5 ±0.2 | 15.5 ±0.1 | 13.1 ±0.2 | 14.2 ±0.0 | 25.5 ±0.1 | 36.1 ±0.1 | 31.8 ±0.2 | 28.9 ±0.1 | 44.2 ±0.2 | 59.5 ±0.0 | 15.6 ±0.2 | 41.6 ±0.1 | 46.8 ±0.0 | 40.3 ±0.1 | 29.7 ±0.0 |
| | **SoTTA** | **29.2** ±0.2 | **31.8** ±0.3 | **29.8** ±0.3 | **26.2** ±0.2 | **27.6** ±0.2 | **37.9** ±0.2 | **44.7** ±0.2 | **42.8** ±0.2 | **37.9** ±0.3 | **52.3** ±0.2 | **60.1** ±0.1 | 24.1 ±0.5 | **50.3** ±0.1 | **53.4** ±0.1 | **48.7** ±0.2 | **39.8** ±0.0 |
| Near | Source | 1.2 ±0.0 | 1.8 ±0.0 | 1.0 ±0.0 | 11.4 ±0.0 | 8.7 ±0.0 | 11.2 ±0.0 | 17.6 ±0.0 | 10.9 ±0.0 | 16.5 ±0.0 | 14.3 ±0.0 | 51.3 ±0.0 | 3.4 ±0.0 | 16.8 ±0.0 | 23.1 ±0.0 | 29.6 ±0.0 | 14.6 ±0.0 |
| | BN stats [27] | 5.8 ±0.5 | 6.9 ±0.1 | 6.7 ±0.0 | 8.5 ±0.7 | 8.5 ±0.3 | 15.5 ±0.0 | 24.6 ±0.1 | 19.1 ±0.4 | 21.5 ±0.1 | 26.8 ±0.0 | 49.7 ±0.1 | 4.5 ±0.2 | 26.2 ±0.1 | 31.7 ±0.0 | 27.3 ±0.2 | 18.9 ±0.1 |
| | PL [17] | 0.5 ±0.1 | 0.7 ±0.1 | 0.6 ±0.0 | 2.2 ±0.1 | 1.6 ±0.3 | 4.4 ±0.2 | 7.2 ±0.5 | 3.8 ±0.4 | 3.6 ±0.2 | 11.4 ±0.8 | 35.6 ±1.3 | 0.6 ±0.0 | 6.5 ±1.3 | 18.7 ±0.6 | 5.7 ±0.2 | 6.9 ±0.0 |
| | TENT [38] | 5.8 ±0.5 | 6.9 ±0.1 | 6.7 ±0.0 | 8.5 ±0.7 | 8.5 ±0.3 | 15.5 ±0.0 | 24.6 ±0.1 | 19.1 ±0.4 | 21.5 ±0.1 | 26.8 ±0.0 | 49.7 ±0.1 | 4.5 ±0.2 | 26.2 ±0.1 | 31.7 ±0.0 | 27.3 ±0.2 | 18.9 ±0.1 |
| | LAME [1] | 1.0 ±0.1 | 1.5 ±0.0 | 0.8 ±0.0 | 10.8 ±1.0 | 8.4 ±0.3 | 11.1 ±0.0 | 17.5 ±0.0 | 10.3 ±0.2 | 16.4 ±0.0 | 14.1 ±0.0 | 51.3 ±0.0 | 3.4 ±0.1 | 16.5 ±0.0 | 23.0 ±0.0 | 29.6 ±0.0 | 14.4 ±0.1 |
| | CoTTA [39] | 6.6 ±0.9 | 7.8 ±0.2 | 7.4 ±0.1 | 11.6 ±1.7 | 11.1 ±0.2 | 23.1 ±0.1 | 30.5 ±0.2 | 24.6 ±0.4 | 24.8 ±0.0 | 36.0 ±0.1 | 53.7 ±0.3 | 5.8 ±1.1 | 31.8 ±0.1 | 41.4 ±0.1 | 34.0 ±0.3 | 23.3 ±0.2 |
| | EATA [28] | 6.6 ±0.1 | 9.1 ±0.1 | 7.7 ±0.2 | 14.1 ±0.0 | 12.9 ±0.1 | 23.3 ±0.2 | 33.5 ±0.1 | 29.0 ±0.0 | 28.9 ±0.2 | 40.1 ±0.1 | 55.4 ±0.1 | **6.4** ±0.8 | 36.9 ±0.3 | 43.7 ±0.3 | 36.5 ±0.1 | 25.6 ±0.1 |
| | SAR [29] | **6.7** ±1.3 | **10.2** ±0.6 | **8.1** ±0.6 | 15.9 ±1.9 | 13.5 ±0.7 | **28.1** ±0.1 | 37.0 ±0.1 | 32.9 ±1.3 | 28.2 ±0.5 | 44.6 ±0.2 | 56.8 ±0.0 | 1.8 ±0.5 | 40.8 ±0.4 | 47.7 ±0.1 | 41.6 ±0.3 | 27.6 ±0.3 |
| | RoTTA [44] | 3.1 ±0.7 | 5.4 ±1.0 | 3.7 ±0.4 | 10.8 ±1.0 | 9.2 ±0.6 | 23.0 ±0.4 | 33.5 ±1.3 | 32.3 ±0.8 | **30.3** ±0.2 | 44.6 ±0.0 | **59.3** ±0.1 | 0.7 ±0.1 | 40.2 ±0.0 | 46.3 ±0.3 | 40.2 ±0.2 | 25.6 ±0.4 |
| | **SoTTA** | 0.3 ±0.1 | 5.8 ±4.0 | 0.5 ±0.2 | **21.7** ±0.8 | **21.5** ±0.8 | 26.9 ±4.9 | **39.0** ±0.6 | **35.0** ±0.5 | 28.1 ±0.5 | **46.5** ±0.2 | 56.1 ±0.2 | 0.5 ±0.1 | **44.8** ±0.1 | **49.2** ±0.3 | **43.2** ±0.1 | **27.9** ±0.3 |
| Far | Source | 1.2 ±0.0 | 1.8 ±0.0 | 1.0 ±0.0 | 11.4 ±0.0 | 8.7 ±0.0 | 11.2 ±0.0 | 17.6 ±0.0 | 10.9 ±0.0 | 16.5 ±0.0 | 14.3 ±0.0 | 51.3 ±0.0 | 3.4 ±0.0 | 16.8 ±0.0 | 23.1 ±0.0 | 29.6 ±0.0 | 14.6 ±0.0 |
| | BN stats [27] | 4.5 ±0.0 | 5.0 ±0.1 | 5.0 ±0.0 | 4.2 ±0.0 | 5.3 ±0.1 | 9.1 ±0.1 | 16.2 ±0.1 | 17.4 ±0.1 | 18.0 ±0.1 | 22.2 ±0.0 | 43.3 ±0.1 | 1.2 ±0.0 | 23.2 ±0.1 | 26.9 ±0.0 | 20.9 ±0.1 | 14.8 ±0.0 |
| | PL [17] | 0.4 ±0.0 | 0.5 ±0.1 | 0.5 ±0.0 | 0.7 ±0.1 | 1.0 ±0.2 | 1.4 ±0.1 | 3.4 ±0.4 | 2.2 ±0.2 | 2.8 ±0.1 | 6.6 ±2.1 | 34.7 ±5.1 | 0.2 ±0.0 | 6.3 ±0.5 | 12.2 ±1.4 | 4.3 ±0.3 | 5.1 ±0.2 |
| | TENT [38] | 4.5 ±0.0 | 5.0 ±0.1 | 5.0 ±0.0 | 4.2 ±0.0 | 5.3 ±0.0 | 9.2 ±0.1 | 16.2 ±0.1 | 17.4 ±0.1 | 18.0 ±0.0 | 22.2 ±0.1 | 43.3 ±0.1 | 1.2 ±0.0 | 23.2 ±0.1 | 26.9 ±0.1 | 20.9 ±0.1 | 14.8 ±0.0 |
| | LAME [1] | 0.7 ±0.0 | 1.1 ±0.0 | 0.5 ±0.0 | 11.4 ±0.0 | 8.6 ±0.0 | 11.1 ±0.0 | 17.5 ±0.0 | 10.3 ±0.0 | 16.3 ±0.0 | 14.0 ±0.0 | 51.3 ±0.0 | 3.3 ±0.0 | 16.5 ±0.0 | 23.0 ±0.0 | 29.6 ±0.0 | 14.4 ±0.0 |
| | CoTTA [39] | 4.4 ±0.5 | 5.1 ±0.7 | 4.5 ±0.2 | 4.2 ±0.3 | 6.2 ±0.2 | 10.7 ±0.8 | 18.9 ±0.7 | 21.6 ±1.0 | 20.0 ±0.8 | 30.0 ±0.2 | 48.2 ±0.3 | 0.9 ±0.4 | 27.9 ±0.9 | 34.8 ±0.7 | 27.0 ±0.2 | 17.6 ±0.2 |
| | EATA [28] | 7.9 ±0.6 | 10.1 ±0.5 | 10.1 ±0.4 | 8.9 ±0.5 | 10.2 ±0.2 | 17.9 ±0.4 | 28.8 ±0.3 | 27.1 ±0.3 | 26.8 ±0.2 | 38.2 ±0.1 | 52.2 ±0.0 | 0.8 ±0.1 | 34.3 ±0.2 | 40.2 ±0.2 | 32.6 ±0.1 | 23.1 ±0.1 |
| | SAR [29] | 3.2 ±0.5 | 4.9 ±1.2 | 3.7 ±1.2 | 3.5 ±1.7 | 5.5 ±1.7 | 20.6 ±1.5 | 32.1 ±0.9 | 31.8 ±0.2 | 26.1 ±0.9 | 43.1 ±0.1 | 54.0 ±0.1 | 0.4 ±0.1 | 39.2 ±0.5 | 45.4 ±0.2 | 39.1 ±0.1 | 23.5 ±0.4 |
| | RoTTA [44] | 13.9 ±0.1 | 15.5 ±0.1 | 16.2 ±0.1 | 12.1 ±0.1 | 12.8 ±0.1 | 25.0 ±0.1 | 35.8 ±0.1 | 33.6 ±0.2 | 29.5 ±0.1 | 45.8 ±0.2 | **59.4** ±0.2 | **8.0** ±0.2 | 41.9 ±0.1 | 47.0 ±0.1 | 41.0 ±0.3 | 29.2 ±0.2 |
| | **SoTTA** | **26.9** ±0.4 | **29.5** ±0.4 | **27.3** ±0.1 | **22.3** ±0.2 | **23.6** ±0.6 | **35.8** ±0.4 | **42.2** ±0.1 | **40.8** ±0.6 | **35.5** ±0.1 | **50.7** ±0.4 | 58.4 ±0.1 | 1.6 ±1.1 | **48.3** ±0.3 | **52.2** ±0.2 | **46.8** ±0.2 | **36.1** ±0.1 |
| Attack | Source | 1.2 ±0.0 | 1.8 ±0.0 | 1.0 ±0.0 | 11.4 ±0.0 | 8.7 ±0.0 | 11.2 ±0.0 | 17.6 ±0.0 | 10.9 ±0.0 | 16.5 ±0.0 | 14.3 ±0.0 | 51.3 ±0.0 | 3.4 ±0.0 | 16.8 ±0.0 | 23.1 ±0.0 | 29.6 ±0.0 | 14.6 ±0.0 |
| | BN stats [27] | 6.4 ±0.0 | 7.7 ±0.1 | 6.7 ±0.0 | 7.2 ±0.1 | 7.3 ±0.1 | 12.5 ±0.9 | 20.0 ±0.1 | 19.2 ±0.2 | 17.6 ±0.1 | 24.8 ±0.1 | 46.1 ±0.1 | 10.5 ±0.1 | 25.1 ±0.1 | 24.5 ±10.3 | 24.9 ±0.1 | 17.4 ±0.8 |
| | PL [17] | 6.5 ±0.7 | 7.8 ±0.5 | 6.3 ±0.4 | 4.0 ±0.2 | 4.4 ±0.3 | 10.5 ±0.6 | 22.2 ±2.7 | 20.0 ±1.7 | 9.7 ±0.7 | 28.8 ±0.4 | 48.6 ±0.4 | 1.9 ±0.2 | 34.5 ±2.8 | 29.8 ±18.9 | 36.0 ±0.6 | 18.1 ±1.3 |
| | TENT [38] | 6.4 ±0.0 | 7.7 ±0.1 | 6.7 ±0.0 | 7.2 ±0.1 | 7.3 ±0.1 | 12.5 ±0.9 | 20.0 ±0.1 | 19.2 ±0.2 | 17.6 ±0.1 | 24.8 ±0.1 | 46.1 ±0.1 | 10.5 ±0.1 | 25.1 ±0.1 | 24.5 ±10.3 | 25.0 ±0.1 | 17.4 ±0.8 |
| | LAME [1] | 0.7 ±0.0 | 1.1 ±0.0 | 0.5 ±0.0 | 11.4 ±0.0 | 8.6 ±0.0 | 10.7 ±0.6 | 17.5 ±0.0 | 10.3 ±0.0 | 16.4 ±0.0 | 14.1 ±0.0 | 51.3 ±0.0 | 3.4 ±0.0 | 16.5 ±0.0 | 18.5 ±7.7 | 29.6 ±0.0 | 14.0 ±0.6 |
| | CoTTA [39] | 17.2 ±0.2 | 18.6 ±0.2 | 17.3 ±0.2 | 14.5 ±0.1 | 15.0 ±0.1 | 27.0 ±2.4 | 32.7 ±0.1 | 31.9 ±0.2 | 28.3 ±0.2 | 40.2 ±0.2 | 52.0 ±0.1 | 18.8 ±0.3 | 38.6 ±0.2 | 34.3 ±16.2 | 38.7 ±0.1 | 28.3 ±1.3 |
| | EATA [28] | 15.3 ±0.1 | 17.5 ±0.2 | 15.5 ±0.2 | 13.9 ±0.2 | 12.9 ±0.2 | 22.0 ±0.1 | 28.8 ±0.2 | 29.0 ±0.4 | 24.3 ±0.1 | 35.5 ±0.1 | 49.2 ±0.1 | 17.3 ±0.8 | 35.7 ±0.2 | 39.1 ±0.1 | 36.0 ±0.1 | 26.1 ±0.1 |
| | SAR [29] | 19.1 ±0.1 | 21.2 ±0.0 | 18.8 ±0.1 | 15.6 ±0.2 | 15.1 ±0.2 | 22.5 ±2.1 | 29.1 ±0.1 | 29.5 ±0.3 | 25.2 ±0.0 | 35.8 ±0.1 | 49.0 ±0.1 | 17.3 ±1.2 | 35.8 ±0.1 | 31.4 ±14.0 | 36.7 ±0.0 | 26.8 ±1.0 |
| | RoTTA [44] | 19.0 ±0.2 | 19.7 ±0.3 | 19.1 ±0.1 | 16.8 ±0.1 | 17.1 ±0.2 | 28.9 ±1.8 | 38.4 ±0.1 | 35.7 ±0.2 | 31.1 ±0.2 | 47.0 ±0.1 | 59.8 ±0.1 | 22.0 ±0.0 | 43.5 ±0.1 | 39.1 ±16.7 | 43.3 ±0.1 | 32.0 ±1.2 |
| | **SoTTA** | **30.9** ±0.3 | **33.5** ±0.2 | **31.7** ±0.2 | **28.3** ±0.3 | **29.4** ±0.0 | **40.0** ±0.3 | **45.4** ±0.2 | **44.2** ±0.2 | **38.9** ±0.2 | **53.1** ±0.1 | **60.5** ±0.0 | **25.4** ±1.4 | **51.3** ±0.2 | **54.5** ±0.2 | **49.5** ±0.1 | **41.1** ±0.1 |
| Noise | Source | 1.2 ±0.0 | 1.8 ±0.0 | 1.0 ±0.0 | 11.4 ±0.0 | 8.7 ±0.0 | 11.2 ±0.0 | 17.6 ±0.0 | 10.9 ±0.0 | 16.5 ±0.0 | 14.3 ±0.0 | 51.3 ±0.0 | 3.4 ±0.0 | 16.8 ±0.0 | 23.1 ±0.0 | 29.6 ±0.0 | 14.6 ±0.0 |
| | BN stats [27] | 7.0 ±0.1 | 7.5 ±0.1 | 7.4 ±0.0 | 5.1 ±0.1 | 6.0 ±0.0 | 7.5 ±0.1 | 11.9 ±0.1 | 12.4 ±0.2 | 11.4 ±0.1 | 10.7 ±0.1 | 34.5 ±0.1 | 4.4 ±0.1 | 16.3 ±0.1 | 23.9 ±0.1 | 25.7 ±0.1 | 12.8 ±0.0 |
| | PL [17] | 0.5 ±0.1 | 0.9 ±0.4 | 1.1 ±0.3 | 0.6 ±0.1 | 0.6 ±0.1 | 0.7 ±0.1 | 1.5 ±0.2 | 1.6 ±0.2 | 1.4 ±0.4 | 1.1 ±0.1 | 19.0 ±5.7 | 0.5 ±0.1 | 2.8 ±1.4 | 11.2 ±2.6 | 7.7 ±1.9 | 3.4 ±0.6 |
| | TENT [38] | 7.0 ±0.1 | 7.5 ±0.1 | 7.4 ±0.0 | 5.1 ±0.1 | 6.0 ±0.0 | 7.6 ±0.1 | 11.8 ±0.1 | 12.3 ±0.1 | 11.3 ±0.1 | 10.6 ±0.1 | 34.5 ±0.1 | 4.5 ±0.1 | 16.3 ±0.1 | 23.9 ±0.1 | 25.6 ±0.1 | 12.8 ±0.0 |
| | LAME [1] | 0.7 ±0.0 | 1.1 ±0.0 | 0.5 ±0.0 | 11.4 ±0.0 | 8.5 ±0.0 | 11.1 ±0.0 | 17.5 ±0.0 | 10.3 ±0.0 | 16.4 ±0.0 | 14.0 ±0.0 | 51.3 ±0.0 | 3.4 ±0.0 | 16.5 ±0.0 | 23.0 ±0.0 | 29.6 ±0.0 | 14.3 ±0.0 |
| | CoTTA [39] | 8.3 ±0.1 | 9.0 ±0.8 | 9.2 ±0.0 | 4.3 ±0.4 | 5.4 ±0.3 | 7.8 ±0.2 | 13.3 ±0.8 | 16.6 ±0.1 | 13.7 ±0.1 | 14.9 ±0.2 | 44.0 ±0.0 | 3.2 ±0.1 | 19.4 ±0.1 | 30.9 ±0.4 | 32.3 ±0.9 | 16.0 ±0.9 |
| | EATA [28] | 14.0 ±0.1 | 14.8 ±0.2 | 14.3 ±0.4 | 8.5 ±0.3 | 9.0 ±0.4 | 12.4 ±0.3 | 21.1 ±0.4 | 22.0 ±0.5 | 19.5 ±0.2 | 24.6 ±0.2 | 46.6 ±0.2 | 2.1 ±0.4 | 28.0 ±0.3 | 37.3 ±0.3 | 36.5 ±0.2 | 20.7 ±0.0 |
| | SAR [29] | 13.9 ±4.0 | 18.8 ±1.5 | 16.3 ±0.5 | 5.1 ±1.2 | 3.4 ±2.1 | 6.6 ±0.4 | 25.0 ±0.5 | 27.6 ±1.7 | 19.8 ±0.2 | 33.9 ±0.1 | 49.8 ±0.1 | 1.0 ±0.1 | 29.0 ±1.4 | 41.2 ±0.1 | 38.3 ±0.4 | 22.0 ±0.4 |
| | RoTTA [44] | 16.2 ±0.2 | 17.2 ±0.4 | 16.5 ±0.2 | 17.7 ±0.1 | 16.6 ±0.1 | 26.8 ±0.2 | 37.3 ±0.1 | 33.4 ±0.1 | 29.3 ±0.1 | 45.0 ±0.1 | **59.4** ±0.1 | 21.0 ±0.1 | 42.4 ±0.3 | 48.2 ±0.3 | 41.3 ±1.5 | 31.2 ±0.2 |
| | **SoTTA** | **27.5** ±0.6 | **30.4** ±0.2 | **28.3** ±0.1 | **26.5** ±0.3 | **27.3** ±0.4 | **37.7** ±0.1 | **43.6** ±0.1 | **42.4** ±0.2 | **36.9** ±0.6 | **51.6** ±0.1 | 59.2 ±0.1 | **23.8** ±0.3 | **49.6** ±0.3 | **53.0** ±0.1 | **48.2** ±0.1 | **39.0** ±0.1 |

Table 8: Classification accuracy (%) and their corresponding standard deviations on ablation study of individual components on CIFAR10-C for 15 types of corruptions under five scenarios. **Bold** numbers are the highest accuracy. Averaged over three different random seeds.

| | Method | Noise | | | Blur | | | | Weather | | | | Digital | | | | Avg. |
|---|---|---|---|---|---|---|---|---|---|---|---|---|---|---|---|---|---|
| | | Gau. | Shot | Imp. | Def. | Gla. | Mot. | Zoom | Snow | Fro. | Fog | Brit. | Cont. | Elas. | Pix. | JPEG | |
| **Benign** | Source | 26.0 ±3.3 | 33.2 ±3.5 | 24.7 ±4.2 | 56.7 ±2.7 | 52.0 ±2.7 | 67.4 ±1.2 | 64.8 ±2.6 | 78.0 ±0.4 | 67.0 ±2.5 | 74.1 ±0.8 | 91.5 ±0.3 | 33.9 ±1.8 | 76.6 ±0.7 | 46.4 ±0.6 | 73.2 ±0.8 | 57.7 ±1.0 |
| | HC | 10.2 ±0.1 | 10.9 ±0.9 | 10.7 ±0.7 | 61.9 ±31.5 | 14.2 ±0.4 | 41.8 ±21.6 | 82.0 ±2.0 | 32.1 ±11.6 | 36.3 ±17.7 | 52.3 ±28.7 | 88.7 ±1.1 | 11.2 ±1.3 | 34.0 ±15.8 | 14.2 ±3.9 | 23.5 ±3.8 | 34.9 ±4.8 |
| | UC | 26.0 ±19.5 | 42.0 ±27.4 | 10.5 ±0.5 | 82.4 ±0.8 | 59.0 ±1.2 | 79.3 ±1.8 | 82.5 ±1.4 | 77.5 ±1.2 | 76.5 ±1.5 | 81.5 ±0.6 | 86.3 ±1.5 | 73.2 ±6.0 | 73.3 ±1.1 | 74.9 ±1.6 | 71.4 ±1.4 | 66.4 ±3.0 |
| | HC + UC (HUS) | 20.7 ±15.1 | 57.8 ±4.7 | 21.1 ±9.5 | 84.6 ±0.9 | 62.4 ±1.8 | 83.4 ±0.5 | 85.2 ±1.1 | 80.5 ±1.4 | 79.3 ±0.5 | 84.5 ±1.0 | 89.8 ±0.3 | 69.5 ±11.5 | 76.4 ±1.3 | 76.3 ±2.2 | 75.1 ±0.4 | 69.8 ±1.1 |
| | ESM | **76.0** ±0.6 | **78.3** ±0.4 | **69.3** ±0.5 | **89.0** ±0.2 | 69.7 ±1.7 | **87.9** ±0.4 | **89.6** ±0.2 | 85.3 ±0.4 | 84.1 ±0.7 | 87.7 ±0.3 | 92.1 ±0.2 | **88.1** ±0.8 | 79.6 ±1.0 | 84.2 ±0.4 | **78.5** ±0.3 | **82.6** ±0.2 |
| | HC + ESM | 74.9 ±0.8 | 78.1 ±0.0 | 69.0 ±0.6 | 88.8 ±0.3 | 70.9 ±0.6 | 87.7 ±0.6 | 89.2 ±0.1 | **85.7** ±0.5 | **84.4** ±0.1 | 87.8 ±0.1 | **92.2** ±0.3 | 84.0 ±0.6 | 79.7 ±0.7 | 83.7 ±0.8 | 78.1 ±0.4 | 82.3 ±0.2 |
| | UC + ESM | 74.9 ±0.5 | 77.1 ±1.1 | 68.2 ±0.5 | 88.7 ±0.4 | **71.0** ±1.5 | 87.4 ±0.3 | 89.1 ±1.0 | 85.0 ±0.5 | 84.0 ±0.3 | 87.8 ±0.9 | 92.0 ±0.3 | 86.2 ±2.4 | **79.8** ±1.0 | **84.4** ±0.6 | 78.0 ±0.8 | 82.2 ±0.2 |
| | HUS + ESM (SoTTA) | 75.0 ±1.1 | 77.5 ±0.6 | 68.8 ±0.7 | 88.8 ±0.4 | 70.7 ±1.2 | 87.5 ±0.5 | 89.0 ±0.5 | 85.4 ±0.3 | 84.0 ±0.7 | **88.2** ±0.2 | 91.9 ±0.1 | 83.9 ±1.5 | **79.8** ±0.4 | 83.9 ±0.5 | 78.3 ±0.7 | 82.2 ±0.3 |
| **Near** | Source | 26.0 ±3.3 | 33.2 ±3.5 | 24.7 ±4.2 | 56.7 ±2.7 | 52.0 ±2.7 | 67.4 ±1.2 | 64.8 ±2.6 | 78.0 ±0.4 | 67.0 ±2.5 | 74.1 ±0.8 | **91.5** ±0.3 | 33.9 ±1.8 | 76.6 ±0.7 | 46.4 ±0.6 | 73.2 ±0.8 | 57.7 ±1.0 |
| | HC | 10.3 ±0.1 | 10.6 ±0.3 | 10.1 ±0.0 | 12.6 ±2.9 | 12.3 ±2.5 | 11.9 ±0.5 | 16.7 ±4.8 | 16.1 ±1.9 | 14.4 ±1.4 | 14.5 ±2.4 | 16.6 ±0.6 | 11.2 ±1.5 | 19.7 ±3.6 | 13.2 ±1.2 | 14.0 ±0.7 | 13.6 ±0.3 |
| | UC | 42.6 ±6.6 | 48.3 ±2.4 | 20.8 ±4.5 | 73.3 ±2.2 | 50.4 ±3.9 | 73.1 ±1.8 | 74.9 ±3.3 | 71.0 ±2.2 | 67.5 ±1.9 | 74.7 ±4.7 | 82.4 ±0.9 | 59.7 ±11.1 | 64.3 ±0.8 | 64.9 ±5.6 | 64.2 ±1.9 | 62.1 ±0.8 |
| | HC + UC (HUS) | 32.8 ±2.5 | 42.5 ±3.5 | 15.9 ±7.3 | 73.8 ±2.5 | 51.1 ±2.1 | 73.8 ±1.9 | 77.8 ±2.8 | 65.1 ±13.4 | 70.8 ±1.1 | 77.6 ±0.5 | 84.6 ±0.8 | 61.6 ±2.7 | 69.7 ±1.2 | 62.6 ±4.7 | 66.2 ±3.2 | 61.7 ±1.3 |
| | ESM | 67.9 ±1.4 | 69.7 ±0.5 | 58.5 ±0.6 | 84.6 ±1.4 | 63.0 ±3.5 | 83.9 ±0.8 | 86.7 ±0.1 | 82.5 ±0.4 | 81.3 ±0.5 | 85.2 ±0.6 | 90.6 ±0.4 | 83.3 ±2.5 | 77.9 ±1.9 | 76.7 ±1.5 | 76.0 ±1.5 | 77.9 ±0.4 |
| | HC + ESM | 73.6 ±1.4 | 75.6 ±1.7 | 64.3 ±3.9 | 87.3 ±1.0 | 66.7 ±0.6 | 86.3 ±0.7 | 87.6 ±0.5 | **84.8** ±0.2 | 83.2 ±0.3 | 86.9 ±0.9 | 90.9 ±0.7 | 87.3 ±0.5 | **79.1** ±0.9 | 82.3 ±0.8 | 77.3 ±1.3 | 80.9 ±0.6 |
| | UC + ESM | 68.1 ±2.3 | 69.4 ±0.9 | 60.3 ±2.9 | 84.8 ±0.6 | 64.6 ±2.3 | 84.2 ±1.1 | 85.5 ±0.2 | 82.2 ±0.6 | 81.7 ±0.7 | 85.3 ±0.4 | 90.8 ±0.4 | 84.4 ±1.0 | 77.0 ±1.2 | 75.5 ±0.1 | 76.5 ±0.9 | 78.0 ±0.4 |
| | HUS + ESM (SoTTA) | 74.3 ±1.4 | 76.7 ±0.9 | 66.5 ±2.2 | 87.5 ±0.1 | 66.9 ±0.8 | 86.4 ±0.6 | 87.8 ±0.5 | 84.4 ±0.6 | 83.8 ±0.2 | 87.2 ±0.5 | 91.3 ±0.2 | 88.4 ±0.7 | 78.7 ±1.1 | 82.4 ±0.5 | 78.0 ±0.6 | 81.4 ±0.5 |
| **Far** | Source | 26.0 ±3.3 | 33.2 ±3.5 | 24.7 ±4.2 | 56.7 ±2.7 | 52.0 ±2.7 | 67.4 ±1.2 | 64.8 ±2.6 | 78.0 ±0.4 | 67.0 ±2.5 | 74.1 ±0.8 | 91.5 ±0.3 | 33.9 ±1.8 | 76.6 ±0.7 | 46.4 ±0.6 | 73.2 ±0.8 | 57.7 ±1.0 |
| | HC | 10.3 ±0.1 | 10.7 ±0.8 | 10.1 ±0.0 | 15.4 ±7.3 | 12.1 ±0.7 | 17.4 ±6.7 | 26.7 ±22.2 | 16.2 ±3.1 | 16.6 ±2.6 | 17.9 ±7.7 | 55.5 ±28.4 | 11.0 ±0.7 | 15.3 ±1.6 | 12.9 ±1.6 | 15.8 ±4.7 | 17.6 ±3.8 |
| | UC | 49.3 ±4.9 | 43.5 ±6.2 | 10.2 ±0.3 | 66.8 ±4.4 | 46.7 ±1.8 | 65.3 ±3.3 | 69.4 ±3.8 | 67.5 ±0.5 | 62.0 ±3.1 | 65.8 ±3.8 | 74.3 ±0.9 | 55.4 ±11.1 | 55.8 ±5.4 | 59.2 ±1.0 | 55.8 ±4.0 | 56.5 ±2.0 |
| | HC + UC (HUS) | 18.5 ±7.1 | 26.2 ±15.7 | 11.9 ±3.3 | 70.0 ±3.0 | 49.2 ±6.2 | 72.0 ±2.9 | 77.7 ±2.9 | 72.8 ±1.0 | 70.2 ±0.3 | 75.2 ±0.6 | 84.2 ±1.0 | 53.9 ±9.4 | 62.9 ±4.8 | 65.2 ±2.3 | 66.3 ±2.8 | 58.4 ±0.5 |
| | ESM | 59.2 ±2.2 | 62.0 ±1.8 | 49.7 ±4.0 | 84.3 ±2.1 | 52.2 ±2.3 | 83.3 ±0.9 | 85.2 ±0.8 | 78.3 ±0.6 | 75.0 ±0.6 | 85.2 ±0.7 | 88.5 ±0.3 | 78.8 ±5.5 | 71.9 ±2.0 | 71.8 ±2.9 | 66.6 ±2.3 | 72.8 ±0.7 |
| | HC + ESM | 60.6 ±3.2 | 64.7 ±3.4 | 55.2 ±7.7 | 84.8 ±0.3 | 55.7 ±8.8 | 82.1 ±1.3 | 83.9 ±1.7 | 81.8 ±1.0 | 82.2 ±1.7 | 83.6 ±2.6 | 90.3 ±0.9 | 82.5 ±0.6 | 70.4 ±5.5 | 76.6 ±2.4 | 69.0 ±6.1 | 74.9 ±2.4 |
| | UC + ESM | 64.0 ±1.9 | 67.7 ±4.6 | 57.3 ±2.9 | 84.2 ±1.1 | 56.5 ±3.5 | 84.5 ±0.8 | 85.9 ±0.7 | 78.7 ±1.6 | 79.0 ±1.1 | 85.1 ±0.8 | 90.8 ±0.4 | 84.9 ±0.9 | 73.5 ±1.0 | 76.8 ±1.1 | 69.3 ±1.9 | 75.9 ±0.5 |
| | HUS + ESM (SoTTA) | 73.3 ±1.2 | 76.3 ±1.9 | 66.3 ±2.5 | 88.5 ±0.6 | 68.3 ±2.3 | 86.8 ±0.7 | 88.3 ±0.2 | 84.1 ±1.0 | 84.2 ±0.6 | 87.2 ±0.4 | 92.0 ±0.2 | 89.0 ±1.1 | 77.8 ±1.8 | 83.8 ±0.9 | 77.8 ±1.2 | 81.6 ±0.6 |
| **Attack** | Source | 26.0 ±3.3 | 33.2 ±3.5 | 24.7 ±4.2 | 56.7 ±2.7 | 52.0 ±2.7 | 67.4 ±1.2 | 64.8 ±2.6 | 78.0 ±0.4 | 67.0 ±2.5 | 74.1 ±0.8 | 91.5 ±0.3 | 33.9 ±1.8 | 76.6 ±0.7 | 46.4 ±0.6 | 73.2 ±0.8 | 57.7 ±1.0 |
| | HC | 10.2 ±0.0 | 10.4 ±0.3 | 10.1 ±0.0 | 15.1 ±7.0 | 11.7 ±0.2 | 21.5 ±4.4 | 31.6 ±11.7 | 17.1 ±3.5 | 13.1 ±0.5 | 22.0 ±6.7 | 32.6 ±10.5 | 10.5 ±0.3 | 18.5 ±4.5 | 11.9 ±0.5 | 17.2 ±0.8 | 16.9 ±1.6 |
| | UC | 44.1 ±29.5 | 47.3 ±32.3 | 10.1 ±0.2 | 82.9 ±1.4 | 63.1 ±2.4 | 81.7 ±1.5 | 84.8 ±1.2 | 79.1 ±1.8 | 79.0 ±0.9 | 83.7 ±1.0 | 88.3 ±0.7 | 81.0 ±0.8 | 73.2 ±1.8 | 78.5 ±3.8 | 73.3 ±1.5 | 70.0 ±3.9 |
| | HC + UC (HUS) | 16.1 ±7.5 | 26.3 ±23.9 | 10.3 ±0.4 | 35.4 ±19.3 | 42.1 ±14.5 | 47.9 ±14.7 | 54.4 ±29.7 | 46.4 ±14.4 | 45.3 ±31.9 | 55.0 ±24.3 | 49.8 ±3.2 | 47.5 ±27.0 | 42.6 ±22.5 | 57.1 ±11.6 | 36.5 ±14.9 | 40.9 ±5.5 |
| | ESM | 77.0 ±0.6 | 79.5 ±0.3 | 71.6 ±0.4 | 89.0 ±0.6 | 71.8 ±0.6 | 88.3 ±0.3 | 89.6 ±0.6 | 86.3 ±0.7 | 85.5 ±0.6 | 88.1 ±0.6 | 92.1 ±0.2 | 87.2 ±1.3 | 80.4 ±0.8 | 85.4 ±0.5 | 79.8 ±0.9 | 83.4 ±0.2 |
| | HC + ESM | 77.8 ±0.3 | 79.7 ±0.3 | 70.9 ±0.2 | 89.3 ±0.3 | 71.8 ±1.4 | 87.9 ±0.4 | 89.6 ±0.4 | 86.1 ±0.5 | 85.4 ±0.2 | 88.7 ±0.5 | 92.2 ±0.2 | 87.3 ±0.5 | 80.4 ±0.3 | 85.7 ±0.6 | 79.8 ±0.2 | 83.5 ±0.2 |
| | UC + ESM | 78.2 ±0.2 | 80.1 ±0.4 | 72.3 ±0.9 | 89.9 ±0.2 | **73.6** ±0.8 | 89.1 ±0.1 | 90.2 ±0.2 | 86.7 ±0.2 | 85.7 ±0.2 | **89.3** ±0.3 | 92.8 ±0.1 | **88.6** ±0.2 | 81.0 ±0.8 | 86.0 ±0.4 | 80.5 ±0.5 | 84.3 ±0.1 |
| | HUS + ESM (SoTTA) | 78.2 ±0.3 | 80.8 ±0.1 | 72.3 ±0.8 | 90.1 ±0.2 | **73.6** ±0.9 | 89.2 ±0.4 | 90.3 ±0.5 | 87.4 ±0.5 | 86.2 ±0.6 | 89.3 ±0.6 | 92.9 ±0.1 | 87.8 ±0.7 | 81.3 ±0.8 | 86.6 ±0.3 | 81.0 ±0.3 | 84.5 ±0.2 |
| **Noise** | Source | 26.0 ±3.3 | 33.2 ±3.5 | 24.7 ±4.2 | 56.7 ±2.7 | 52.0 ±2.7 | 67.4 ±1.2 | 64.8 ±2.6 | 78.0 ±0.4 | 67.0 ±2.5 | 74.1 ±0.8 | **91.5** ±0.3 | 33.9 ±1.8 | 76.6 ±0.7 | 46.4 ±0.6 | 73.2 ±0.8 | 57.7 ±1.0 |
| | HC | 10.3 ±0.3 | 10.3 ±0.1 | 10.1 ±0.0 | 17.5 ±9.5 | 11.8 ±1.0 | 18.1 ±3.7 | 18.9 ±5.6 | 19.0 ±1.3 | 14.1 ±1.0 | 25.4 ±10.9 | 41.6 ±6.3 | 11.1 ±0.5 | 15.1 ±0.6 | 13.7 ±0.6 | 14.6 ±1.1 | 16.8 ±0.2 |
| | UC | 24.8 ±25.5 | 41.8 ±27.4 | 10.1 ±0.0 | 72.4 ±3.4 | 53.0 ±2.2 | 70.0 ±1.5 | 74.1 ±2.2 | 69.9 ±2.1 | 69.9 ±1.2 | 66.6 ±10.0 | 77.9 ±0.6 | 67.7 ±1.9 | 64.2 ±2.1 | 67.2 ±1.1 | 63.2 ±2.9 | 59.5 ±3.0 |
| | HC + UC (HUS) | 21.7 ±14.1 | 40.5 ±19.4 | 10.7 ±1.0 | 74.0 ±4.6 | 46.6 ±13.0 | 72.5 ±4.1 | 78.1 ±3.6 | 73.3 ±0.9 | 71.3 ±1.9 | 76.6 ±2.3 | 81.1 ±1.2 | 40.9 ±17.3 | 65.9 ±0.9 | 67.4 ±2.7 | 63.3 ±6.0 | 58.9 ±2.6 |
| | ESM | 59.4 ±2.1 | 59.9 ±1.4 | 51.7 ±1.5 | 62.0 ±14.5 | 44.3 ±7.0 | 57.1 ±15.7 | 65.8 ±5.7 | 65.6 ±2.1 | 67.4 ±3.7 | 60.9 ±4.7 | 77.8 ±2.6 | 56.2 ±7.1 | 51.9 ±4.3 | 63.8 ±11.3 | 63.6 ±6.3 | 60.5 ±5.3 |
| | HC + ESM | 65.0 ±2.0 | 68.2 ±2.0 | 58.0 ±3.2 | 74.8 ±3.4 | 48.4 ±4.2 | 67.6 ±1.3 | 75.1 ±2.8 | 71.4 ±0.9 | 70.8 ±4.0 | 74.0 ±1.4 | 81.0 ±0.2 | 70.8 ±3.7 | 62.4 ±2.3 | 72.9 ±3.5 | 70.1 ±2.3 | 68.7 ±1.1 |
| | UC + ESM | 70.8 ±4.7 | 75.9 ±3.7 | 64.7 ±2.9 | 83.4 ±2.0 | 62.2 ±2.1 | 82.2 ±4.6 | 84.6 ±0.9 | 81.2 ±0.9 | 80.1 ±1.2 | 82.6 ±2.4 | 90.1 ±0.5 | 79.8 ±2.2 | 73.6 ±4.1 | 77.7 ±5.2 | 76.9 ±1.0 | 77.7 ±1.8 |
| | HUS + ESM (SoTTA) | 73.3 ±1.5 | 77.7 ±0.8 | 66.8 ±1.8 | 86.1 ±2.1 | 64.0 ±2.8 | 84.3 ±0.7 | 86.6 ±1.1 | 83.1 ±0.7 | 82.0 ±1.8 | 85.7 ±2.7 | 91.1 ±0.4 | 84.1 ±2.4 | 77.1 ±3.3 | 81.6 ±2.8 | 77.2 ±2.2 | 80.0 ±1.4 |

Table 9: Classification accuracy (%) and their corresponding standard deviations on ablation study of the size of Noise on CIFAR10-C for 15 types of corruptions. **Bold** numbers are the highest accuracy. Averaged over three different random seeds.

| | | Noise | | | Blur | | | | Weather | | | | Digital | | | | |
| | Method | Gau. | Shot | Imp. | Def. | Gla. | Mot. | Zoom | Snow | Fro. | Fog | Brit. | Cont. | Elas. | Pix. | JPEG | Avg. |
| --- | --- | --- | --- | --- | --- | --- | --- | --- | --- | --- | --- | --- | --- | --- | --- | --- | --- |
| 5000 | Source | 26.0 ±3.3 | 33.2 ±3.5 | 24.7 ±4.2 | 56.7 ±2.7 | 52.0 ±2.7 | 67.4 ±1.2 | 64.8 ±2.6 | 78.0 ±0.4 | 67.0 ±2.5 | 74.1 ±0.8 | **91.5** ±0.3 | 33.9 ±1.8 | 76.6 ±0.7 | 46.4 ±0.6 | 73.2 ±0.8 | 57.7 ±1.0 |
| | BN stats [27] | 59.6 ±0.2 | 61.6 ±0.7 | 51.5 ±0.6 | 66.9 ±1.2 | 49.8 ±0.9 | 65.3 ±0.1 | 68.6 ±0.7 | 71.2 ±0.5 | 71.5 ±0.2 | 65.0 ±0.6 | 84.2 ±0.3 | 70.0 ±1.1 | 58.8 ±1.2 | 63.5 ±0.5 | 65.4 ±0.9 | 64.9 ±0.4 |
| | PL [17] | 59.9 ±4.7 | 61.3 ±3.6 | 52.1 ±1.7 | 66.4 ±1.4 | 46.1 ±4.2 | 62.7 ±1.5 | 67.2 ±2.7 | 69.6 ±1.5 | 69.2 ±1.3 | 65.4 ±2.0 | 83.8 ±1.2 | 66.8 ±4.4 | 55.5 ±3.0 | 64.9 ±1.2 | 65.6 ±4.3 | 63.8 ±0.3 |
| | TENT [38] | 64.3 ±2.0 | 70.0 ±2.4 | 59.3 ±0.1 | 68.8 ±4.2 | 47.7 ±3.1 | 65.8 ±3.0 | 72.4 ±4.9 | 73.6 ±5.0 | 73.4 ±5.3 | 65.4 ±6.0 | 88.2 ±0.8 | 69.0 ±2.6 | 63.7 ±3.8 | 72.2 ±1.2 | 67.8 ±5.6 | 68.1 ±1.3 |
| | LAME [1] | 22.0 ±3.6 | 28.9 ±3.7 | 18.8 ±3.5 | 52.2 ±2.1 | 51.2 ±3.5 | 64.9 ±0.5 | 61.5 ±1.4 | 78.9 ±0.6 | 68.0 ±4.1 | 72.3 ±1.4 | 90.3 ±0.3 | 27.6 ±1.3 | 75.3 ±0.7 | 43.6 ±0.9 | 73.8 ±1.0 | 55.3 ±0.5 |
| | CoTTA [39] | 68.6 ±1.5 | 69.7 ±1.6 | 61.8 ±1.8 | 64.9 ±3.8 | 53.1 ±4.0 | 62.1 ±2.7 | 68.8 ±2.0 | 72.6 ±1.2 | 75.9 ±0.4 | 64.5 ±3.8 | 86.1 ±0.3 | 62.1 ±1.5 | 59.7 ±2.8 | 71.0 ±2.3 | 70.2 ±1.7 | 67.4 ±1.6 |
| | EATA [28] | 58.4 ±0.3 | 61.7 ±2.8 | 45.1 ±6.5 | 58.8 ±3.7 | 38.7 ±9.3 | 57.9 ±2.5 | 64.5 ±3.9 | 62.7 ±1.9 | 63.3 ±3.3 | 62.2 ±3.6 | 76.0 ±0.7 | 54.6 ±16.1 | 48.2 ±2.9 | 64.6 ±2.5 | 60.5 ±1.7 | 58.5 ±0.8 |
| | SAR [29] | 60.9 ±1.3 | 63.0 ±1.9 | 53.6 ±2.5 | 67.5 ±0.8 | 50.5 ±0.4 | 65.9 ±0.5 | 69.1 ±0.4 | 71.2 ±0.6 | 71.4 ±0.2 | 65.4 ±0.3 | 84.2 ±0.3 | 70.3 ±0.9 | 59.4 ±0.7 | 63.7 ±0.5 | 65.7 ±0.6 | 65.5 ±0.3 |
| | RoTTA [44] | 64.9 ±0.5 | 67.0 ±0.8 | 56.9 ±1.2 | 81.4 ±0.6 | 59.9 ±0.9 | 81.1 ±0.8 | 83.1 ±0.4 | 80.1 ±0.7 | 78.3 ±1.1 | 78.9 ±0.5 | 91.0 ±0.4 | 68.0 ±4.6 | 72.8 ±0.7 | 73.9 ±0.2 | 72.9 ±0.9 | 74.0 ±0.8 |
| | SoTTA | **74.1** ±1.0 | **77.3** ±0.9 | **67.4** ±0.2 | **86.2** ±1.9 | **64.7** ±2.5 | **85.0** ±2.3 | **87.5** ±1.1 | **84.3** ±3.4 | **82.3** ±2.2 | **85.0** ±2.5 | 91.4 ±0.3 | **83.5** ±3.4 | **77.8** ±2.2 | **82.7** ±1.4 | **78.6** ±0.9 | **80.5** ±1.0 |
| 10000 | Source | 26.0 ±3.3 | 33.2 ±3.5 | 24.7 ±4.2 | 56.7 ±2.7 | 52.0 ±2.7 | 67.4 ±1.2 | 64.8 ±2.6 | 78.0 ±0.4 | 67.0 ±2.5 | 74.1 ±0.8 | **91.5** ±0.3 | 33.9 ±1.8 | 76.6 ±0.7 | 46.4 ±0.6 | 73.2 ±0.8 | 57.7 ±1.0 |
| | BN stats [27] | 51.7 ±0.3 | 53.9 ±0.6 | 45.5 ±0.7 | 52.7 ±2.0 | 41.5 ±1.7 | 51.0 ±0.7 | 55.1 ±1.5 | 62.8 ±0.7 | 63.8 ±0.2 | 53.8 ±0.6 | 76.9 ±0.3 | 55.8 ±2.5 | 46.8 ±1.8 | 54.8 ±0.7 | 56.4 ±1.0 | 54.8 ±0.8 |
| | PL [17] | 47.6 ±9.9 | 52.7 ±2.4 | 44.7 ±4.3 | 48.9 ±12.6 | 36.1 ±5.1 | 49.4 ±1.8 | 54.1 ±2.9 | 61.9 ±2.4 | 56.5 ±4.4 | 50.9 ±1.3 | 77.1 ±3.7 | 45.2 ±4.8 | 43.1 ±4.5 | 49.4 ±5.6 | 59.5 ±4.7 | 51.8 ±0.9 |
| | TENT [38] | 54.0 ±6.7 | 57.1 ±5.6 | 36.7 ±9.1 | 48.9 ±6.8 | 28.3 ±4.5 | 50.5 ±3.1 | 51.0 ±5.0 | 64.0 ±4.1 | 64.7 ±5.2 | 49.5 ±1.9 | 80.5 ±1.4 | 43.7 ±3.0 | 38.4 ±2.2 | 56.7 ±6.4 | 57.0 ±4.5 | 52.1 ±0.4 |
| | LAME [1] | 21.8 ±3.5 | 28.6 ±3.7 | 18.5 ±3.1 | 51.6 ±2.3 | 50.8 ±3.6 | 64.3 ±0.2 | 60.9 ±1.8 | 78.4 ±0.5 | 67.3 ±3.8 | 71.7 ±1.2 | 90.5 ±0.2 | 27.0 ±1.2 | 75.1 ±0.7 | 43.0 ±0.9 | 73.4 ±1.0 | 54.9 ±0.6 |
| | CoTTA [39] | 60.4 ±2.1 | 60.3 ±3.5 | 52.4 ±1.6 | 47.3 ±3.0 | 41.6 ±0.4 | 44.1 ±2.7 | 52.0 ±4.7 | 62.7 ±0.6 | 66.6 ±0.8 | 47.7 ±2.4 | 79.0 ±1.7 | 44.7 ±1.1 | 42.8 ±4.3 | 60.2 ±0.5 | 60.2 ±1.0 | 54.8 ±1.3 |
| | EATA [28] | 42.2 ±1.1 | 41.0 ±1.1 | 33.2 ±5.9 | 32.7 ±5.1 | 25.0 ±1.5 | 27.9 ±2.1 | 34.3 ±5.4 | 40.8 ±2.7 | 42.6 ±6.5 | 31.6 ±11.5 | 61.5 ±5.7 | 20.3 ±2.2 | 27.5 ±4.1 | 35.8 ±4.5 | 43.1 ±8.3 | 36.0 ±0.8 |
| | SAR [29] | 57.5 ±1.0 | 59.3 ±0.2 | 49.6 ±1.7 | 57.2 ±1.1 | 43.7 ±1.7 | 54.4 ±1.5 | 59.4 ±1.6 | 64.8 ±1.0 | 65.4 ±0.3 | 57.9 ±0.4 | 77.1 ±0.2 | 60.2 ±1.8 | 50.0 ±1.2 | 58.3 ±0.6 | 59.8 ±0.1 | 58.3 ±0.3 |
| | RoTTA [44] | 64.4 ±0.5 | 66.9 ±0.8 | 56.1 ±1.4 | 80.1 ±0.4 | 59.1 ±0.6 | 79.8 ±0.5 | 82.2 ±0.2 | 79.7 ±0.4 | 78.7 ±0.4 | 77.8 ±0.4 | 91.2 ±0.5 | 69.0 ±4.0 | 72.3 ±1.2 | 73.4 ±0.2 | 72.8 ±0.7 | 73.6 ±0.5 |
| | SoTTA | **73.3** ±1.5 | **77.7** ±0.8 | **66.8** ±1.8 | **86.1** ±2.1 | **64.0** ±2.8 | **84.3** ±0.7 | **86.6** ±1.1 | **83.1** ±2.4 | **82.0** ±3.3 | **85.7** ±2.8 | 91.1 ±0.3 | **84.1** ±2.4 | **77.1** ±3.3 | **81.6** ±2.2 | **77.2** ±1.0 | **80.0** ±1.4 |
| 20000 | Source | 26.0 ±3.3 | 33.2 ±3.5 | 24.7 ±4.2 | 56.7 ±2.7 | 52.0 ±2.7 | 67.4 ±1.2 | 64.8 ±2.6 | 78.0 ±0.4 | 67.0 ±2.5 | 74.1 ±0.8 | **91.5** ±0.3 | 33.9 ±1.8 | 76.6 ±0.7 | 46.4 ±0.6 | 73.2 ±0.8 | 57.7 ±1.0 |
| | BN stats [27] | 41.3 ±0.7 | 42.9 ±1.0 | 37.2 ±0.4 | 37.4 ±2.1 | 32.6 ±1.4 | 36.1 ±1.0 | 39.6 ±1.4 | 52.0 ±0.7 | 52.7 ±0.5 | 40.6 ±1.2 | 65.0 ±0.5 | 37.4 ±4.1 | 33.9 ±1.8 | 44.1 ±0.9 | 44.4 ±0.5 | 42.5 ±0.8 |
| | PL [17] | 25.5 ±1.1 | 22.6 ±3.2 | 27.6 ±4.2 | 20.5 ±7.0 | 21.7 ±5.0 | 20.2 ±6.7 | 21.8 ±8.3 | 50.0 ±8.4 | 41.7 ±13.4 | 24.0 ±9.0 | 60.8 ±3.6 | 17.7 ±9.2 | 21.3 ±5.0 | 23.1 ±10.1 | 29.5 ±8.3 | 28.5 ±2.3 |
| | TENT [38] | 21.5 ±4.5 | 19.7 ±2.3 | 21.4 ±2.2 | 14.2 ±1.4 | 18.5 ±3.4 | 17.8 ±4.4 | 15.3 ±3.5 | 44.2 ±6.7 | 36.6 ±4.8 | 15.2 ±1.3 | 63.8 ±1.8 | 19.9 ±2.7 | 13.3 ±1.8 | 28.1 ±10.7 | 25.0 ±8.3 | 25.0 ±2.9 |
| | LAME [1] | 21.7 ±3.4 | 28.2 ±3.6 | 18.1 ±3.0 | 50.4 ±2.7 | 49.6 ±3.0 | 63.6 ±2.0 | 60.1 ±1.8 | 77.9 ±0.5 | 66.1 ±3.3 | 71.0 ±0.1 | 90.6 ±0.1 | 26.6 ±1.4 | 75.0 ±0.5 | 42.6 ±0.7 | 73.3 ±1.0 | 54.3 ±0.6 |
| | CoTTA [39] | 42.7 ±1.5 | 46.7 ±2.6 | 39.0 ±3.0 | 31.0 ±0.7 | 29.8 ±2.2 | 32.0 ±1.4 | 35.3 ±2.4 | 50.9 ±3.9 | 55.3 ±3.5 | 31.9 ±1.2 | 67.6 ±1.2 | 28.9 ±5.6 | 29.5 ±3.2 | 45.9 ±5.2 | 46.8 ±2.8 | 40.9 ±1.7 |
| | EATA [28] | 22.3 ±3.8 | 23.9 ±3.5 | 19.2 ±1.1 | 15.1 ±1.6 | 16.1 ±4.6 | 15.9 ±3.1 | 17.4 ±2.3 | 21.8 ±5.5 | 19.5 ±0.9 | 15.1 ±1.2 | 32.7 ±9.4 | 14.7 ±1.7 | 15.0 ±2.0 | 20.9 ±0.6 | 23.1 ±2.9 | 19.5 ±0.6 |
| | SAR [29] | 41.6 ±1.9 | 43.7 ±0.9 | 39.6 ±2.6 | 33.4 ±8.6 | 29.3 ±5.4 | 34.6 ±3.8 | 38.1 ±3.2 | 55.8 ±3.1 | 56.1 ±3.0 | 38.5 ±4.0 | 68.4 ±3.2 | 33.0 ±9.8 | 28.9 ±7.2 | 46.6 ±2.8 | 45.3 ±0.8 | 42.2 ±1.9 |
| | RoTTA [44] | 62.5 ±0.5 | 64.5 ±1.1 | 54.6 ±1.7 | 78.9 ±0.4 | 58.3 ±0.4 | 79.0 ±0.4 | 81.3 ±0.7 | 80.0 ±1.0 | 79.0 ±0.4 | 77.3 ±0.5 | 91.3 ±0.4 | 69.0 ±0.4 | 71.5 ±0.3 | 73.4 ±0.6 | 72.4 ±0.6 | 72.9 ±0.3 |
| | SoTTA | **73.2** ±1.0 | **75.6** ±2.8 | **63.3** ±3.8 | **83.2** ±2.8 | **61.0** ±3.5 | **84.5** ±2.6 | **86.3** ±2.4 | **82.6** ±0.6 | **81.0** ±3.3 | **84.8** ±1.8 | 89.7 ±0.3 | **82.8** ±4.9 | **72.9** ±2.4 | **81.1** ±1.5 | **77.5** ±1.0 | **78.6** ±1.5 |

# C  Additional ablative studies

We conducted experiments to understand the sensitivity of our two hyperparameters: confidence threshold ($C_0$) and BN momentum ($m$). We varied $C_0$ and $m$ and reported the corresponding accuracy.

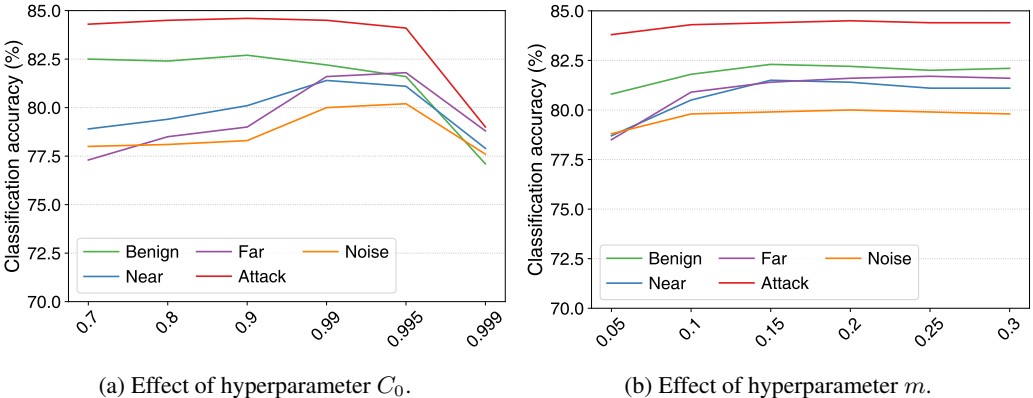

(a) Effect of hyperparameter $C_0$.  (b) Effect of hyperparameter $m$.

Figure 8: Effect of hyperparameters on the model accuracy on CIFAR10-C for 15 types of corruptions under five scenarios: Benign, Near, Far, Attack, and Noise. Averaged over three different random seeds.

**Confidence threshold.**  Our result shows that the selection of $C_0$ shows similar patterns across different scenarios (Benign $\sim$ Noise). The result illustrates a tradeoff; a low $C_0$ value does not effectively reject noisy samples, while a high $C_0$ value filters benign data. We found a proper value of $C_0$ (0.99) that generally works well across the scenarios. Also, we found that the optimal $C_0$ depends primarily on in-distribution data. Our interpretation is that setting different $C_0$ values for CIFAR10-C, CIFAR100-C, and ImageNet-C is straightforward as they have a different number of classes (10, 100, and 1,000), which leads to different ranges of the model's confidence.

**BN momentum.**  Across the tested range, the variations in performance were found to be negligible. This finding indicates that choosing a low momentum value from within the specified range ([0.05, 0.3]) is adequate to maintain a favorable performance. Please note that setting a high momentum would corrupt the result, which is implicated by the algorithms directly utilizing test-time statistics (e.g., TENT) suffering from accuracy degradation with noisy data streams (e.g., TENT: 81.0% $\rightarrow$ 52.1% for Noise at Table 1).

# D  Further discussions

## D.1  Theoretical explanation of the impact of noisy data streams

We provide a theoretical explanation of the impact of noisy data streams with a common entropy minimization as an example. With the Bayesian-learning-based frameworks [2, 7], we can express the posterior distribution $p$ of the model in terms of training data $D$ and benign test data $B$ in test-time adaptation:

$$\log p(\theta|D, B) = \log q(\theta) - \frac{\lambda_B}{|B|} \sum_{b=1}^{|B|} H(y_b|x_b). \tag{7}$$

The posterior distribution of model parameters depends on the prior distribution $q$ and the average of entropy $H$ of benign samples with a multiplier $\lambda$. Here, we incorporate the additional noisy data stream $N$ into Equation 7 and introduce a new posterior distribution considering noisy streams:

$$\log p(\theta|D, B, N) = \log q(\theta) - \frac{\lambda_B}{|B|} \sum_{b=1}^{|B|} H(y_b|x_b) - \frac{\lambda_N}{|N|} \sum_{n=1}^{|N|} H(y_n|x_n). \tag{8}$$

Table 10: Average classification accuracy (%) and their corresponding standard deviations on ablation study of the effect of high-confidence uniform-class continual memory of SoTTA on CIFAR10-C. **Bold** numbers are the highest accuracy. Averaged over three different random seeds.

|  | Benign | Near | Far | Attack | Noise | Avg |
|---|---|---|---|---|---|---|
| SoTTA (w/o High-confidence) | 82.2 ±0.2 | 78.0 ±0.4 | 75.9 ±0.5 | 84.3 ±0.1 | 77.7 ±0.7 | 79.6 ±0.2 |
| SoTTA (w/o Uniform-class) | **82.3 ±0.2** | 80.9 ±0.6 | 74.9 ±2.4 | 83.5 ±0.2 | 68.7 ±7.0 | 78.0 ±2.0 |
| SoTTA (w/o Continual) | 81.0 ±0.5 | 79.5 ±0.3 | 75.5 ±1.8 | 84.4 ±0.2 | 65.7 ±7.0 | 77.2 ±1.8 |
| SoTTA | 82.2 ±0.3 | **81.4 ±0.5** | **81.6 ±0.6** | **84.5 ±0.2** | **80.0 ±1.4** | **81.9 ±0.5** |

With Equation 7 and Equation 8, we can now derive model parameter variations caused by noisy test samples:

$$\log p(\theta|D, B) - \log p(\theta|D, B, N) = \frac{\lambda_N}{|N|} \sum_{n=1}^{M} H(y_n|x_n). \tag{9}$$

Equation 9 implies that the (1) model adapted only from benign data and (2) model adapted with both benign and noisy data differ by the amount of the average entropy of noisy samples. This also suggests that a high entropy from severe noisy samples would result in a significant model drift in adaptation (i.e., model corruption).

## D.2 Comparison with previous TTA methods

### D.2.1 EATA and SAR

While SoTTA, EATA [28], and SAR [29] all leverage sample filtering strategy, the key distinction of input-wise robustness of SoTTA and EATA/SAR lies in three aspects: (1) Our high-confidence sampling strategy in SoTTA aims to filter noisy samples by utilizing only the samples with high confidence, while both EATA and SAR use a different approach that excludes a few high-entropy samples, particularly during the early adaptation stage. In our preliminary study, we found that our method excludes 99.98% of the noisy samples, whereas EATA and SAR exclude 33.55% of such samples. (2) While EATA and SAR adapt to every incoming low-entropy sample, SoTTA leverages a uniform-class memory management approach to prevent overfitting. As shown in Figure 5b, noisy samples often lead to imbalanced class predictions, and these skewed distributions could lead to an undesirable bias in $p(y)$ and thus might negatively impact TTA objectives, such as entropy minimization. The ablation study in Table 10 shows the effectiveness of uniform sampling with a 3.9%p accuracy improvement. (3) EATA and SAR reset the memory buffer and restart the sample collection process for each adaptation. This strategy is susceptible to overfitting due to a smaller number of samples used for adaptation and the temporal distribution drift of the samples. In contrast, our continual memory management approach effectively mitigates this issue by retaining high-confidence uniform-class samples in the memory, as shown in Table 10.

We acknowledge that both SoTTA and SAR utilize sharpness-aware minimization proposed by Foret et al. [4]. However, we clarify that the motivation behind using SAM is different. While SAR intends to avoid model collapse when exposed to samples with large gradients, we aim to enhance the model's robustness to noisy samples with high confidence scores. As illustrated in Figure 6, we observed that entropy-sharpness minimization effectively prevents the model from overfitting to noisy samples. As a result, while our algorithm led to marginal performance degradation in noisy settings (82.2% → 80.0% for Noise), EATA and SAR showed significant degradation (EATA 82.4% → 36.0% for Noise; SAR 78.3% → 58.3% for Noise).

### D.2.2 RoTTA

Regarding our high-confidence uniform sampling technique, RoTTA [44] could be compared. First of all, RoTTA's objective is different from ours; RoTTA focused on temporal distribution changes of test streams without considering noisy samples. Similar to SoTTA, RoTTA's memory bank maintains recent high-confidence samples. However, RoTTA has no filtering mechanism for low-confidence samples, which makes RoTTA fail to avoid noisy samples, especially in the early stage of TTA. In contrast, our confidence-based memory management scheme effectively rejects noisy samples, and

Table 11: Average classification accuracy (%) of ODIN+TTA on CIFAR10-C. **Bold** numbers are the accuracy with improvement from normal TTAs. Averaged over three different random seeds.

| | Benign | | Near | | Far | | Attack | | Noise | |
|---|---|---|---|---|---|---|---|---|---|---|
| Method | w/o ODIN | w/ ODIN | w/o ODIN | w/ ODIN | w/o ODIN | w/ ODIN | w/o ODIN | w/ ODIN | w/o ODIN | w/ ODIN |
| Source | 57.7 ±1.0 | 57.7 ±1.0 | 57.7 ±1.0 | 57.7 ±1.0 | 57.7 ±1.0 | 57.7 ±1.0 | 57.7 ±1.0 | 57.7 ±1.0 | 57.7 ±1.0 | 57.7 ±1.0 |
| BN stats [27] | 78.2 ±0.3 | 78.2 ±0.3 | 76.5 ±0.4 | 76.5 ±0.4 | 75.4 ±0.3 | **75.9** ±0.4 | 55.8 ±1.4 | 55.8 ±1.4 | 55.9 ±0.8 | **56.7** ±0.9 |
| PL [17] | 78.4 ±0.3 | **78.8** ±0.5 | 73.1 ±0.3 | **74.3** ±0.6 | 71.3 ±1.0 | **71.6** ±0.8 | 66.5 ±1.1 | 66.5 ±1.1 | 52.1 ±0.4 | 52.1 ±0.4 |
| TENT [38] | 81.5 ±1.0 | 81.5 ±1.0 | 74.5 ±0.8 | **76.1** ±0.6 | 73.5 ±1.1 | **74.7** ±1.3 | 69.0 ±0.9 | **69.1** ±1.0 | 54.4 ±0.3 | **56.2** ±0.6 |
| LAME [1] | 56.1 ±0.3 | 56.1 ±0.3 | 56.7 ±0.5 | 56.7 ±0.5 | 55.7 ±0.4 | 55.7 ±0.4 | 56.2 ±0.5 | 56.2 ±0.5 | 54.9 ±0.5 | **55.2** ±0.7 |
| CoTTA [39] | 82.2 ±0.3 | 82.2 ±0.3 | 78.2 ±0.3 | 78.2 ±0.4 | 73.6 ±0.9 | 73.6 ±0.9 | 69.6 ±1.3 | 69.6 ±1.3 | 57.8 ±0.8 | **62.0** ±1.3 |
| EATA [28] | 82.4 ±0.3 | 82.4 ±0.3 | 63.9 ±0.4 | **69.2** ±0.4 | 56.3 ±0.5 | **59.9** ±0.6 | 70.9 ±0.7 | 70.9 ±0.7 | 36.0 ±0.8 | **50.8** ±1.1 |
| SAR [29] | 78.4 ±0.7 | 78.4 ±0.7 | 72.8 ±8.2 | 72.8 ±8.2 | 75.7 ±3.1 | **76.0** ±3.1 | 56.2 ±1.8 | 56.2 ±1.8 | 58.7 ±0.3 | 58.7 ±0.3 |
| RoTTA [44] | 75.3 ±0.7 | 75.3 ±0.7 | 77.5 ±0.5 | 77.5 ±0.5 | 77.0 ±0.9 | 77.0 ±0.9 | 78.4 ±0.8 | 78.4 ±0.8 | 73.5 ±0.5 | 73.5 ±0.5 |
| SoTTA | 82.1 ±0.4 | 82.1 ±0.4 | 81.6 ±0.4 | 81.6 ±0.4 | 81.7 ±0.5 | **82.0** ±0.8 | 84.5 ±0.3 | 84.5 ±0.3 | 81.5 ±1.2 | 81.5 ±1.2 |

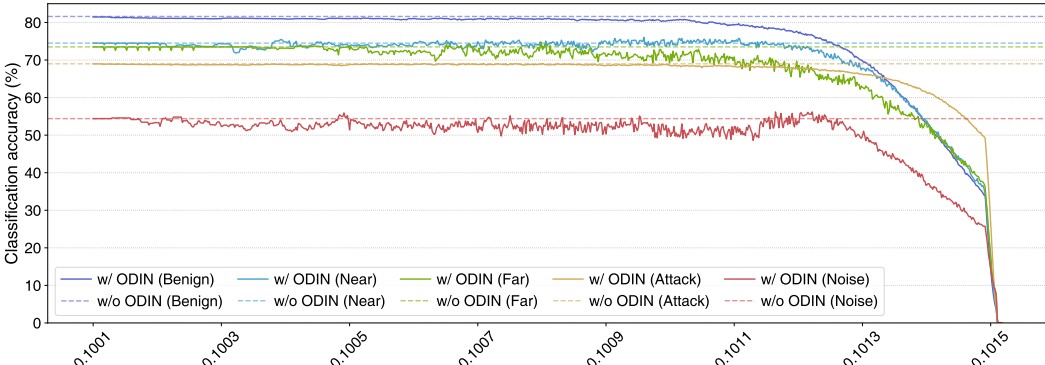

Figure 9: Effect of OOD threshold $\delta$ on classification accuracy (%) of ODIN+TENT on CIFAR10-C. Averaged over three different random seeds.

thus it prevents potential model drift from the beginning of TTA scenarios. As a result, our approach outperforms RoTTA in noisy test streams (e.g., 5.4%p better than RoTTA on CIFAR10-C).

### D.3 Comparison with out-of-distribution detection algorithms

We discussed the limitation of applying out-of-distribution detection to TTA in Section 5. Still, we are curious about the effect of applying out-of-distribution algorithms to our scenario. To this end, we conduct experiments using one of the out-of-distribution algorithms, ODIN [20], in our noisy data streams. Specifically, we filtered OOD samples detected by ODIN and performed TTA algorithms on the samples left.

Note that similar to prior studies on OOD, ODIN uses a thresholding approach to predict whether a sample is OOD. It thus requires validation data with binary labels indicating whether it is in-distribution or OOD to decide the best threshold $\delta$. However, in TTA scenarios, validation data is not provided, which makes it difficult to apply OOD algorithms directly in our scenario. We circumvented this problem using the labeled test batches to get the best threshold. Following the original paper, we searched for the best threshold from 0.1 to 0.12 with a step size of 0.000001, which took over 20,000 times longer than the original TTA algorithm.

Table 11 shows that the impact of discarding OOD samples with ODIN is negligible, yielding only a 0.3%p improvement in the average accuracy despite a huge computation cost. Also, Figure 9 shows the high sensitivity of ODIN with respect to threshold hyperparameter $\delta$, which implies that applying OOD in TTA is impractical.

We conclude the practical limitations of OOD detection algorithms for TTA as follows: (1) OOD methods assume that a model is fixed during test time, while a model changes continually in TTA. (2) As previously noted, most OOD algorithms require labels for validation data unavailable in TTA scenarios. Even using the same test dataset for selecting the threshold, the performance improvement was marginal. (3) Low performance possibly results from the fact that OOD detection studies are

built on the condition that training and test domains are the same, which differs from TTA's scenario. These collectively make it difficult to apply OOD detection studies directly to TTA scenarios.

### D.4 Applying to other domains

While this study primarily focuses on classification tasks, there are other tasks where test-time adaptation would be useful. Here we discuss the applicability of SoTTA to (1) image segmentation and (2) object detection, which are crucial in autonomous driving scenarios.

For image segmentation, when noisy objects are present in the input, the model might produce noisy predictions on those pixels, leading to detrimental results. Extending SoTTA to operate at the pixel level would allow it to be compatible with the segmentation task while minimizing the negative influences of those noisy pixels on model predictions in test-time adaptation scenarios.

Similarly, SoTTA could be tailored to object detection's classification (recognition) task. For example, in the context of the YOLO framework [32], SoTTA could filter and store grids with high confidence for test-time adaptation, enhancing detection accuracy. However, our current approach must address the localization task (bounding box regression) during test-time adaptation. Implementing this feature is non-trivial and would require careful consideration and potential redesign of certain aspects of our methodology. Accurately localizing bounding boxes during test-time adaptation presents an exciting avenue for future research.

## E License of assets

**Datasets** CIFAR10/CIFAR100 (MIT License), CIFAR10-C/CIFAR100-C (Creative Commons Attribution 4.0 International), ImageNet-C (Apache 2.0), and MNIST (CC-BY-NC-SA 3.0).

**Codes** Torchvision for ResNet18 (Apache 2.0), the official repository of CoTTA (MIT License), the official repository of TENT (MIT License), the official repository of LAME (CC BY-NC-SA 4.0), the official repository of EATA (MIT License), the official repository of SAR (BSD 3-Clause License), and the official repository of RoTTA (MIT License).

