# OpenReview forum: "SoTTA: Robust Test-Time Adaptation on Noisy Data Streams"
_NeurIPS.cc/2023/Conference — NeurIPS 2023 poster_

### Official Review · Reviewer_HUhd · 2023-06-29

**Soundness:** 2 fair
**Presentation:** 3 good
**Contribution:** 2 fair
**Rating:** 5
**Confidence:** 5

**Summary:**

The authors point out that model may suffer from non-interest samples while TTA. Existing TTA methods are not robust to these samples. To address these issues, the authors proposed a methods called SoTTA with two key components, input-wise robustness via high-confidence uniform-class sampling and parameter-wise robustness via entropy-sharpness minimization.

**Strengths:**

The authors focus on TTA in the wild and point out that TTA with non-interest would lead to performance degradation.

**Weaknesses:**

The proposed method seems to be not novel.

A) The proposed “Input-wise robustness via high-confidence uniform-class sampling” method excludes low-confidence samples in TTA. Such samples filtration strategies have been explored in ETA [Efficient Test-Time Model Adaptation without Forgetting].

B) The proposed “Parameter-wise robustness via entropy-sharpness minimization” method introduces entropy-sharpness minimization, which have been explored in SAR [Towards Stable Test-time Adaptation in Dynamic Wild World].

It would be better for the authors to clarify the differences between the proposed methods and the TTA methods I mentioned above.

In the experimental results, I found the proposed always outperforms SAR. But in my understanding, the main components of these two methods are almost the same. Both of them include low-confidence sample filtration strategy and entropy-sharpness minimization. So I have no idea why the proposed method can yield much better results.

**Questions:**

See comments above.

**Limitations:**

See comments above.

---

> ### Author Rebuttal · Authors · 2023-08-09
>
> We sincerely appreciate your time and effort in providing us with positive comments. We respond to your question in what follows. Please also refer to the *global response* we have posted together.
>
> ---
>
> **Weakness 1. It would be better for the authors to clarify the differences between the proposed methods and the TTA methods I mentioned above.**
>
> While SoTTA (ours), EATA, and SAR all leverage sample filtering strategy, the key distinction of input-wise robustness of SoTTA and EATA/SAR lies in two aspects: (1) characteristics of the samples filtered out and (2) memory management strategies.
>
> First, our high-confidence sampling strategy in SoTTA aims to filter non-interest samples by utilizing only the samples with high confidence, while EATA and SAR use the same algorithm that excludes a few high-entropy samples, particularly during the early adaptation stage. As evidence, we show that our method excludes 99.98% of the non-interest samples, whereas EATA and SAR exclude 33.55% of such samples.
>
> Second, while EATA and SAR adapt to every incoming low-entropy sample, SoTTA leverages a uniform-class sampling approach to prevent overfitting. As shown in Figure 5, non-interest samples often lead to imbalanced class predictions, and these skewed distributions could lead to an undesirable bias in p(y) and thus might negatively impact TTA objectives, such as entropy minimization. The ablation study in Table 2 of our paper shows the effectiveness of uniform sampling with a 9.8% accuracy improvement.
>
> We acknowledge that both SoTTA and SAR utilize sharpness-aware minimization proposed by Foret et al. [r1]. However, we clarify that the motivation behind using SAM is different. While SAR intends to avoid model collapse when exposed to samples with large gradients, we aim to enhance the model's robustness to non-interest samples with high confidence scores. As illustrated in Figure 6 in the manuscript, we observed that entropy-sharpness minimization effectively prevents the model from overfitting to non-interest samples.
>
> In conclusion, while our algorithm results in marginal performance degradation in noisy settings (82.9% -> 81.0% for Noise), EATA and SAR show significant degradation: EATA 82.4% -> 36.6% for Noise and SAR 78.3% -> 58.3% for Noise. The detailed results of SoTTA and SAR are included in the manuscript, and the result of EATA is included in the global response #1.
>
>
>
>
> **Weakness 2. In the experimental results, I found the proposed always outperforms SAR. But in my understanding, the main components of these two methods are almost the same. Both of them include low-confidence sample filtration strategy and entropy-sharpness minimization. So I have no idea why the proposed method can yield much better results.**
>
>
> We summarize the key difference between SoTTA and SAR in the comments above. To conclude, It is precisely due to the robustness of our method against non-interest samples that SoTTA outperforms SAR. By reducing the influence of non-interest samples and considering class imbalance through uniform-class sampling, our approach mitigates the risk of model corruption caused by biased learning from a few non-interest samples. It results in better overall performance, as demonstrated in our experimental results.
>
> We thank the reviewer for pointing out this, and we will revise our manuscript to clarify the difference between SoTTA and SAR.
>
>
> [r1] Foret, Pierre, et al. "Sharpness-aware Minimization for Efficiently Improving Generalization." International Conference on Learning Representations. 2020.

---

> > ### Comment · Reviewer_HUhd · 2023-08-16
> > **Further Comments**
> >
> > Thank you for the authors' response.
> >
> > The authors highlight two main distinctions: *(1) characteristics of the samples filtered out and (2) memory management strategies.* However, in their detailed response, I struggled to grasp the exact disparity in memory management.
> >
> > Moreover, I'm seeking a clear understanding of the technical contrasts between the proposed method and ETA/SAR. For instance, while ETA performs operation "A", your SoTTA performs operation "B". In your scenario, "B" proves more effective than "A," leading to improved accuracy.
> >
> > The differences presented in the current version of the response are somewhat general. I recognize that your method's motivation varies from ETA/SAR, yet I'm specifically interested in perceiving technical distinctions.

---

> > > ### Author Response · Authors · 2023-08-16
> > > **Thank you for your response**
> > >
> > > We appreciate your response to our rebuttal.
> > >
> > > We first clarify the main technical distinctions between SoTTA (ours) and EATA/SAR regarding sample management.
> > >
> > > EATA and SAR employ the same strategy of "excluding high-entropy samples" as follows:
> > >
> > > ---
> > >
> > > Input: test data stream $\mathbf{x}_t$, memory $M$ with capacity $N$, entropy threshold $E_0$
> > >
> > > for test time $t \in \\{1, \cdots, T\\}$ do
> > >
> > > &nbsp;&nbsp; if $E(\mathbf{x}_t; \theta) < E_0$ then
> > >
> > > &nbsp;&nbsp;&nbsp;&nbsp; Add $(\mathbf{x}_t, \hat{y_t})$ to $M$
> > >
> > > &nbsp;&nbsp; if $t$ % batch_size == $0$ then
> > >
> > > &nbsp;&nbsp;&nbsp;&nbsp; Update model $\theta$ with $M$
> > >
> > > &nbsp;&nbsp;&nbsp;&nbsp; Set $M$ = $\emptyset$   &nbsp;&nbsp;&nbsp;*# re-collect data from scratch*
> > >
> > > ---
> > >
> > > Our memory management scheme is as follows:
> > >
> > > ---
> > >
> > > Input: test data stream $\mathbf{x}_t$, memory $M$ with capacity $N$, confidence threshold $C_0$
> > >
> > > for test time $t \in \\{1, \cdots, T\\}$ do
> > >
> > > &nbsp;&nbsp; if $C(\mathbf{x}_t; \theta) > C_0$ then
> > >
> > > &nbsp;&nbsp;&nbsp;&nbsp; if $|M| < N$ then
> > >
> > > &nbsp;&nbsp;&nbsp;&nbsp;&nbsp;&nbsp; Add $(\mathbf{x}_t, \hat{y_t})$ to $M$
> > >
> > > &nbsp;&nbsp;&nbsp;&nbsp; else
> > >
> > > &nbsp;&nbsp;&nbsp;&nbsp;&nbsp;&nbsp; $\mathcal{Y}^* \gets$ the most prevalent class(es) in $M$
> > >
> > > &nbsp;&nbsp;&nbsp;&nbsp;&nbsp;&nbsp; if $\hat{y}_t \notin \mathcal{Y}^*$ then  &nbsp;&nbsp;&nbsp;*# balancing classes*
> > >
> > > &nbsp;&nbsp;&nbsp;&nbsp;&nbsp;&nbsp;&nbsp;&nbsp; Randomly discard $(\mathbf{x}_i, \hat{y}_i)$ from $M$ where $\hat{y}_i \in \mathcal{Y}^*$
> > >
> > > &nbsp;&nbsp;&nbsp;&nbsp;&nbsp;&nbsp; else
> > >
> > > &nbsp;&nbsp;&nbsp;&nbsp;&nbsp;&nbsp;&nbsp;&nbsp; Randomly discard $(\mathbf{x}_i, \hat{y}_i)$ from $M$ where $\hat{y}_i = \hat{y}_t$
> > >
> > > &nbsp;&nbsp;&nbsp;&nbsp;&nbsp;&nbsp; Add $(\mathbf{x}_t, \hat{y}_t)$ to $M$
> > >
> > > &nbsp;&nbsp; if $t$ % batch_size == $0$ then
> > >
> > > &nbsp;&nbsp;&nbsp;&nbsp; Update model $\theta$ with $M$
> > >
> > > ---
> > >
> > > SoTTA’s two technical distinctions regarding memory management strategy are (1) **uniform-class memory management** and (2) **continual memory management**.
> > >
> > > First, EATA and SAR filter out low-entropy data without considering their distribution, which suffers from skewed predicted distributions of non-interest samples. This has a detrimental effect on TTA objectives, such as entropy minimization. Our proposed **uniform-class memory management (Uniform-class)** addresses this issue, resulting in improved accuracy.
> > >
> > > Second, EATA and SAR gather only low-entropy samples until a batch-sized number of test samples pass by. These gathered samples are then utilized for adaptation. Subsequently, EATA and SAR reset the memory buffer and restart the sample collection process for each adaptation. This strategy is susceptible to overfitting due to a smaller number of samples used for adaptation and the temporal distribution drift of the samples. In contrast, our **continual memory management (Continual)** approach effectively mitigates this issue by retaining high-confidence uniform-class samples in the memory.
> > >
> > > We also provide the results of an ablation study that demonstrates the effectiveness of each technique:
> > >
> > > | | Benign | Near | Far | Attack | Noise | Avg |
> > > |---|:---:|:---:|:---:|:---:|:---:|:---:|
> > > | SoTTA (w\o Uniform-class) | 83.1±0.5 | 76.7±4.6 | 66.7±5.0 | 83.9±0.8 | 52.3±19.2 | 72.5 |
> > > | SoTTA (w\o Continual) | 81.0±0.5 | 79.5±0.3 | 75.5±1.8 | 84.4±0.2 | 65.7±7.0 | 77.2 |
> > > | SoTTA | 82.9±0.4 | 81.4±0.5 | 81.6±0.5 | 84.5±0.3 | 81.0±1.5 | 82.3 |
> > >
> > >
> > > Thank you again for your valuable feedback and suggestions. We will carefully incorporate the discussions and results in our final manuscript. Please don’t hesitate to leave additional comments if you have any follow-up questions or discussions.
> > >
> > > Best,
> > >
> > > Authors

---

> > > > ### Comment · Reviewer_HUhd · 2023-08-21
> > > > **Further comments**
> > > >
> > > > Thanks for your detailed responses. Currently, I understand the differences between your method and related works.
> > > >
> > > > I would raise the scoring to 5.

---

> > > > > ### Author Response · Authors · 2023-08-21
> > > > > **Thank you for your follow-up response**
> > > > >
> > > > > We sincerely appreciate your response. Thank you once again for your valuable feedback and suggestions, as well as for recognizing the value of our contributions.
> > > > >
> > > > > Best regards,
> > > > >
> > > > > Authors.

---

### Official Review · Reviewer_6Z6q · 2023-07-04

**Soundness:** 1 poor
**Presentation:** 3 good
**Contribution:** 2 fair
**Rating:** 5
**Confidence:** 5

**Summary:**

This paper studies a practical problem of test-time adaptation where non-interest testing samples may appear and mislead the adaptation. This problem is quite serious in practical applications, and the problem setting is relatively novel. To address this problem, the authors propose the SoTTA method, which solves this problem in two aspects. For input-wise robustness, SoTTA filters out the non-interest samples with a high-confidence uniform-class memory buffer. For parameter-wise robustness, entropy-sharpness minimization is adopted to ensure that the landscape is smooth during the adaptation. The proposed method is evaluated on two benchmark datasets under one standard setting and four robust settings with different types of non-interest samples. The results show that SoTTA gives state-of-the-art performance compared to existing TTA methods.

**Strengths:**

1. This paper studies a novel problem setting of test-time adaptation, where non-interest testing samples may appear and mislead the adaptation. This is a relatively novel and practical problem in real applications.
2. The overall idea of this paper makes sense. This paper tackles the harmful impact of non-interest with a filter and robust loss function. The experiments in the main paper also show that the proposed method gives state-of-the-art performance.

**Weaknesses:**

1. First, this paper ignores one common TTA benchmark, i.e., CIFAR100. Additionally, the results of ImageNet-C have been hidden in the appendix. These key results should be presented in the main paper with comprehensive analysis.
2. One TTA algorithm, EATA [1], adopts a similar sample selection strategy during adaptation. This method should be taken into consideration in the experiment parts because it has the potential to handle the problem of non-interest samples.
3. This paper does not provide any discussion about the selection and sensitivity of threshold C0. Intuitively, the optimal C0 depends on both in-distribution data and out-of-distribution data, and it is not easy to decide in real applications.
4. There is more related work that can be discussed to further improve this paper. For test-time adaptation, the above-mentioned EATA method [1] and recent advanced TTA methods [2-4] about robustness and evaluation should be considered. For out-of-distribution detection, methods [5-6] that can be efficiently optimized in an unsupervised manner should be taken into consideration.

**Reference**

[1] Shuaicheng Niu, Jiaxiang Wu, Yifan Zhang, Yaofo Chen, Shijian Zheng, Peilin Zhao, Mingkui Tan: Efficient Test-Time Model Adaptation without Forgetting. ICML 2022: 16888-16905

[2] Hao Zhao, Yuejiang Liu, Alexandre Alahi, Tao Lin: On Pitfalls of Test-Time Adaptation. ICML 2023

[3] Zhi Zhou, Lan-Zhe Guo, Lin-Han Jia, Dingchu Zhang, Yu-Feng Li: ODS: Test-Time Adaptation in the Presence of Open-World Data Shift. ICML 2023

[4] Tong Wu, Feiran Jia, Xiangyu Qi, Jiachen T. Wang, Vikash Sehwag, Saeed Mahloujifar, Prateek Mittal: Uncovering Adversarial Risks of Test-Time Adaptation. ICML 2023

[5] Zhi Zhou, Lan-Zhe Guo, Zhanzhan Cheng, Yu-Feng Li, Shiliang Pu: STEP: Out-of-Distribution Detection in the Presence of Limited In-Distribution Labeled Data. NeurIPS 2021: 29168-29180

[6] Jiangpeng He, Fengqing Zhu: Out-Of-Distribution Detection In Unsupervised Continual Learning. CVPR Workshops 2022: 3849-3854

**Questions:**

1. The experiment shows that RoTTA outperforms existing TTA methods even in standard settings. Can the authors explain why this phenomenon occurs? Intuitively, the techniques mentioned in this article are all aimed at addressing the problem of non-interesting samples. What factors enable them to achieve better performance when non-interesting samples do not exist?
2. Can the authors provide some sensitivity analysis regarding the threshold C0? It is very important for the actual effect of filtering out non-interesting samples. An adaptive scheme for setting C0 would also be acceptable.
3. One naive way to solve the proposed setting is to adopt an out-of-distribution process before test-time adaptation. How does this naive method work in the proposed setting?

**[IMPORTANT]** I would like to raise my score if you successfully address my concerns in the Weakness and Question sections. Overall, this paper is interesting and I appreciate it. However, it is currently slightly below the level of acceptance.

**Limitations:**

The authors have properly discussed the limitations.

---

> ### Author Rebuttal · Authors · 2023-08-09
>
> We sincerely appreciate your time and effort in providing us with positive comments. We respond to your question in what follows. Please also refer to the *global response* we have posted together.
>
> ---
> **Weakness 1.**
>
> In our current manuscript, the CIFAR100 dataset acted as one of the non-interest scenarios ("Near") as it has similar characteristics to CIFAR10 and ImageNet, as detailed in Section 4 of our manuscript. However, we acknowledge the importance of the CIFAR100 benchmark. We will conduct experiments on the CIFAR100 dataset and include the results in the final manuscript.
>
> For your concerns regarding ImageNet-C, we put its result in the appendix due to space limitations. We will restructure the evaluations section to incorporate the ImageNet-C result in the main paper.
>
>
> **Weakness 2.**
>
> Thank you for your valuable suggestion. We include the results of EATA and discussions in global response #1. To summarize, we found that EATA shows significant accuracy degradation with the presence of non-interest samples (e.g., 82.4% -> 36.6% for Noise).
>
>
> **Weakness 3.**
>
> Thank you for your valuable suggestion. We include the sensitivity of the hyperparameter C0 and discussions in global response #2.
>
>
> **Weakness 4.**
>
> We want to highlight that our problem setting and approach differ from those studies. EATA [1] is primarily concerned with preventing catastrophic forgetting, which differs from our goal of progressively adapting to new test samples of interest. We have experimented with EATA in global response #1. DIA [4] 's primary focus is designing an attack method for TTA, which we use as the attack scenario in our experiment (Section 4 of our manuscript). TTAB [2] and ODS [3] are tailored to address distributional shifts but do not consider non-interest samples. The focus of OOD methods [5, 6] is the detection of OOD samples within the same domain, which is not suitable for TTA scenarios. We have included the experimental results and limitations of OOD detection methods in response to your Question 3 below. We will include discussions on these studies in our final manuscript.
>
>
> **Question 1.**
>
> We assume the reviewer asked about our algorithm (SoTTA), not RoTTA.
>
> Our interpretation of this result is two-fold. First, our high-confidence uniform-class sampling strategy filters not only non-interest samples but also benign samples that would negatively impact the algorithm’s objective. This implies that there exist samples that are more beneficial for adaptation, which aligns with the findings that high-entropy samples harm adaptation performance [1]. Second, entropy-sharpness minimization helps ensure both robustness to non-interest samples and generalizability of the model by preventing model drifts, leading to performance improvement with benign samples. This is also consistent with SAR [r1], which argues that the entropy-sharpness minimization is robust to large gradients, which could harm the adaptation performance.
>
>
> **Question 2.**
>
> Thank you for your valuable suggestion. We include the sensitivity of the hyperparameter C0 and discussions in global response #2.
>
>
> **Question 3.**
>
> Following your suggestion, we conducted additional experiments using one of the famous OOD detection algorithms, ODIN [r2], in our noisy data streams. Specifically, we filtered OOD samples detected by ODIN and performed TTA algorithms on the samples left.
>
> Similar to prior studies on OOD, ODIN uses a thresholding approach to predict whether a sample is OOD. It thus requires validation data with binary labels indicating whether it is in-distribution or OOD to decide the best threshold. However, in TTA scenarios, validation data is not provided, which makes it hard to apply OOD algorithms directly in our scenario. We circumvented this problem using the labeled test batches to get the best threshold. Following the original paper, we searched for the best threshold from 0.1 to 0.12 with a step size of 0.000001, which took over 20,000 times longer than the original TTA algorithm.
>
> The result (Table 8 in the one-page PDF) shows that the impact of discarding OOD samples with ODIN (with-ODIN) is negligible, yielding only a 0.331% improvement in the average accuracy despite a huge computation cost.
>
> There exist practical limitations of OOD detection algorithms for TTA. (1) OOD methods assume that a model is fixed during test time, while a model changes continually in TTA. (2) As previously noted, most OOD algorithms require labels for validation data unavailable in TTA scenarios. Even using the same test dataset for selecting the threshold, the performance improvement was marginal. (3) Low performance possibly results from the fact that OOD detection studies are built on the condition that training and test domains are the same, which differs from TTA’s scenario. These collectively make it difficult to apply OOD detection studies directly to TTA scenarios. We discussed these limitations in Section 5 of our manuscript.
>
> We will include the whole result in our appendix in our final manuscript.
>
>
> [r1] Niu, Shuaicheng, et al. "Towards Stable Test-time Adaptation in Dynamic Wild World." ICLR ‘22
>
> [r2] Liang, Shiyu, Yixuan Li, and R. Srikant. "Enhancing The Reliability of Out-of-distribution Image Detection in Neural Networks." ICLR ‘18

---

> > ### Comment · Reviewer_6Z6q · 2023-08-15
> >
> > Your rebuttal makes sense. I hope you can add the corresponding results and include discussions on the listed studies in the final manuscript.
> >
> > I decide to raise my score.

---

> > > ### Author Response · Authors · 2023-08-16
> > > **Thank you for your response**
> > >
> > > Thank you for your response to our rebuttal! We will add the results and discussions in the rebuttal to the final manuscript.
> > >
> > > Thank you again for your valuable feedback and suggestions.
> > >
> > > Best,
> > >
> > > Authors.

---

### Official Review · Reviewer_gePD · 2023-07-06

**Soundness:** 2 fair
**Presentation:** 2 fair
**Contribution:** 2 fair
**Rating:** 4
**Confidence:** 4

**Summary:**

This article presents a new Test-Time Adaptation (TTA) scenario, wherein the model is adapted to noisy test streams. To address the challenges posed by this scenario, the paper introduces the Screening-out Test-Time Adaptation (SoTTA) algorithm, which leverages input-wise and parameter-wise robustness. The effectiveness of SoTTA is demonstrated through extensive comparison experiments conducted on TTA benchmarks.

**Strengths:**

1. This work proposes a novel test time adaptation setup, which considers the noisy data (non-interest) in real world application during test phase, whose motivation is convincing.
2. The significant performance gain shows SoTTA addresses the challenges under noisy test stream well.


**Weaknesses:**

1. The initial component of the method is inspired by the memory bank concept introduced in RoTTA, albeit in a simplified form. The second component, pioneered by SAR[1], represents a novel approach within the TTA domain. However, it is worth noting that the methodology section may lack originality.
2. The sensitivity analysis and selection criterion for key hyperparameters, specifically $m,C_0$ , are not included in the current study.
3. The motivation of this paper stems primarily from experimental findings, and it would be advantageous to analyze the motivation or methods from a theoretical perspective.
4. The paper contains several typographical errors, and it would be valuable to thoroughly revise it.

[1] Towards stable test-time adaptation in dynamic wild world


**Questions:**

1. What would be the outcome if CSTU and ESM were combined? Would the performance surpass that of SoTTA?
2. It is acknowledged that in real-world scenarios, the non-interest scenes mentioned in the experimental section may occur simultaneously. Have any relevant experiments been conducted to address this?


**Limitations:**

Actually, the problem setting is interesting to some extent, however, the techniques in this paper are similar to several compared methods, which may limit the novelty of this paper.

---

> ### Author Rebuttal · Authors · 2023-08-09
>
> We sincerely appreciate your time and effort in providing us with positive comments. We respond to your question in what follows. Please also refer to the *global response* we have posted together.
>
> ---
> **Weakness 1.**
>
> We appreciate the reviewer's insightful comment. Regarding the memory bank concept introduced in RoTTA, it is essential to note that the fundamental objective of RoTTA differs from our proposed method. RoTTA's memory bank primarily focuses on maintaining recent high-confidence samples. However, without a dedicated filtering-out method for low-confidence samples, RoTTA fails to avoid non-interest samples, especially in the early stage of test-time adaptation. In contrast, our confidence-based memory management scheme effectively rejects non-interest samples, and thus it rejects potential model drift from the beginning of TTA scenarios. As a result, our approach outperforms RoTTA in noisy test streams, as shown in Table 1 of our manuscript.
>
> Regarding the second component, we acknowledge that both SoTTA and SAR utilize sharpness-aware minimization proposed by Foret et al. [r1]. However, the motivation behind using SAM is different. While SAR intends to avoid model collapse when exposed to samples with large gradients, we aim to enhance the model's robustness to non-interest samples with high confidence scores. As illustrated in Figure 6 in the manuscript, we observed that entropy-sharpness minimization effectively prevents the model from overfitting to non-interest samples.
>
> **Weakness 2.**
>
> Thank you for your valuable suggestion. We include the sensitivity analysis of the hyperparameters at global response #2.
>
>
> **Weakness 3.**
>
> We appreciate your suggestion. The current manuscript highlights the practical issues of noisy data streams in real-world TTA scenarios from experimental findings and proposes a methodology to address this problem effectively.
>
> Here, we provide the theoretical motivation for handling noisy data streams in TTA scenarios. With the Bayesian-learning-based frameworks [r2, r3], we can express the posterior distribution of the model in terms of training ($D$) / benign test data ($B$) in test-time adaptation:
>
> Equation 1.
> $\log p(\theta | D, B) = \log q(\theta) - \frac{\lambda_B}{|B|} \sum_{b=1}^{|B|} H(y_b | x_b)$
>
> The posterior distribution of model parameters depends on the prior distribution ($q$) and the average of entropy ($H$) of benign samples with a certain weight ($\lambda$).
>
> Here, we incorporate the additional noisy data stream ($N$ into Equation 1 and introduce the new posterior distribution for our target scenario with noisy streams:
>
> Equation 2.
> $\log p(\theta | D, B, N) = \log q(\theta) - \frac{\lambda_B}{|B|} \sum_{b=1}^{|B|} H(y_b | x_b)  - \frac{\lambda_N}{|N|} \sum_{n=1}^{|N|} H(y_n | x_n)$
>
> We can now derive model parameter variations for benign and non-interest test samples:
>
> Equation 3.
> $\log p(\theta | D, B) - \log p(\theta | D, B, N)= \frac{\lambda_N}{|N|} \sum_{n=1}^{M} H(y_n | x_n)$
>
>
> Equation 3 implies that the (1) model adapted only from benign ($B$) and (2) model adapted with both benign ($B$) and noise ($N$) differs by the amount of the average entropy of noisy (non-interest) samples. The equation also suggests that a high entropy from severe noise would result in a significant model drift in adaptation.
>
> We appreciate the reviewer's constructive feedback. We will incorporate and discuss theoretical analysis further in our final manuscript.
>
>
>
> **Weakness 4.**
>
>
> Thank you for your careful examination of our paper. We apologize for the typographical errors in the initial submission, and we will thoroughly revise the manuscript.
>
>
>
> **Question 1.**
>
> Here, we compare the original SoTTA with the CSTU+ESM version (under the same setting as Table 1 in our manuscript).
>
> | | Benign | Near | Far | Attack | Noise | Avg |
> |---|:---:|:---:|:---:|:---:|:---:|:---:|
> | SoTTA | 82.9±0.4 | 81.4±0.5 | 81.6±0.5 | 84.5±0.3 | 81.0±1.5 | 82.3  |
> | CSTU+ESM | 83.0±0.2 | 81.0±0.0 | 78.5±0.2 | 83.7±0.2 | 78.7±0.9 | 81.0 |
>
> The above table shows that CSTU shows comparable accuracy to SoTTA (HUS+ESM) without non-interest samples (Benign). However, CSTU shows 2.3x higher performance degradation with Noise samples (- 4.3%p) than SoTTA (-1.9%p). This result originates from CSTU's structure of only maintaining the "recent high-confidence samples." CSTU does not discard any low-confident samples when the memory is not full, which is susceptible to harmful samples in the early adaptation stage.
>
>
> **Question 2.**
>
> Our methodology filters non-interest samples, which could be easily extended to simultaneous non-interest scenes. We expect to further develop robust TTA methods in more complex settings, such as mixed-domain, shuffled-class scenarios [r4].
>
>
>
> [r1] Foret, Pierre, et al. "Sharpness-aware Minimization for Efficiently Improving Generalization." International Conference on Learning Representations. 2020.
>
> [r2] Brahma, Dhanajit, and Piyush Rai. "A Probabilistic Framework for Lifelong Test-Time Adaptation." Proceedings of the IEEE/CVF Conference on Computer Vision and Pattern Recognition. 2023.
>
> [r3] Yves Grandvalet and Yoshua Bengio. "Semi-supervised learning by entropy minimization." Advances in neural information processing systems, 17, 2004.
>
> [r4] Marsden, Robert A., Mario Döbler, and Bin Yang. "Universal Test-time Adaptation through Weight Ensembling, Diversity Weighting, and Prior Correction." arXiv preprint arXiv:2306.00650 (2023).

---

> > ### Comment · Reviewer_gePD · 2023-08-17
> > **Official Comment by Reviewer gePD**
> >
> > Thanks for the detailed response, and some of my concerns have been addressed. Indeed, the test time setting on Noisy Data Streams is interesting and challenging. However, I still believe the techniques in this paper are similar to several SOTA TTA methods, and I will keep my score.

---

> > > ### Author Response · Authors · 2023-08-17
> > > **Thank you for your response**
> > >
> > > We are pleased that our rebuttal addressed your concerns. We also appreciate your acknowledging that our problem setting is interesting and challenging. Regarding the techniques, we still believe that our method is carefully designed and has novel components to tackle the challenging problem, as we wrote in our rebuttal. As a result, our method advances SOTA TTA methods with notable performance improvements (e.g., on average, +14.58%p better than SAR and +5.82%p better than RoTTA on CIFAR10-C).
> > >
> > > Thank you again for your response. We will strengthen our paper based on your valuable comments and feedback.
> > >
> > > Best,
> > >
> > > Authors

---

### Official Review · Reviewer_kiEt · 2023-07-06

**Soundness:** 3 good
**Presentation:** 3 good
**Contribution:** 3 good
**Rating:** 7
**Confidence:** 5

**Summary:**

This paper proposes screening-out test-time adaptation which is claimed robust to non-interest samples. It filters out the impact of non-interest samples with a high-confidence uniform-class sampling. It proposes entropy-sharpness minimization to deal with large gradients. Experiments are completed on CIFAR10-C and ImageNet-C.

**Strengths:**

1. The proposed problem (screening-out test-time adaptation) is novel and sharp.
2. This paper has a good presentation. For example, Fig.1 and Fig.2 are very clear.
3. Entropy-sharpness minimization seems reasonable as discussed in Section 3.2.
4. Experiments are completed on both small and large datasets including ImageNet-C. As shown in Tab.1, the improvement is good enough.

**Weaknesses:**

1. The problem is interesting but the proposed two methods are straightforward. I am a little worried about the technical novelty.
2. The design of input-wise robustness is hard coding. Although some samples are bad for optimization, they may be essential for the perception.
3. Only classification is completed. However, segmentation is also very important in autonomous driving scenarios.

**Questions:**

1. Can it also work on segmentation and detection? Please discuss.

**Limitations:**

Please refer to "weakness".

---

> ### Author Rebuttal · Authors · 2023-08-09
>
> We sincerely appreciate your time and effort in providing us with positive comments. We respond to your question in what follows. Please also refer to the *global response* we posted together.
>
> ---
> **Weakness 1. The problem is interesting but the proposed two methods are straightforward. I am a little worried about the technical novelty.**
>
> We appreciate the reviewer's insightful comment. Regarding high-confidence uniform sampling, RoTTA could be compared with our method. Regarding the memory bank concept introduced in RoTTA, it is essential to note that the fundamental objective of RoTTA differs from our proposed method. RoTTA's memory bank primarily focuses on maintaining recent high-confidence samples. However, without a dedicated filtering method for low-confidence samples, RoTTA fails to avoid non-interest samples, especially in the early stage of test-time adaptation. In contrast, our confidence-based memory management scheme effectively rejects non-interest samples, and thus it rejects potential model drift from the beginning of TTA scenarios. As a result, our approach outperforms RoTTA in noisy test streams, as shown in Table 1 of our manuscript.
>
> Regarding the second component, we acknowledge that both SoTTA and SAR utilize sharpness-aware minimization proposed by Foret et al. [r1]. However, we clarify that the motivation behind using SAM is different. While SAR intends to avoid model collapse when exposed to samples with large gradients, we aim to enhance the model's robustness to non-interest samples with high confidence scores. As illustrated in Figure 6 in the manuscript, we observed that entropy-sharpness minimization effectively prevents the model from overfitting to non-interest samples.
>
>
>
>
> **Weakness 2. The design of input-wise robustness is hard coding. Although some samples are bad for optimization, they may be essential for the perception.**
>
> Low-confidence samples could be beneficial for the perception. However, we want to note the risks of model corruption associated with including such low-confidence samples in the selection. In our discussion of the effect of hyperparameters, as detailed in the global response #2, we observed that lowering the confidence threshold (C0) resulted in performance degradation (e.g., 81.0% (C0=0.99) -> 77.7% (C0=0.7) in Noise scenario; see Figure 8 in the attached PDF). This observation supports the need for a cautious approach in dealing with low-confidence samples. In light of your feedback, we will incorporate this discussion of the trade-off between the perception and risks of model corruption in our filtering strategy into the manuscript.
>
>
>
> **Weakness 3. Only classification is completed. However, segmentation is also very important in autonomous driving scenarios. / Question 1. Can it also work on segmentation and detection? Please discuss.**
>
> We appreciate your insightful comment. While our work primarily focuses on classification tasks, we acknowledge the importance of image segmentation and object detection, particularly in autonomous driving scenarios.
>
> For image segmentation, when non-interest objects are present in the input, the model might produce noisy predictions on those pixels, leading to detrimental results. Extending our method to operate at the pixel level would allow it to be compatible with the segmentation task while minimizing the negative influences of those noisy pixels on model predictions in test-time adaptation scenarios.
>
> Similarly, our method could be tailored to object detection's classification (recognition) task. For example, in the context of the YOLO framework, our method could filter and store grids with high confidence for test-time adaptation, enhancing detection accuracy. However, our current approach must address the localization task (bounding box regression) during test-time adaptation. Implementing this feature is non-trivial and would require careful consideration and potential redesign of certain aspects of our methodology. Accurately localizing bounding boxes during test-time adaptation presents an exciting avenue for future research.
>
> We sincerely appreciate the reviewer's insightful comment and will incorporate this discussion into our manuscript.

---

### Author Rebuttal · Authors · 2023-08-09

## Global Response

Dear Reviewers,

We sincerely appreciate your efforts and time in reviewing our manuscript.

The contribution of our work lies in investigating the crucial yet unexplored challenge of test sample diversity in real-world scenarios. Notably, we unveil that existing TTA algorithms suffer from significant performance degradation with such noisy streams. As a solution, we propose SoTTA that is robust to the sample diversity and outperforms the state-of-the-art baselines.


We appreciate the reviewer's constructive and insightful comments on our paper. To answer your concerns and questions, we have written the responses with the following additional experiments and clarification:


Additional experiments on EATA and RoTTA to clarify the differences between the methods and our approach (6Z6q, HUhd, gePD)
Sensitivity analysis of the hyperparameters (gePD, 6Z6q)
Clarification of the effectiveness of our input-wise robustness method compared with other existing approaches (kiEt, gePD)
Clarification of the difference between SoTTA and SAR (gePD, HUHd)
Theoretical analysis regarding the motivation for handling noisy data streams in TTA scenarios (gePD)
Additional experiment to compare our method with an OOD detection method (6Z6q)
Clarification of our evaluation scenario with CIFAR100 (6Z6q)
Comparison between our method and relevant TTA methods to clarify the focus of our work (6Z6q)
Discussion of the possibilities of applying our methods in broader scenarios such as image segmentation and object detection (kiEt)

Additionally, we summarize two major issues as follows:

### 1. Comparison with EATA [c1] (Reviewer 6Z6q and HUhd)

Two reviewers (6Z6q and HUhd) identified EATA [c1] as a critical baseline for its similarity in sample selection with our approach. In response, we evaluated EATA under two conditions on our noisy data stream scenario with CIFAR10-C: (1) EATA, where we applied the method directly to the noisy data stream, encompassing Fisher importance calculation and model adaptation in the presence of non-interest samples, and (2) EATA-Clean, where we gave EATA an advantage by excluding the non-interest samples when calculating the Fisher information matrix. We ensured that the experimental settings remained consistent with those described in our original submission, as noted in Table 1 of our manuscript.

The experiment results are presented in the table below:

| | Benign | Near | Far | Attack | Noise | Avg |
|---|:---:|:---:|:---:|:---:|:---:|:---:|
| SoTTA | 82.9±0.4 | 81.4±0.5 | 81.6±0.5 | 84.5±0.3 | 81.0±1.5 | 82.3 |
| EATA-Clean | 82.4±0.2 | 64.4±1.5 | 57.6±2.6 | 70.9±0.7 | 36.6±1.2 | 62.4 |
| EATA | 82.4±0.2 | 63.9±0.4 | 56.3±0.5 | 70.9±0.6 | 36.0±0.8 | 61.9 |

From the results, we observed a significant accuracy degradation for EATA in the presence of non-interest samples, dropping from 82.4% to 36.6% when confronted with noisy data (Noise). Our proposed SoTTA algorithm utilizes high-confidence sampling, effectively excluding a more significant number of non-interest samples. SoTTA filtered out 99.98% of non-interest samples, whereas EATA only excluded 33.55% in the memory-filling stage.

In addition, EATA calculates the Fisher importance based on a few initial test-time samples (e.g., 2000 samples [c1]) to calculate parameter importance and prevent catastrophic forgetting. This approach becomes problematic as including non-interest samples in the initial sample set corrupts the Fisher importance values, resulting in a degradation of accuracy (EATA-Clean 62.4% -> EATA 61.9%).



### 2. Effect of hyperparameters C0, m (Reviewer gePD and 6Z6q)

In response to the reviewers' (gePD and 6Z6q) request for the sensitivity analysis of hyperparameters, we conducted experiments to present the results for two specific hyperparameters: confidence threshold (C0) and BN momentum (m). Notably, we varied C0 within the range of [0.7, 0.999] and m within [0.05, 0.3] and reported the corresponding accuracy. Please refer to the detailed results in the attached PDF file.


Confidence threshold (C0) (Figure 8a in the one-page PDF):

Our result shows that the selection of C0 shows similar patterns across different scenarios (Benign ~ Noise). The result illustrates a tradeoff; a low C0 value does not effectively reject non-interest samples, while a high C0 value filters benign data. We found a proper value of C0 (0.99) that generally works well across the scenarios. Also, we found that the optimal C0 depends primarily on in-distribution data. Our interpretation is that setting different C0 values for CIFAR10 and ImageNet is straightforward as they have a different number of classes (10 vs. 1000), which leads to different ranges of the model’s confidence.


BN momentum (m) (Figure 8b in the one-page PDF):

Across the tested range, the variations in performance were found to be negligible. This finding indicates that choosing a low momentum value from within the specified range ([0.05, 0.3]) is adequate to maintain a favorable performance. Please note that setting a high momentum would corrupt the result, which is implicated by the algorithms directly utilizing test-time statistics (e.g., TENT) suffering from accuracy degradation with noisy data streams (e.g., TENT: 81.0% -> 52.1% for Noise at Table 1 in the manuscript).


We will incorporate these findings and discussions into the final manuscript. Thank you again for your thoughtful review, and we are open to any further suggestions or questions you may have.

[c1] Shuaicheng Niu, Jiaxiang Wu, Yifan Zhang, Yaofo Chen, Shijian Zheng, Peilin Zhao, Mingkui Tan: Efficient Test-Time Model Adaptation without Forgetting. ICML 2022: 16888-16905

---

### Decision · Program_Chairs · 2023-09-21

**Decision:**

Accept (poster)

**Comment:**

This work on test-time adaptation focuses on adaptation in the presence of noisy/irrelevant/adversarial data: it contributes an experimental protocol for this setting, comprehensive experiments, and a technically straightforward but effective method (SoTTA). Four expert reviewers with backgrounds on adaptation/generalization, open world modeling, and data-efficient learning mainly vote for acceptance (kiEt: 7, HUhd: 5, 6Z6q: 5) with one vote for rejection (gePD: 4). The authors provided a rebuttal, three of the four reviewers replied, and scores and two of three borderline scores were raised (6Z6q, HUhd). SoTTA shares common elements with recent more robust adaptation methods (especially RoTTA and SAR), but it does differ in (arguably minor) details that nevertheless improve results. This is demonstrated in comparison with strong and up-to-date baselines (including recent methods like SAR). At the same time, the experiments are informative about the more "open world" setting, which is an important direction of progress for test-time adaptation. The AC sides with acceptance in alignment with the majority of reviewers and the positive direction of the scores after rebuttal. The AC encourages the authors to incorporate the feedback from reviewers, and in particular emphasizes the inclusion of the sensitivity analysis (gePD, 6Z6q) and the discussion of TTA methods that are closely related in technique or setting (gePD, 6Z6q: EATA, ODS, ...). Although some of these papers are concurrent, referencing and discussing such work is more informative to the reader and makes for a more connected community.

Note: the AC asks the authors to reflect on their choice of "non-interest" terminology, and consider aligning with existing terminology such as "open world" or "distractor" samples, so that this work is more accessible to the existing audiences for this topic.